# *Salmonella* Typhimurium effector SseI regulates host peroxisomal dynamics to acquire lysosomal cholesterol

Desh Raj[1,2], Abhilash Vijay Nair[3], Anmol Singh[3], Swarnali Basu[1], Kabita Sarkar[1], Jyotsna Sharma [1,2], Shiva Sharma[1,2], Sanmi Sharma[1], Manisha Rathore[4], Shriya Singh [5], Shakti Prakash [1,2], Simran [2,6], Shikha Sahu[7], Aman Chandra Kaushik[8], Mohammad Imran Siddiqi[2,8], Uday C Ghoshal [7], Tulika Chandra[9], Vivek Bhosale[2,10], Arunava Dasgupta[2,5], Shashi Kumar Gupta[1,2], Sonia Verma [2,6], Rajdeep Guha [2,4], Dipshikha Chakravortty [3✉], Veena Ammanathan [1✉] & Amit Lahiri [1,2✉]

## Abstract

***Salmonella enterica* serotype Typhimurium (*Salmonella*) resides and multiplies intracellularly in cholesterol-rich compartments called *Salmonella*-containing vacuoles (SCVs) with actin-rich tubular extensions known as *Salmonella*-induced filaments (SIFs). SCV maturation depends on host-derived cholesterol, but the transport mechanism of low-density lipoprotein (LDL)-derived cholesterol to SCVs remains unclear. Here we find that peroxisomes are recruited to SCVs and function as pro-bacterial organelle. The *Salmonella* effector protein SseI is required for the interaction between peroxisomes and the SCV. SseI contains a variant of the PTS1 peroxisome-targeting sequence, GKM, localizes to the peroxisomes and activates the host Ras GTPase, ADP-ribosylation factor-1 (ARF-1). Activation of ARF-1 leads to the recruitment of phosphatidylinsolitol-5-phosphate-4 kinase and the generation of phosphatidylinsolitol-4-5-bisphosphate on peroxisomes. This enhances the interaction of peroxisomes with lysosomes and allows for the transfer of lysosomal cholesterol to SCVs using peroxisomes as a bridge. *Salmonella* infection of peroxisome-depleted cells leads to the depletion of cholesterol on the SCVs, resulting in reduced SIF formation and bacterial proliferation. Taken together, our work identified peroxisomes as a target of *Salmonella* secretory effectors, and as conveyance of host cholesterol to enhance SCV stability, SIF integrity, and intracellular bacterial growth.**

**Keywords** Peroxisomes; *Salmonella* Typhimurium; Cholesterol; Peroxisome-targeting Sequence; ARF1 Activation
**Subject Categories** Membranes & Trafficking; Microbiology, Virology & Host Pathogen Interaction; Organelles

## Introduction

Enteric pathogen *Salmonella enterica* serotype Typhimurium (*Salmonella*/STM), the causative agent of gastroenteritis in humans, can invade and survive within macrophages, as well as epithelial cells and replicate in distinct membrane-bound vesicles known as *Salmonella*-containing vacuoles (SCVs) (Feasey et al, 2012). They develop tubular extensions called *Salmonella*-induced filaments (SIFs) that are important for the growth and survival of *Salmonella* in the host cells (Knuff and Finlay, 2017). SCVs are unique compartments that selectively acquire late endocytic markers like LAMP1/2 and exclude specific markers like mannose-6-phosphate receptors (M6PR) (Garcia-del Portillo and Finlay, 1995; Kehl et al, 2020). This leads to SCVs having a membrane composition similar to lysosomes but devoid of hydrolytic enzymes delivered by M6PR (McGourty et al, 2012). Formation and maintenance of SCV require the expression of bacterial effector proteins encoded by type three secretion systems (TTSS), namely *Salmonella* pathogenicity islands 1/2 (SPI1/2) (Galan 2001; Chakravortty et al, 2005). While SPI1 TTSS is required for the entry of the bacteria in the epithelial cells, SPI2 TTSS leads to the secretion of more than 30 proteins across the SCV membrane in both epithelial cells and macrophages (Miao and Miller, 2000; Miao et al, 2003). SPI2 TTSS also promotes intracellular replication of *Salmonella* by employing multiple strategies to subvert host responses such as altering antibacterial peptide secretion, remodeling of cytoskeleton, perturbing endo-lysosomal trafficking, interference with lysosomal activity, and acquiring metabolites required for bacterial growth

[1]Pharmacology Division, CSIR-Central Drug Research Institute, Lucknow, India. [2]Academy of Scientific and Innovative Research (AcSIR), Ghaziabad, India. [3]Department of Microbiology and Cell Biology, Indian Institute of Science, Bangalore, India. [4]Laboratory Animal Facility Division, CSIR-Central Drug Research Institute, Lucknow, India. [5]Molecular Microbiology and Immunology Division, CSIR-Central Drug Research Institute, Lucknow, India. [6]Neuroscience & Ageing Biology Division, CSIR-Central Drug Research Institute, Lucknow, India. [7]Department of Gastroenterology, Sanjay Gandhi Postgraduate Institute of Medicine, Lucknow, India. [8]Biochemistry and Structural Biology Division, CSIR-Central Drug Research Institute, Lucknow, India. [9]Department of Transfusion Medicine, King Georges' Medical University, Lucknow, India. [10]Toxicology and Experimental Medicine Division, CSIR-Central Drug Research Institute, Lucknow, India. ✉E-mail: dipa@iisc.ac.in; veena.pdf@cdri.res.in; amit.lahiri@cdri.res.in

(Chakravortty et al, 2002; Prost and Miller, 2008; D'Costa et al, 2015; Jennings et al, 2017; Knuff-Janzen et al, 2020).

Cholesterol accumulates excessively (about 30% of total cellular cholesterol) around SCV and is essential for its stability and SIF formation (Catron et al, 2002; Nawabi et al, 2008; Kolodziejek et al, 2019). Interestingly, *Salmonella* cannot synthesize cholesterol, and the sterol that accumulates in SCV is not of biosynthetic origin, highlighting the need for studies to understand cellular cholesterol transport to SCV (Garner et al, 2002; Catron et al, 2004; Cain et al, 2005; Nawabi et al, 2008). Studies have shown that two TTSS effector proteins, SseJ and SseL, are involved in the cholesterol transport to SCV (Ohlson et al, 2005). These proteins are shown to recruit the host cholesterol transport protein, oxysterol binding protein 1 (Osbp1), to the cytosolic surface of SCV, facilitating non-vesicular transport of cholesterol to SCV (Kolodziejek et al, 2019). However, the mechanism by which *Salmonella* acquires cellular cholesterol during SCV maturation is unclear.

As cholesterol is a critical component of membranes, its cellular concentration and trafficking are tightly regulated (Radulovic et al, 2022; Naito et al, 2023; Zhou et al, 2023). Cholesterol homeostasis is maintained by balanced uptake of exogenous low-density lipoprotein (LDL) and de novo biosynthesis in the ER. A study by Gilk et al, showed that *Salmonella* can invade and replicate intracellularly even without 24-Dehydrocholesterol Reductase (DHCR24), an enzyme involved in the de novo cholesterol synthesis (Gilk et al, 2013). This indicates that LDL-derived cholesterol and not de novo synthesis is essential for *Salmonella* growth (Gilk et al, 2013). Exogenous LDL, after internalization, gets hydrolyzed in the lysosomes to release cholesterol, which is later conveyed to other organelles (Chu et al, 2015). For instance, lysosomes form protein bridges with ER and peroxisomes to transport cholesterol (Du et al, 2011; Chu et al, 2015). While some studies report delayed fusion of SCV with lysosomes (Buchmeier and Heffron, 1991; Brumell et al, 2001), others show that the potency of lysosomes is reduced in the *Salmonella*-infected cells (Eswarappa et al, 2010; McGourty et al, 2012). Therefore, it is still controversial whether SCV does or does not fuse with active/potent lysosomes. Here, we questioned how LDL-derived cholesterol can be transported from lysosomes to the growing SCV.

Peroxisomes are cellular organelles containing abundant oxidative enzymes found in all eukaryotic cells. Peroxisomes acquire most of their proteins by selective import from cytosol containing a short signal sequence at the C-terminus (s/a-k/r-l/m) known as peroxisomal targeting sequence 1 (PTS1) (Keller et al, 1987; Gould et al, 1989). Recent studies indicate the role of peroxisomes in immune cell activation, inflammation, and cholesterol transport (Chu et al, 2015; Di Cara et al, 2017; Di Cara, 2020; Nath et al, 2022). Mapping the organelle contacts revealed that peroxisomes are predominantly in close contact with ER, mitochondria, lipid droplets, and lysosomes (Shai et al, 2016; Shai et al, 2018). In addition, peroxisomes can interact with lysosomes to acquire LDL-derived cholesterol. This is mediated by forming membrane contacts between peroxisomal phosphatidyli-nositol (4,5) phosphate (PI (4,5) P2 or PIP2) and Synaptotagmin-7 (Syt7) present on the lysosomes. Cholesterol is then redistributed from the peroxisomes to ER by tethering PIP2 and Extended-synaptotagmin-1 (E-Syt1) expressed on the ER (Xiao et al, 2019; Thallmair et al, 2023; Chu et al, 2015; Lee and Bensinger, 2022).

In the current study, we report that peroxisomes play a crucial role in *Salmonella* growth in both human primary macrophages and

epithelial cells. We have identified a SPI2 TTSS effector of *Salmonella*, SseI, to contain a putative PTS1-like signal. SseI is a 37 kDa secretory protein, and previous studies have implicated its role in deciding the migration of *Salmonella*-infected dendritic cells (Bhaskaran and Stebbins, 2012; Carden et al, 2017; Brink et al, 2018). We observed that SseI gets recruited to the peroxisomes and activates a host small GTPase, ADP-ribosylation factor-1 (ARF1). Activation of ARF1 leads to enhanced peroxisomal PIP2 levels. Subsequently, this causes higher lysosome-peroxisome and peroxisome-SCV contacts, providing SCV and the growing SIF the necessary cholesterol. Per our understanding, this is the first report of a bacterial protein having a functional PTS1-like signal, by which *Salmonella* reroute host lysosomal cholesterol using peroxisome as a bridge.

# Results

## Live *Salmonella* Typhimurium temporally interacts with peroxisomes in the epithelial cells and primary human macrophages

Since little is known about the role of peroxisomes during *Salmonella* infection, we initiated our study by monitoring the dynamics of host peroxisomes (peroxisomal number and interaction with bacterial vacuoles) during the progression of *Salmonella* Typhimurium 14028s (*Salmonella*/ STM) infection. Remarkably, we observed the interaction of SCV (co-stained by LAMP1 and bacteria) with peroxisomes (marked by PEX14) in the human HeLa epithelial cell line. This interaction between SCV and peroxisomes increased with time, starting from 3 h of infection to 12 h (Fig. 1A,C). We confirmed this temporal interaction between the organelles by also probing with additional markers such as Rab7 (co-stained with bacteria to label SCV) and catalase (to label peroxisomes) (Fig. 1B,D). Next, we performed live cell imaging of HeLa cells expressing 3xMyc-EGFP-PEX26 infected with STM to understand this dynamic interaction better. Cells were also loaded with lysotracker to label acidic vesicles, including SCVs. As shown in Fig. 1E and Movie EV1, multiple dynamic colocalizations of peroxisomes with SCVs as well as lysosomes were observed throughout the experimental setup.

As *Salmonella* infects both epithelial cells and macrophages physiologically, we next studied this interaction in human blood monocyte-derived macrophages (MDM). Interestingly, we could also visualize the STM-peroxisome colocalization in the human primary macrophages as well. To better understand the significance of this association, MDM were next infected for 6 h with either the heat-killed STM or inert latex bead (Fig. 1F). Evidently, the interaction between peroxisomes and SCVs was lost in all the conditions except with live virulent STM infection (Fig. 1G). Overall, we conclude that live STM infection leads to peroxisome-SCV contact formation in both epithelial HeLa cell line and human primary macrophages.

## Peroxisomes are required for efficient intracellular replication of *Salmonella* Typhimurium

Previous studies reported that viruses like HCMV, HSV-1, and SARS-CoV-2 increase peroxisome biogenesis with a concomitant rise in the peroxisomal protein levels (Jean Beltran et al, 2018; Knoblach et al, 2021). A recent study also showed that *Mycobacterium* tuberculosis infection leads to the induction of peroxisome biogenesis (Behera et al, 2022).

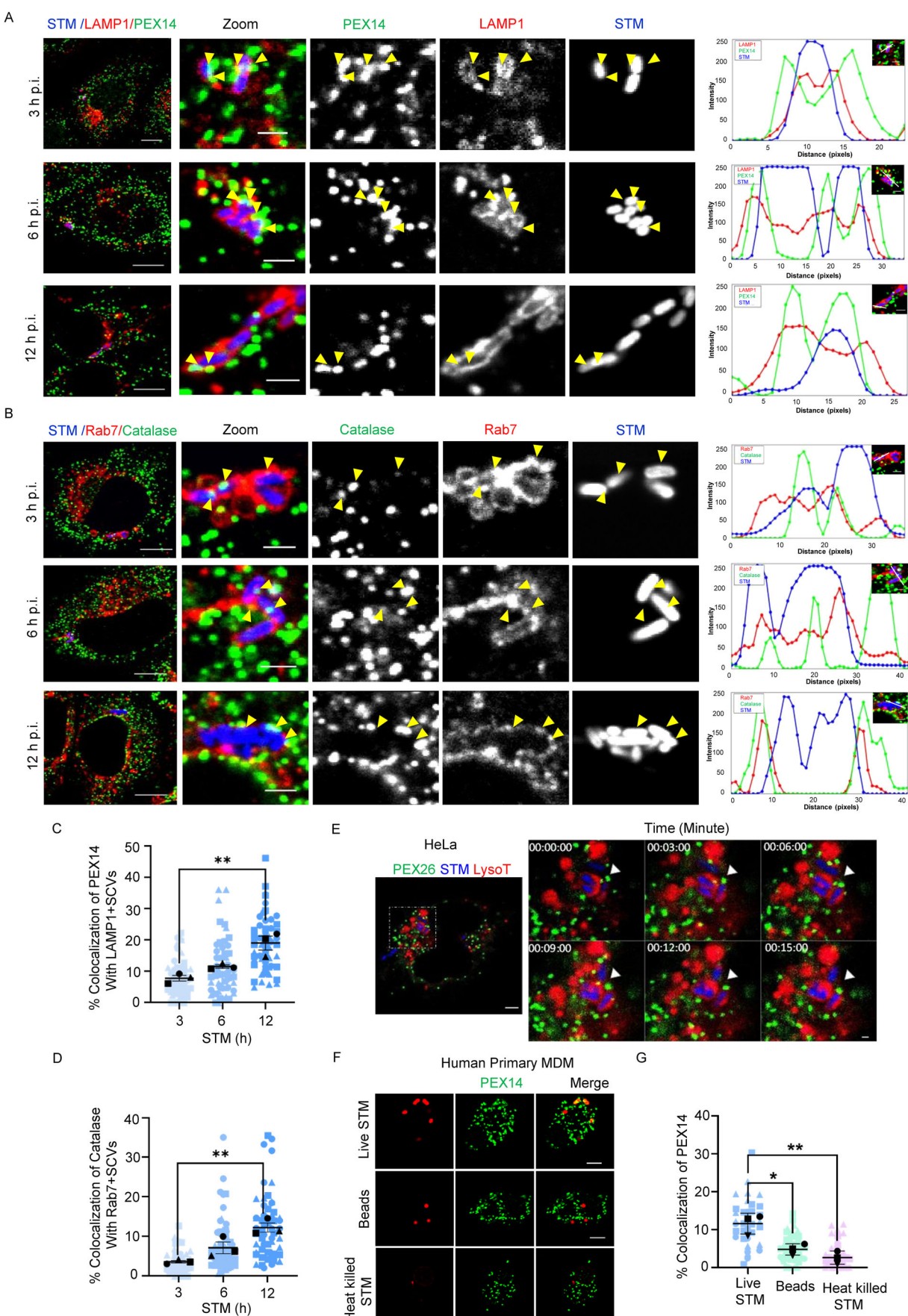

**Figure 1. Live STM temporally interacts with peroxisomes.**

(A) HeLa cells were infected with GFP-expressing STM (blue) and immunostained for LAMP1 (red) and PEX14 (green). Representative confocal images of single Z-planes are shown at 3, 6, and 12 h post-infection. Yellow arrowheads indicate colocalization of PEX14 with SCVs. Scale bars: 10 μm (main panel), 2 μm (zoom). The intensity profile shows the overlap of green and red fluorescence signals. (B) HeLa cells were infected with GFP-expressing STM (blue) and immunostained for Rab7 (red) and catalase (green). Representative confocal images of single Z-planes are shown at 3, 6, and 12 h post-infection. Yellow arrowheads indicate colocalization of catalase with SCVs. Scale bars: 10 μm (main panel), 2 μm (zoom). The intensity profile shows the overlap of green and red fluorescence signals. (C) Graph showing the percentage of PEX14 colocalization with SCVs over time. Data for three independent experiments containing more than 180 cells are shown. (D) Graph indicating the percentage colocalization of catalase with SCVs is shown. Data for three independent experiments containing more than 180 cells are shown. (E) and EV-1. Live cell imaging of HeLa cells stably expressing PEROXO-tag (3xMyc-EGFP-PEX26) infected with mCherry-STM (MOI = 50). Cells were also treated with 100 nM of LysoTracker (blue) for 15 min before imaging to label acidic vacuoles. Time-lapse imaging was performed 5 h post-infection. Images were captured approximately every minute (80 s) till 15 min. White arrowheads indicate regions of interaction. Images shown are single Z slices for the indicated time points. Scale bars: 10 μm. (F) Microscopy images of MDM infected with live or heat-killed STM or latex beads (red) for 6 h. The cells were stained with anti-PEX14 antibody (green). Representative confocal images of single Z-planes are shown. Scale bar: 5 μm. (G) Graph indicating the percentage colocalization of PEX14 with STM is shown. Data for three independent experiments containing more than 180 cells are shown. Data information: Data were analyzed using one-way ANOVA (Sidak's multiple comparisons test), (C) (**$p = 0.0028$), (D) (**$p = 0.0028$), and (G) (*$p = 0.0134$ and **$p = 0.0035$). Source data are available online for this figure.

Therefore, the potential of STM to alter peroxisome biogenesis was investigated next. Proteomic analysis performed during the early time point (6 h) of STM infection in HeLa cells showed no change in the peroxisome-associated proteins (31 peroxisomal proteins out of 3412 cellular proteins detected). It includes proteins related to the de novo biogenesis (PEX3), peroxisomal fission (PEX11B), membrane (PEX14, ATP Binding Cassette Subfamily D Member 1 (ABCD1)), and matrix (catalase) of peroxisomes (Fig. 2A; Appendix Fig. S1A). Similarly, during the later time point of infection (12 h), there was no significant increase in the peroxisome biogenesis proteins. Protein expression analysis by immunoblotting for PEX14 and PEX3 did not change (Fig. EV1A,B) validating proteomic results. Expression of peroxisome biogenesis-related transcripts such as *PEX14*, *PEX5*, and *PEX11B* were not altered after 6 and 12 h of STM infection (Fig. 2B–D). Similarly, the activity of the most abundant peroxisomal enzyme, catalase, did not change after infection, similar to proteomic analysis (Fig. 2E). In addition, the peroxisome numbers (Fig. 2F) also remained unaltered after STM infection. Taken together, these results indicate that this interaction between STM and peroxisomes occurs without any increase in the number of peroxisomes. However, certain peroxisomal enzymes of plasmalogen synthesis (FAR1, AGPS) and fatty acid oxidation (ACAA1, HSD17B4) were significantly altered in both proteomics and expression analysis (Fig. EV1C,D; Appendix Fig. S1B,C).

This altered temporal interaction could either benefit the host by inducing antibacterial pathways or facilitate bacterial survival. To discriminate between these two possibilities, we silenced *PEX14* in human MDM and HeLa cells. The *PEX14* silencing was validated by measuring both mRNA and protein levels (Fig. EV1E; Appendix Fig. S1D,E). These cells also exhibited reduced peroxisomal abundance (Appendix Fig. S1F,G). In addition, we measured the import efficiency of matrix proteins by using EGFP-SKL. SKL is a prototype PTS1 motif that is efficiently and exclusively transported to peroxisomes (Gould et al, 1989). As seen in the figure, diffused EGFP-SKL was observed, indicating its cytoplasmic localization and impairment in the import of PTS1-containing matrix proteins (Appendix Fig. S1H). In these *PEX14* deficient MDM and HeLa cells, we observed reduced bacterial proliferation (Fig. 2G,H). To further verify if peroxisomes facilitate STM growth, we generated a *PEX5* knockout cell line (*PEX5* KO) using CRISPR/Cas9 in the HeLa cells (Fig. EV1F). PEX5 is the receptor protein that binds to proteins with PTS1 and helps them shuttle to the peroxisomal matrix. We validated knockout of *PEX5* in multiple ways. Firstly, we observed that the number of peroxisomes labeled by PEX14 is

decreased in the *PEX5* deficient cells. Those remnant peroxisomes are also bigger, typical of dysfunctional peroxisomes observed in the PEX5 KO cells (Titorenko and Terlecky, 2011; Ott et al, 2023) (Fig. EV1G). In addition, we also observed impairment in the import efficiency of matrix proteins to peroxisomes (labeled by ABCD1) in the generated *PEX5* KO cells (Fig. EV1H,I). Similar results were also obtained while staining a peroxisomal matrix protein, catalase which remained majorly cytosolic in the *PEX5* deficient cells (Appendix Fig. S1I). Interestingly, these cells, which are devoid of functional peroxisomes, showed reduced STM replication (Fig. 2I) without any change in the bacterial invasion (Appendix Fig. S1J). Similarly, there was no change in the host LAMP1 levels after infection in the *PEX5* KO cells (Appendix Fig. S1K). In addition, treatment with a chemical inducer of peroxisome biogenesis, 4-Phenylbutyric acid (4-PBA) displayed enhanced bacterial replication (Fig. 2J). Treatment with 4-PBA heightened both mRNA and protein levels of PEX14 as well as induced peroxisomal abundance in cells (Fig. EV1J; Appendix Fig. S1L–N) These results suggest that functional peroxisomes are required for intracellular STM proliferation.

We next used *Caenorhabditis elegans* (*C. elegans*) as the model system for STM infection. Here, bacteria exist in the intestinal lumen to establish colonization and do not enter the intestinal cells. SPI-2 secretion apparatus (encoded by SsaV) is rather dispensable during *Salmonella* infection in *C. elegans* (Labrousse et al, 2000; Sem and Rhen, 2012). This is because *Salmonella* is not present in the vacuoles; therefore, bacterial genes involved in vacuolar growth do not play a significant role. In this model system, where STM is not present in the SCVs, we see that silencing *Prx5* (homolog of mammalian *PEX5*) did not affect the growth of bacteria (Fig. EV1K). These results led us to hypothesize that peroxisomes are essential in the vacuolar replication of intracellular STM, and a functional SPI2 system is probably involved in this interaction.

Next, to verify if peroxisomes contribute to the formation of SCVs, we infected *PEX5* KO HeLa cells, and the formation of SIFs post-infection was studied temporally. Our results indicate that the area occupied by SCV containing SIFs was drastically reduced in the *PEX5* KO HeLa cells compared to the control (WT) HeLa cells (Fig. 2K,L). None of the siRNA or inhibitors tested altered cell viability (Fig. EV1L–N). Collectively, it can be deduced that peroxisomes help in the formation of filaments that are essential during the maturation of SCV and bacterial growth, thereby serving as a pro-*Salmonella* organelle.

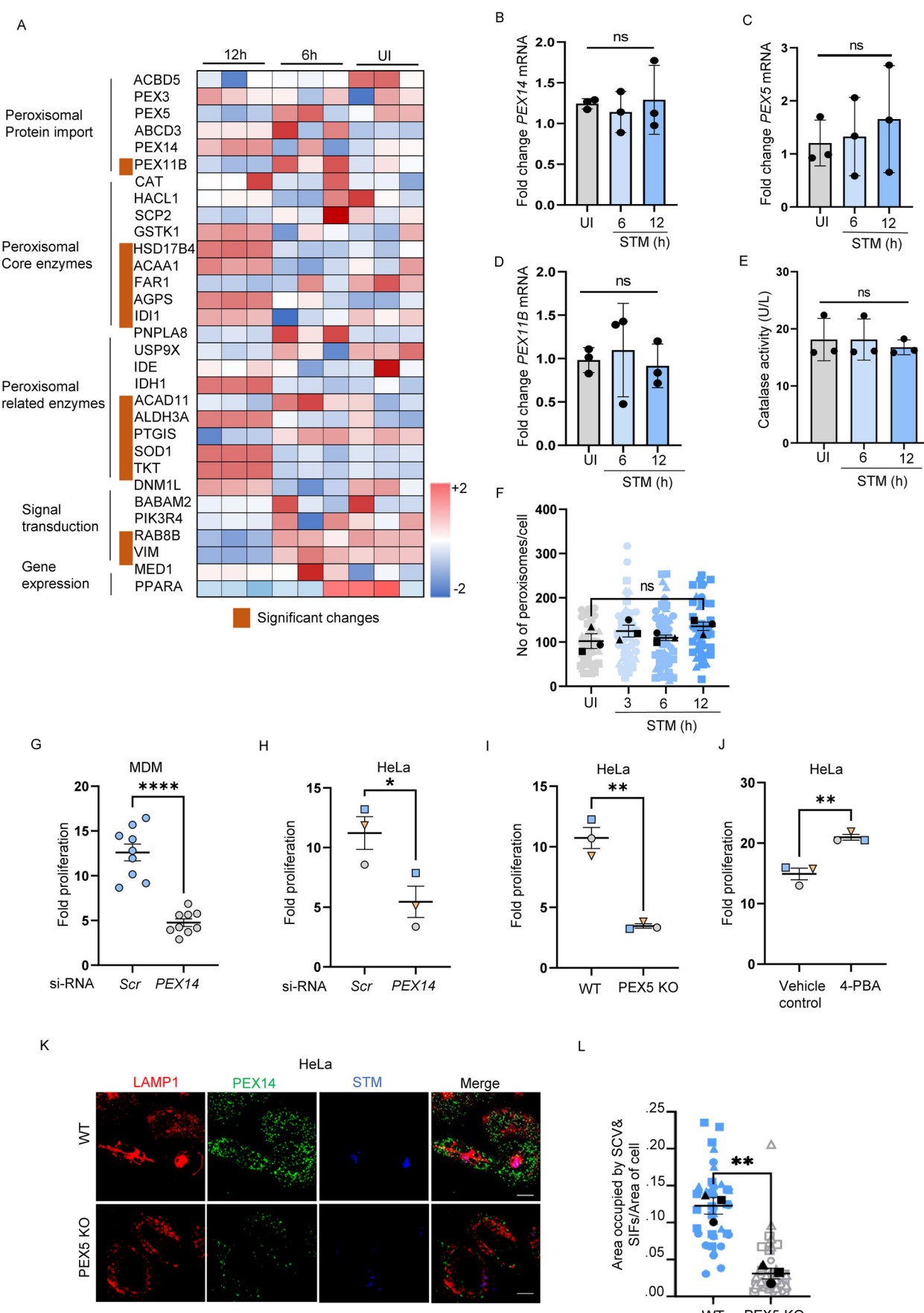

**Figure 2.   Peroxisomes are required for efficient intracellular replication of STM.**

(A) Mass spectrometry-based quantification of peroxisome-related proteins after 6 and 12 h of STM infection in HeLa cells. Among the 3412 proteins measured, 31 proteins associated with peroxisome and their changes in comparison with uninfected cells are shown in the heat map. (B–D) Fold change in mRNA levels of indicated peroxisomal genes (*PEX14* (B), *PEX5* (C), and *PEX11B* (D)) in HeLa cells post 6 and 12 h of STM infection compared with uninfected (UI). Statistics were performed on the mean of three independent experiments (black dots). (E) Catalase activity in HeLa cells was measured at indicated time points post-STM infection. Data represent the mean of catalase activity from three independent experiments. (F) Graph indicating the number of peroxisomes per HeLa cell infected with STM (MOI = 50) after 3, 6, and 12 h of infection. Data for three independent experiments containing more than 200 cells are shown. (G) Change in fold proliferation of STM (MOI = 10) in MDM cells after silencing *PEX14*. Each dot represents one donor, and the data for 9 donors ($n = 9$) is shown here. (H) Change in fold proliferation of STM (MOI = 10) in HeLa cells after silencing *PEX14*. Three independent experiments were performed. Each dot represents the mean of an independent experiment. (I) Change in fold proliferation of STM (MOI = 10) in WT and *PEX5* KO HeLa cells. Three independent experiments were performed. Each dot represents the mean of an independent experiment. (J) Change in fold proliferation of STM (MOI = 10) after treatment with 4-PBA (2 mM) in HeLa cells. Three independent experiments were performed. Each dot represents the mean of an independent experiment. (K) Representative confocal micrographs of HeLa cells infected with GFP-STM (Blue) at MOI of 50 for 12 h. Cells were fixed and stained using an anti-LAMP1 antibody (red) and an anti-PEX14 antibody (green). Scale bars: 10 μm. (L) Graph for the area occupied by SCVs and Sif/Area of the cell. Data for three independent experiments containing more than 80 cells are shown. Data information: (B–F) Statistical significance was determined using one-way ANOVA; 'ns' indicates non-significant difference. (G–L) Statistical significance was determined using student's *t*-test, (G) (****$p < 0.0001$), (H) (*$p = 0.0387$), (I) (**$p = 0.0012$), (J) (**$p = 0.0049$), and (L) (**$p = 0.0024$). Source data are available online for this figure.

## In silico analysis identified a *Salmonella* effector protein, SseI to contain putative peroxisome-targeting sequence

SPI2 TTSS helps to maintain SCV integrity and intracellular proliferation of *Salmonella* (Jennings et al, 2017). To identify the bacterial factors responsible for SCV-peroxisome colocalization, we next infected HeLa cells with the Δ*ssaV* strain (where functional SPI2 needle is absent and secretion of other *Salmonella* SPI2 effector proteins is prevented). Interestingly, Δ*ssaV* infected cells displayed significantly decreased colocalization with peroxisomes (Fig. 3A) compared to the WT *Salmonella*. We therefore argued that the SPI2 bacterial effector protein(s) might be playing a crucial role in the peroxisome-SCV dynamic contacts seen during *Salmonella* infection.

To further pin down the bacterial protein, we queried the *Salmonella* TTSS effector proteins to identify those that contain a putative host peroxisome-targeting sequence 1 (PTS1). We used an in silico software, "PTS1 predictor," that has a built-in algorithm to analyze the 12 amino acids of the C terminal for PTS1 motifs (Neuberger et al, 2003a; Neuberger et al, 2003b). The software classifies the query sequences either as "Not targeted," "Twilight Zone," or "Targeted" based on the score and *p*-value generated by the algorithm. Strikingly, the analysis of *Salmonella* SPI2 effector proteins predicted one effector, SseI, to contain a variant of host PTS1 tripeptide, 'GKM' (Fig. EV2A). The presence of this PTS1-like motif in SseI was conserved across multiple *Salmonella* serovars infecting different hosts (Fig. 3B). Microscopic analysis of the SseI deletion strain exhibited significantly reduced interaction with the peroxisomes (Fig. 3C,D). Tagging the identified PTS1-like motif with EGFP showed colocalization with peroxisomes. It is to be noted that unlike EGFP-SKL, the identified 'GKM' motif shows cytoplasmic signal along with peroxisomal localization (Fig. 3E,F). Similar dual targeting by other PTS1 variants-containing proteins is reported in the literature. For instance, human Epoxide Hydrolase, which contains 'SKM' motif also shows dual cytosolic and peroxisomal localization (Beetham et al, 1993; Arand et al, 1991). During STM infection, we observed that proliferation of the Δ*ssaV* strain and the Δ*sseI* strain also showed a significant growth reduction in the HeLa cells and primary macrophages. In addition, complementation of SseI by plasmid expression restored the growth defect of the Δ*sseI* strain (Fig. 3G— HeLa cells; Fig. EV2B—MDM).

To experimentally verify if SseI gets targeted to peroxisomes, we generated N-terminal HA-tagged clones containing full-length SseI.

We expressed the construct in HeLa cells stably expressing PEROXO-tag (3XMyc-EGFP-PEX26) for easy peroxisome purification (Ray et al, 2020). The validity of the assay system and purity of the peroxisomal isolation are shown in Fig. EV2C,D. We observed the presence of SseI after isolating peroxisomes, and the removal of either the predicted PTS1-like tripeptide or a point mutation in PTS1 (K321A) resulted in negligible SseI accumulation on the isolated peroxisomes (Fig. 3H,I). Transfection of PEROXO-tag neither affected the proliferation of HeLa cells nor the invasion capability of STM (Fig. EV2E,F). Next, we wanted to probe if SseI secreted from the WT STM also localizes to peroxisome during infection. As commercial antibodies do not exist against SseI, purified SseI protein was injected into mice, and polyclonal antisera was used to detect SseI. Data shown in Fig. 3J clearly demonstrated the presence of SseI after isolating peroxisomes during WT *Salmonella* infection in the HeLa cells.

However, these results do not comment whether SseI is present on the membrane or matrix of peroxisomes. it is known that PTS1-motif-containing proteins bind to a peroxisomal receptor protein, PEX5, that transports them onto the organelle matrix (Wang and Subramani, 2017). Therefore, we probed to find if SseI enters the peroxisome matrix using protease protection assay. Organelle fractions were treated with protease K with or without Triton X-100. Adding Triton X-100 will permeabilize the peroxisome, giving access to proteinase K to degrade the matrix proteins (Connolly et al, 1989). As shown in Fig. 3K, negligible amounts of SseI were protected in the absence of the detergent. This data clearly showed that although SseI can enter the matrix, it is predominantly present on the membrane of peroxisomes. Catalase was used as the marker for matrix protein and it is completely protected in the absence of Triton X-100. PEX14 was used as a marker of peroxisome membrane protein that gets degraded even in the absence of Triton X-100. Therefore, our results show that in spite of SseI being a PEX5 interactor, it does not efficiently transport the bound SseI into the peroxisomal matrix. This raises an interesting question on the site of SseI action. To understand the significance of SseI localization on the membrane of peroxisomes, we replaced the 'GKM' motif with 'SKL' on SseI. SKL is the prototype PTS1 motif that is efficiently imported to peroxisomes. As expected, SseI tagged with SKL efficiently gets targeted to peroxisomal matrix as validated by protease protection assay (Fig. EV2G). In such a scenario, we observe that SseI::SKL cannot rescue the growth defect of the Δ*sseI* strain (Fig. 3L). This indicates that GKM motif-mediated localization of SseI on peroxisome-membrane is essential for its mode of action. Further,

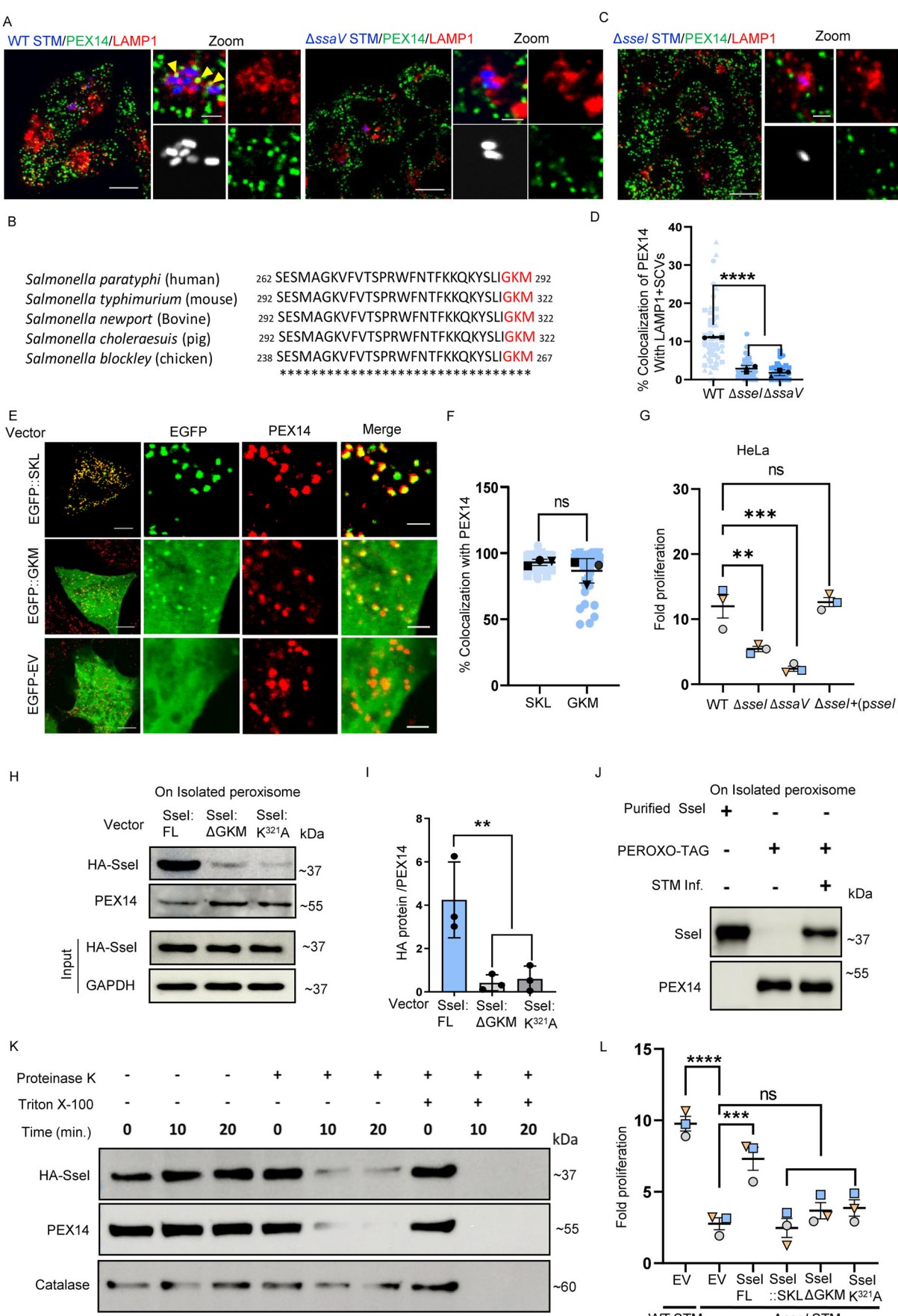

**Figure 3. *Salmonella* effector protein, SseI, contains a putative PTS1 motif and interacts with peroxisomes.**

(A) Confocal microscopy of HeLa cells infected with the wild-type (WT) or Δ*ssaV Salmonella* Typhimurium (STM) (MOI = 50) (blue) for 6 h. Cells were immunostained for LAMP1 (red) and PEX14 (green). Structures co-stained for LAMP1 and STM were defined as *Salmonella*-containing vacuoles (SCVs). Yellow arrowheads indicate PEX14 colocalization with SCVs. Scale bars: 10 μm (main panel), 2 μm (inset). (B) Sequence alignment of conserved PTS1-like variant (GKM, red) in the C-terminus of multiple *Salmonella* species infecting different hosts. (C) Confocal microscopy of HeLa cells infected with Δ*sseI* GFP-STM (blue) (MOI = 50) for 6 h. Staining and image acquisition were performed as in (A). (D) Quantification of percentage PEX14 colocalization with SCVs. Data represent the mean ± SEM of three independent experiments. (E) Confocal microscopy of HeLa cells transfected with pEGFP::SKL, pEGFP::GKM, or empty vector (EV) pEGFP-EV plasmids. Cells were immunostained for PEX14 (red). Zoom panel indicate colocalization of GFP and PEX14 puncta. Scale bars: 10 μm (main panel), 2 μm (inset). (F) Quantification of percentage PEX14 colocalization EGFP::SKL and EGFP::GKM. Data are represented from three independent experiments. (G) Fold proliferation of WT STM, Δ*ssaV* STM, Δ*sseI* STM, and Δ*sseI* STM (MOI = 10) complemented with pQE60-SseI in HeLa cells. Data represent the mean ± SEM of three independent experiments. (H) HeLa cells were transfected with N-terminally HA-tagged SseI constructs (full-length, ΔGKM, K$^{321}$A) and a PEROXO-Tag plasmid. Peroxisomes were subsequently pulled down, and eluted samples were subjected to immunoblot analysis. SseI protein levels were determined using anti-HA antibodies, while protein loading was controlled by immunoblotting for PEX14. (I) Quantification of HA-SseI after peroxisome pulldown. Band intensity was normalized to PEX14. Data are represented from three independent experiments. (J) HeLa cells transfected with PEROXO-tag were infected with STM (MOI = 10) for 6 h. Peroxisomes were isolated from both infected and uninfected HeLa cells. Purified SseI protein was used as a positive control. Immunoblot analysis was performed on these samples using anti-SseI antibodies. (K) Immunoblot analysis of HA-SseI protein after protease protection assay on isolated peroxisomes. Catalase and PEX14 were used as controls for the matrix and membrane. (L) Fold proliferation of WT STM, Δ*sseI* STM, and Δ*sseI* STM complemented (MOI = 10) with pQE60-SseI clones (full length (FL), SKL, ΔGKM, K$^{321}$A) in HeLa cells. Data represent the mean ± SEM of three independent experiments. Data analysis: Data were analyzed using one-way ANOVA (Sidak's multiple comparisons test), (D) (****$p < 0.0001$), (G) (**$p = 0.0053$, ***$p = 0.0004$), (I) (**$p = 0.0083$), and (L) (****$p < 0.0001$, ***$p = 0.0009$). (F) Data were analyzed using student's *t*-test in with 'ns' denotes no significant difference. Source data are available online for this figure.

rescue in bacterial replication of Δ*sseI* strain was observed in the case of cells expressing the full-length SseI compared to the cells expressing either SseI-ΔGKM or SseI-K321A. Since SseI bound to PEX5 does not get delivered to peroxisomes, it could impact the dissociation of PEX5 from cargo and the availability for its recycling for binding of new cargo can be affected. In order to test this possibility, we overexpressed SseI and studied the import of catalase. As seen in Fig. EV2H, SseI overexpression does not affect the catalase localization in isolated peroxisomes. This indicates that SseI does not affect the import efficiency of other matrix proteins to peroxisomes.

SseI was previously found to regulate cell migration (McLaughlin et al, 2009; McLaughlin et al, 2014; Carden et al, 2017; Brink et al, 2018). Our next results were also in a similar line, showing that SseI overexpression led to higher HeLa cell migration. However, this effect on cell migration was not affected by mutation on the PTS1-like motif(K321A) (Fig. EV2I,J).

## SseI binds to host GTPase ARF1 and regulates its activity

We proceeded with the speculation that SseI might sequester/activate the functionality of host proteins on the peroxisome membrane. From the available literature (Sontag et al, 2016; Walch et al, 2021) certain human proteins are predicted to interact with SseI (Fig. EV3A). We looked for the presence of these proteins (IQ motif containing GTPase activating protein (IQGAP), Thyroid hormone receptor interacting protein 6 (TRIP6), and ARF1) on the isolated peroxisomes, and one among them, ARF1, was found on the peroxisomes (Fig. 4A). ARF1 is a small Ras GTPase and is involved in protein trafficking. It is kept either active (GTP-bound) or inactive (GDP-bound) by Guanine exchange factors (GEFs) or GTPase activating proteins (GAPs), respectively. ARF1 was previously reported to interact with PEX35 in the peroxisome and was found to be involved in vesicle migration (Yofen et al, 2017). Recent studies have also indicated the role of ARF1 in the synthesis of PIP2 (Skippen et al, 2002), which is essential for organelle docking and movement (Skippen et al, 2002; Heuvingh et al, 2007).

Co-immunoprecipitation studies showed that while SseI interacts with endogenous ARF1 and PEX5, SseI-ΔGKM showed binding to ARF1 but not with PEX5 (Fig. 4B). To confirm this result, another

approach was used. Purified SseI protein (His-tagged) showed in vitro binding with HA-tagged ARF1 from overexpressed HeLa cell lysates (Fig. 4C, protein purity—Fig. EV3B). Further, infection with WT STM activated ARF1 more when compared to the Δ*sseI* strain (Fig. 4D,E), while the total ARF1 levels remained unchanged. Thus, these results suggest that SseI interacts with ARF1 and possibly mimics the role of a host ARF1- GEF during infection. We then studied the level of activated peroxisomal ARF1. We observed an increase in the activated ARF1 level on peroxisomes after infection with wild-type STM. However, this increase was not observed following infection with Δ*sseI* STM. Similarly, we performed complementation with constructs expressing either SseI-ΔGKM or SseI::SKL in Δ*sseI* STM. In both these cases, we did not observe an increase in the levels of activated ARF1 on peroxisomes. This result indicates that peroxisomal membrane localization of SseI by GKM motif is essential for increasing the levels of peroxisomal active ARF1 (Fig. 4F).

Following this, we investigated the importance of ARF1 activation in STM proliferation by silencing *ARF-1*. Interestingly, we observed a reduction in bacterial proliferation during WT STM infection after silencing *ARF1*. On the other hand, in the cells where constitutively active ARF1 (Q71L) was expressed (Dascher and Balch, 1994), the growth defect of the Δ*sseI* strain reverted similar to that of WT strain (Fig. 4G). We could effectively silence *ARF*1 (Fig. EV3C), and its knockdown had a marginal effect on the cell viability (Fig. EV3D).

## Activated ARF1 induces PIP2 synthesis to strengthen the peroxisome contacts with SCVs

ARF1 was previously shown to localize with yeast peroxisomes as well as with purified rat liver peroxisomes (Enkler et al, 2023). It regulates PI(4,5)P2 (or PIP2) levels at the membrane in HL60 cells by activating phosphatidylinsolitol5-phosphate-4 kinases (PI5P4Ks) (Skippen et al, 2002; Just and Peranen, 2016). Also, PIP2 has been previously shown to help in organelle contact and organelle movement (Yin and Janmey, 2003; Martin, 2015; Xiao et al, 2019). As we observed increased recruitment of peroxisomes to SCVs after infection and SseI binds to ARF1, we looked closely at the organellar PIP2 level.

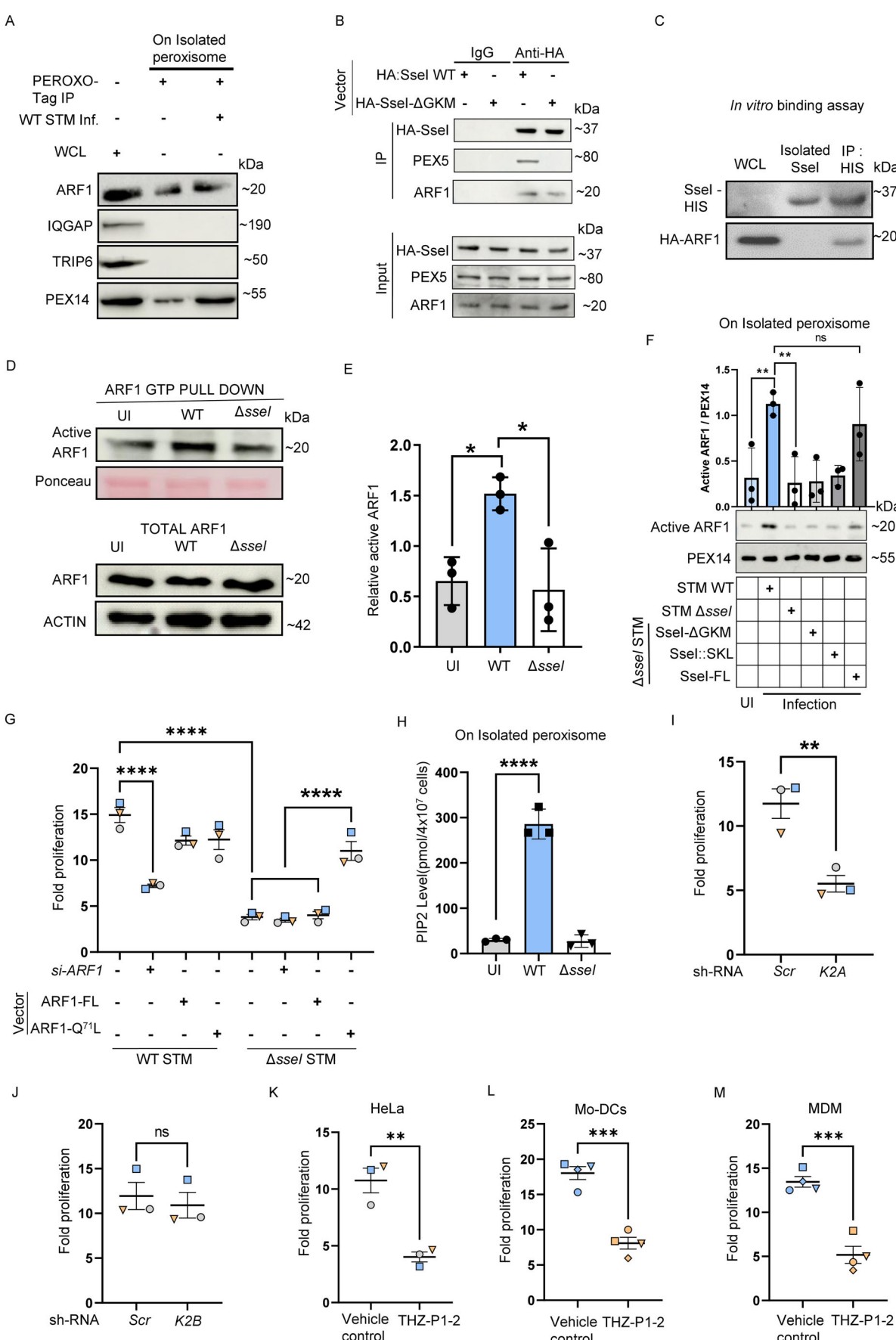

◄

**Figure 4.    SseI binds to host GTPase ARF1 and regulates its activity to induce PIP2 synthesis on peroxisome membrane.**

(A) Immunoblot analysis of SseI-interacting proteins (IQGAP, TRIP6, and ARF1) on isolated peroxisomes from STM-infected (MOI = 10) and uninfected HeLa cells, with whole cell lysate (WCL). The representative blot was probed with anti-IQGAP, anti-TRIP6, and anti-ARF1 antibodies. (B) Immunoblot analysis of proteins after SseI pulldown from HeLa cells expressing HA-tagged SseI WT and SseI ΔGKM. Cell lysates (input) were used for SseI pulldown with anti-HA antibody. IgG was used as a control. The representative blot was probed with anti-HA, anti-ARF1, and anti-PEX5 antibodies. (C) Immunoblot analysis of in vitro binding between purified His-tagged SseI and HA-tagged ARF1 in HeLa whole cell lysates (WCL). The representative blot was probed with anti-His and anti-HA antibodies. (D) Immunoblot analysis of Active ARF1 was pulled down from WT, ΔsseI STM-infected (MOI = 10), and uninfected HeLa cells. Immunoblot analysis was performed to detect active ARF1. Ponceau staining was used as a loading control for the pulldown samples. Total ARF1 levels in whole cell lysates were assessed by immunoblotting, with actin as a loading control. (E) Densitometry analysis of ARF1 activation in WT and ΔsseI STM (MOI = 10) infected HeLa cells relative to uninfected controls. Data represent the mean ± SEM of three independent experiments. (F) Immunoblot and densitometry analysis of Active ARF1 protein levels after ARF1-GTP pulldown from isolated peroxisomes, cells infected with WT or ΔsseI STM, or ΔsseI STM complemented with pQE60-SseI clones (ΔGKM, SKL, full length (FL) (MOI = 10). Densitometry analysis represents three independent experiments. (G) Fold proliferation of WT or ΔsseI STM (MOI = 10) in HeLa cells transfected with full-length or constitutively active (Q71L) ARF1 or silenced ARF1. Data represent the mean ± SEM of three independent experiments. (H) ELISA analysis of PI (4,5) P2 (PIP2) levels on isolated peroxisomes from PEROXO-tag expressing HeLa cells infected with WT or ΔsseI STM. Data from three independent experiments are shown. (I, J) Fold proliferation of STM in HeLa cells after silencing PIP4K2A (I) or PIP4K2B (J). Data represent the mean ± SEM of three independent experiments. (K) Fold proliferation of WT STM (MOI = 10) in HeLa cells treated with PIP4K inhibitor THZ-P1-2 (1 μM). Data represent the mean ± SEM of three independent experiments. (L, M) Fold proliferation of WT STM (MOI = 10) in human monocyte-derived dendritic cells (Mo-DCs) (L) and human monocyte-derived macrophages (MDMs) (M) treated with PIP4K inhibitor THZ-P1-2 (1 μM). Data are from four healthy human donors. Data information: Data were analyzed using one-way ANOVA (Sidak's multiple comparisons test), (E) (*p = 0.0211, *p = 0.0138), (F) (**p = 0.0092, **p = 0.0058), (G) (****p < 0.0001), and (H) (****p < 0.0001), 'ns' denotes no significant difference. (I–M) Data were analyzed using student's t-test, (I) (**p = 0.0091), (J, K) (**p = 0.0046), (L) (***p = 0.0002), and (M) ((***p = 0.0003). 'ns' denotes no significant difference. Source data are available online for this figure.

We observed that WT *Salmonella* infection enhanced PIP2 levels on the isolated peroxisomes. Notably, infection with the ΔsseI strain failed to induce PIP2 accumulation on the peroxisomes, suggesting the crucial role of SseI in this connection (Fig. 4H). To rule out if purified SseI can directly bind to any of the phospholipids, including PIP2, we did a protein-lipid overlay assay by using a PIP strip assay (Chu et al, 2015) and found that it binds to none of the lipids tested (Fig. EV3E).

We next wanted to evaluate the regulators of PIP2 synthesis downstream of active ARF1, specifically on the peroxisome membrane. Although phosphatidylinositol 4-phosphate-5 kinases (PI4P5K) are responsible for synthesizing most of the cellular PIP2 (Rameh et al, 1997), the non-canonical PI5P4K (Ravi et al, 2021) regulates membrane PIP2 generation (Hu et al, 2018). Mammals encode three isoforms of PI5P4Ks—α, β, γ encoded by three different genes, namely, PI5P4K2A, PI5P4K2B, and PI5P4K2C. The role of PI5P4Ks during bacterial infection remains understudied. Western blot analysis revealed that WT STM infection specifically enhances PI5P4K2A but not the PIP4K2B isoform. In contrast, the ΔsseI strain failed to upregulate PI5P4K2A protein during infection in the HeLa cells (Fig. EV3F,G). The silencing *PI5P4K2A* but not *PI5P4K2B* showed reduced STM proliferation (Figs. 4I,J and EV3H–K). These results show that only PI5P42A but not PI5P4K2B impact STM growth.

Importantly, blocking the PI5P4 kinase activity with a specific inhibitor THZ-P1-2 (Sivakumaren et al, 2020) leads to a significant growth defect of the WT strain in HeLa cells, human blood-derived dendritic cells (Mo-DC), and human blood-derived primary macrophages (MDM) (Fig. 4K–M). THZ-P1-2 did not change cell viability at the concentration tested (Fig. EV3L). These data revealed that SseI activates host ARF1 on the peroxisomes, leading to an increase in the levels of PIP2 on peroxisomes mediated by the PI5P4K2A kinase.

## LDL-derived cholesterol routed via lysosome-peroxisome is essential for *Salmonella* growth

The above results led us to interrogate why *Salmonella* displays such an elaborate mechanism to alter peroxisomal PIP2 level to enhance peroxisome movement and/or connection with the SCV. It is shown that PIP2 generated at the peroxisome is essential for interaction with

lysosomes (Chu et al, 2015) and mediates lipid trafficking to mitochondria (Ravi et al, 2021) and ER (Xiao et al, 2019).

Hence, we asked if STM infection similarly induced cholesterol transfer from lysosome to SCV through peroxisome. Multiple approaches were used to test this hypothesis. There were enhanced lysosome-peroxisome contacts during STM infection (Fig. 5A). Further, we validated the cholesterol accumulation on the isolated peroxisomes after STM infection. As shown in Fig. 5B, peroxisomes showed higher cholesterol content when infected with the WT bacteria compared to ΔsseI STM 6 h post-infection. Similarly, complementing SseI by plasmid expression in the ΔsseI STM rescued the cholesterol accumulation on peroxisomes. Interestingly, complementing with SseI- ΔGKM or K321A mutants failed to accumulate cholesterol similar to the ΔsseI STM strain. This result shows that cholesterol accumulation on peroxisomes induced by SseI depends on GKM motif that mediates the interaction of peroxisomes with SCVs. To quantify the cholesterol levels on SCVs, we stained cellular cholesterol using a cholesterol-binding fluorescent dye, filipin (Maxfield and Wustner, 2012). Consistent with the previous findings, WT STM infection led to cholesterol accumulation on the SCVs. In contrast, the ΔsseI strain failed to sequester cholesterol on the SCV. Finally, we used the *PEX5* KO cells and observed that the WT STM infection failed to accumulate cholesterol in the SCV (Fig. 5C,D). Taken together, these results demonstrate that cholesterol accumulates in SCVs during maturation, and both host peroxisomes and bacterial SseI effector work hand in hand for this transport.

We next silenced the lysosomal cholesterol transporter, Niemann-Pick disease, type C1 (*NPC1*) and peroxisome membrane protein, ATP binding cassette subfamily D member 1 (*ABCD1*), whose function was previously documented in lysosomal cholesterol transfer to other host organelles (Chu et al, 2015). Knockdown of these proteins is known to perturb the lysosome-peroxisome contacts, thereby reducing cholesterol transport to peroxisomes. As shown in Fig. 5E,F, silencing of either *NPC1* or *ABCD1* showed reduced intracellular *Salmonella* replication. Knockdown efficiency is presented in Fig. EV4A,B. Similarly, while adding exogenous LDL to the media enhanced bacterial replication, adding an NPC1 inhibitor, U18666A, either alone or with LDL, prevented bacterial replication (Fig. 5G). Similarly, silencing LDL receptor (*LDLR*)

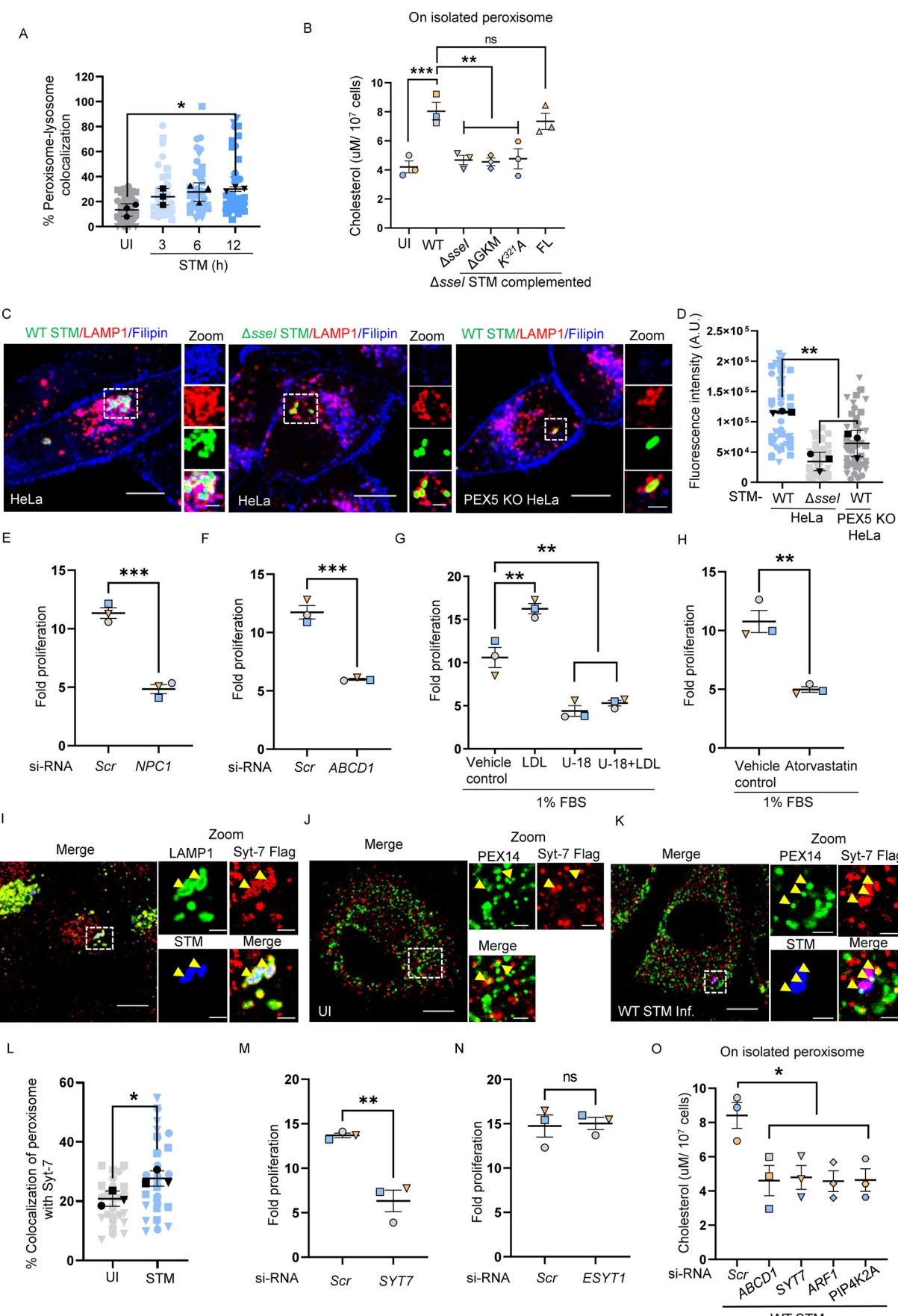

**Figure 5. Syt7 on SCV tethers PIP2 on peroxisome to facilitate cholesterol transfer.**

(A) Graph representing the percentage of peroxisome-lysosome colocalization in HeLa cells after infection with WT STM with MOI = 10. Interaction of PEX14 and LAMP1 was monitored at 3, 6, and 12-h post-infection. Data for three independent experiments containing more than 200 cells are shown. (B) Cholesterol levels on isolated peroxisomes from HeLa cells infected with indicated bacterial strains (MOI = 10) for 6 h. Data represent the mean ± SEM of three independent experiments. (C) Representative confocal micrographs of HeLa and PEX5 knockout (KO) HeLa cells infected with either GFP-tagged WT STM or GFP-tagged ΔsseI STM (MOI = 50) for 6 h. Cells were immunostained with anti-LAMP1 antibody to label SCVs (red); cholesterol was stained using filipin (blue) and STM (green). Scale bars: 10 μm (main panel), 2 μm (inset). (D) Graph representing the change in fluorescence intensity of filipin, indicating cholesterol on SCVs. Data represent the mean ± SEM from three independent experiments with more than 180 cells analyzed. (E) Graph representing the changes in fold proliferation of STM (MOI = 10) after silencing NPC1 in HeLa cells. Data represent the mean ± SEM of three independent experiments. (F) Graph representing the changes in fold proliferation of STM (MOI = 10) after silencing ABCD1 in HeLa cells. Data represent the mean ± SEM of three independent experiments. (G) Change in fold proliferation of WT STM with LDL/U18666A in the media containing 1% FBS in HeLa cells. Data represent the mean ± SEM of three independent experiments. (H) Graph representing the changes in fold proliferation of STM (MOI = 10) after treatment with Atorvastatin (1 μM), the media containing 1% FBS in HeLa cells. Data represent the mean ± SEM of three independent experiments. (I) Representative confocal micrographs of HeLa cells infected with either GFP-tagged WT STM with Syt-7 Flag overexpressing plasmid. In Zoom, the yellow arrowhead represents the colocalization between Syt-7 (red), LAMP1 (green), and STM (blue). Scale bars: 10 μm (main panel), 2 μm (inset). (J) Representative confocal micrographs of uninfected HeLa cells with Syt-7 flag overexpressing plasmid. In Zoom, the yellow arrowheads represent the colocalization between Syt-7 (red) and PEX14 (green). Scale bars: 10 μm (main panel), 2 μm (inset). (K) Representative confocal micrographs of Syt-7 flag overexpressing HeLa cells infected with GFP-tagged WT STM (blue). In Zoom, the yellow arrowheads represent the colocalization between Syt-7 (red), PEX14 (green), and STM (blue). Scale bars: 10 μm (main panel), 2 μm (inset). (L) Graph representing the percentage colocalization of peroxisome and Syt-7 in HeLa cells after infection with WT STM and uninfected cells. Data of three independent experiments containing more than 80 cells are shown. (M, N) Change in fold proliferation of WT STM (MOI = 10) after silencing Syt-7 (M) or E-Syt1 (N) in HeLa cells. Data represent the mean ± SEM of three independent experiments. (O) Cholesterol levels on isolated peroxisomes from HeLa cells silenced for NPC1, SYT7, ARF1, and PIP4K2A and infected with WT STM (MOI = 10) for 6 h. Data represent the mean ± SEM of three independent experiments. Data analysis: Data were analyzed using one-way ANOVA (Sidak's multiple comparisons test), (A) (*$p$ = 0.0452), (B) (***$p$ = 0.0007, **$p$ = 0.0023) (D) (**$p$ = 0.0018), (G) (**$p$ = 0.0019), and (O) (*$p$ = 0.0144). Data were analyzed using student's $t$-test, (E) (***$p$ = 0.0004), (F) (***$p$ = 0.0006), (H) (**$p$ = 0.0039), (L) (*$p$ = 0.0313), (M) (**$p$ = 0.0041). 'ns' denotes non-significant difference. Source data are available online for this figure.

reduced STM proliferation (Fig. EV4C,D). These experiments suggest that exogenous LDL is essential for STM growth.

To better understand the contribution of de novo synthesized cholesterol during STM infection, we used a chemical inhibitor of de novo synthesis, atorvastatin. The analysis was performed in either 1% FBS or lipoprotein-free media to remove exogenous cholesterol (Xiao et al, 2023). In both these scenarios, where LDL-mediated cholesterol is restricted, we observed a reduction in the fold proliferation of STM after atorvastatin treatment (Figs. 5H and EV4E). This is probably due to the need for sterol intermediates derived from the de novo pathway apart from cholesterol, as previously reported before (Catron et al, 2004). To verify this possibility, we silenced the de novo synthesis pathway in its initial or final stages to differentiate sterol intermediates from cholesterol. While silencing 3-hydroxy-3-methylglutaryl coenzyme A, (HMGCoA) leads to complete perturbation of de novo pathway, silencing DHCR24, only affects cholesterol production from desmosterol. We see that silencing HMG CoA affects STM proliferation but silencing DHCR24 does not have any impact on STM proliferation (Fig. EV4F–I). In addition, intracellular fold proliferation of STM in HeLa cells maintained in 1% FBS showed significant growth reduction compared to cells maintained in 10% FBS (Fig. EV4J). Taken together, these results suggest that Salmonella acquires cholesterol from LDL processed in lysosomes and not de novo pathway. None of the siRNA or inhibitors tested altered cell viability (Appendix Fig. S2A–F).

## Syt7 tethers SCV to peroxisome to facilitate cholesterol transfer

Transport of molecules between organelles requires the participation of tethering proteins to bring the organelles into proximity, thereby facilitating transfer. Our next objective was to identify the binding partner on SCVs that tethers peroxisome in contact. Syt7 and E-Syt1 present on lysosomes and ER, respectively, primarily interact with peroxisomal PIP2 to transport cholesterol (Chu et al, 2015; Xiao et al, 2019; van den Boomen et al, 2020). Syt7-FLAG showed localization on SCVs (Fig. 5I) and interacted with peroxisomes. This interaction of Syt7

with peroxisomes increases during infection (Fig. 5J–L). In addition, bacterial replication was restricted when cells were silenced for SYT7 (Fig. 5M, silencing efficiency—Fig. EV4K). In contrast, silencing E-SYT1 did not affect bacterial proliferation (Figs. 5N and EV4L). Similarly, endogenous antibody staining of E-Syt1 did not show localization with SCVs (Fig. EV4M). However, Syt7, both by endogenous antibody staining as well as overexpression (Syt7-FLAG) clearly shows colocation with peroxisomes. Further, to validate that peroxisomes do not contain Syt7, peroxisomes were isolated after infection. As seen in Fig. EV4N, pure fraction of isolated peroxisomes does not contain Syt7. None of the siRNA tested altered cell viability (Appendix Fig. S2G–I).

Next, the peroxisomal cholesterol accumulation was measured in HeLa cells after NPC1, SYT7, ARF-1, and PI5P4K2A knockdown. A significant reduction of cholesterol in the purified peroxisomes was observed at all the conditions tested (Fig. 5O). These data demonstrated that the LDL-derived cholesterol in the lysosomes gets transported to SCVs by using peroxisome as a bridge. Altogether, these experiments suggest that transport of cholesterol from the peroxisome to SCVs is mediated by PIP2 on peroxisome membrane and Syt7 on membrane of the SCVs.

## SseI mutant is attenuated in the animal model of infection

Having established the crucial role of SseI during in vitro growth of STM, we set out to test whether this protein is essential for STM pathogenicity in vivo in the mouse model of infection. Based on the previously published reports on role of SPI-2 effector proteins (Coombes et al, 2005; DasSarma et al, 2014; Jiang et al, 2021; Chatterjee et al, 2023), we administered 10^7 CFU of STM by oral gavage for 7 days. The dissemination of bacteria to organs such as the liver, spleen, mesenteric lymph node (MLN), and intestine was studied seven days post-infection. Data shown in Fig. 6 clearly indicated that the mutant strain grew significantly less when compared to its WT counterpart. This hypovirulence in the ΔsseI

strain was observed in all the mice organs tested, like the intestine, MLN, spleen, and liver (Fig. 6A–D) like known attenuated mutant (Δ*ssaV* strain). There was also rescue of the reduced growth seen with Δ*sseI* strain after complementing with plasmid expression of SseI, indicating the in vivo essentiality of SseI.

To further evaluate the loss of virulence of the Δ*sseI* strain, we studied the ability of the knockout strain to compete with the WT strain. As seen in Fig. 6E,F, Δ*sseI* strain showed a competitive index defect in liver and spleen analyzed 7 days post-infection. Similarly, complementing SseI during Δ*sseI* STM infection displayed similar virulence as wild-type strain. Interestingly, we also observe that this competence is hindered when complemented with SseI-ΔGKM or SseI-K321A. Therefore, we propose that loss of SseI binding to peroxisomes results in reduced virulence. This result also indicates the in vivo relevance of peroxisomes and cholesterol trafficking during *Salmonella* infection.

Collectively, our findings demonstrate that STM infection perturbs mammalian peroxisomal dynamics, and transports LDL-derived lysosomal cholesterol to SCVs using peroxisome as a bridge. Mechanistically, we show that a *Salmonella* virulence protein, SseI, targets host peroxisome and modulates host organellar cholesterol homeostasis to stabilize SCV and SIF integrity.

# Discussion

Peroxisomes are morphologically and functionally remodeled during viral infection (Cook et al, 2019; Ferreira et al, 2022). However, the role of peroxisomes during bacterial infection remains understudied. Pellegrino et al, showed that peroxisomal ROS activity and an increase in peroxisomal biogenesis are critical for controlling cytosolic *Mycobacterium* infection (Behera et al, 2022). Similarly, another study demonstrated the essentiality of peroxisomes for efficient phagocytosis of *Escherichia coli* in the murine macrophages (Di Cara et al, 2017; Nath et al, 2022). Interestingly, during *Chlamydia* infection, although peroxisomes get recruited near the bacteria, they are not essential for the bacterial replication (Boncompain et al, 2014). These studies indicate that the role of peroxisomes during bacterial infection has evolved differently across the bacterial species.

In this study, we demonstrate: (i) *Salmonella* infection induces transient interactions between SCVs and peroxisomes; (ii) This interaction aids in the accumulation of cholesterol around SCVs necessary for SCV and SIF maintenance and bacterial growth, (iii) A bacterial protein in *Salmonella*, SseI contains a PTS1-like motif that mediates the interaction of SCVs with peroxisomes; (iv) SseI activates a host GTPase, ARF1, to induce PIP2 levels on peroxisomes, which facilitates the interaction of peroxisomes with SCVs using Syt7 as the tethering protein on SCVs (Schematic summary).

Peroxisomes are remarkably plastic concerning their size, number, and intracellular localization (Platta and Erdmann, 2007). Most eukaryotic peroxisomal matrix proteins contain a tripeptide PTS1 motif at the C-terminus. Although the consensus motif for PTS1 is (S/A/C)-(K/R/H)-(L/M), many studies suggest degenerated consensus in the motif (Ast et al, 2013). Interestingly, parasites like trypanosomes and multiple fungal species have PTS1-containing proteins, which are critical for their virulence (Ding et al, 2000; Freitag et al, 2012). In addition, rotavirus structural protein VP4 contains a PTS1 sequence and was shown to migrate

to host peroxisome (Mohan et al, 2002). Current study shows that *Salmonella* SPI2 effector protein SseI (also called SrfH) has a conserved PTS1-like motif (GKM) at the C-terminus and is targeted to the host peroxisome membrane. In addition, C-terminal tagged SseI was previously shown to be targeted to the host cell plasma membrane after palmitoylation (Miao et al, 2003; Hicks et al, 2011). Such dual targeting of peroxisomal proteins in eukaryotes is also reported previously. For instance, catalase A of *S. cerevisiae* contains PTS1-like motif and a mitochondrial targeting signal motif (Petrova et al, 2004). The distribution of the protein between peroxisomes and mitochondria depends on the growth environment. In such dual-targeting proteins, localization studies by fluorescent tagging are difficult. Buch et al used photobleaching in live cells to differentiate peroxisomal localization of bile acid-CoA:amino acid N-acyltransferase from cytoplasmic localization (Buch et al, 2009).

Other studies have shown that SseI inhibits dendritic cell migration and chemotaxis (McLaughlin et al, 2009; McLaughlin et al, 2014; Carden et al, 2017; Brink et al, 2018). On the other hand, SseI promoted murine macrophage motility (Worley et al, 2006). Further, the C-terminal of SseI also contains a catalytic triad (Cys178, His216, and Asp231) that is structurally similar to members of the cysteine protease superfamily and possesses glutamine deamidase activity. It was also shown to act on host proteins such as α-subunit of heterotrimeric GTPases leading to its constitutive activation (Bhaskaran and Stebbins, 2012; Brink et al, 2018). What can then be the significance of the PTS1-like domain of SseI and how is it distinct from its published functions during infection? Our results indicate that the effect of SseI on cellular migration was independent of its peroxisome localization, as the PTS-1 mutant of SseI showed a similar effect on HeLa cell migration to the WT-SseI protein. However, PTS1 mutant stains (SseI-ΔGKM/SseI-K321A) showed less cholesterol accumulation in the peroxisome, clearly indicating the peroxisome targeted SseI helps in cholesterol transport to the SCV and opens a different perspective on the metabolic function of SseI.

In the *PEX5* KO HeLa cells, growth of the WT *Salmonella* was reduced significantly, indicating that peroxisome is a pro-*Salmonella* organelle. *Salmonella* showed colocalization with host peroxisome starting from 3 h of infection, and we observed this interaction with SCV in multiple cell types, including epithelial (HeLa cell line) and immune cells (human primary macrophages), without any increase in the peroxisome biogenesis. Some of the peroxisomal enzymes were upregulated after *Salmonella* infection, and its significance is yet to be determined. However, the activity of catalase, which is the most abundant peroxisomal enzyme, was not altered after *Salmonella* infection. Further, in the *C. elegans* infection model, peroxisome knockdown did not affect *Salmonella* growth. In *C. elegans*, *Salmonella* grows in the intestinal lumen, and SPI2 TTSS is non-essential (Labrousse et al, 2000), making us believe that the peroxisomal colocalization was indeed crucial for intracellular vacuolar growth. Accordingly, the Δ*sseI Salmonella* strain, which failed to colocalize with the peroxisome, showed a prominent growth defect compared to the WT *Salmonella*. This growth defect was seen in both cellulo and in vivo analysis, demonstrating the importance of peroxisomal colocalization for bacterial growth. Interestingly, we observe that replacing 'GKM' motif with 'SKL' prototype PTS1 does not rescue the growth defect of Δ*sseI* strain. This is because SseI::SKL is translocated to the peroxisomal matrix and is not available on the membrane for ARF1 activation.

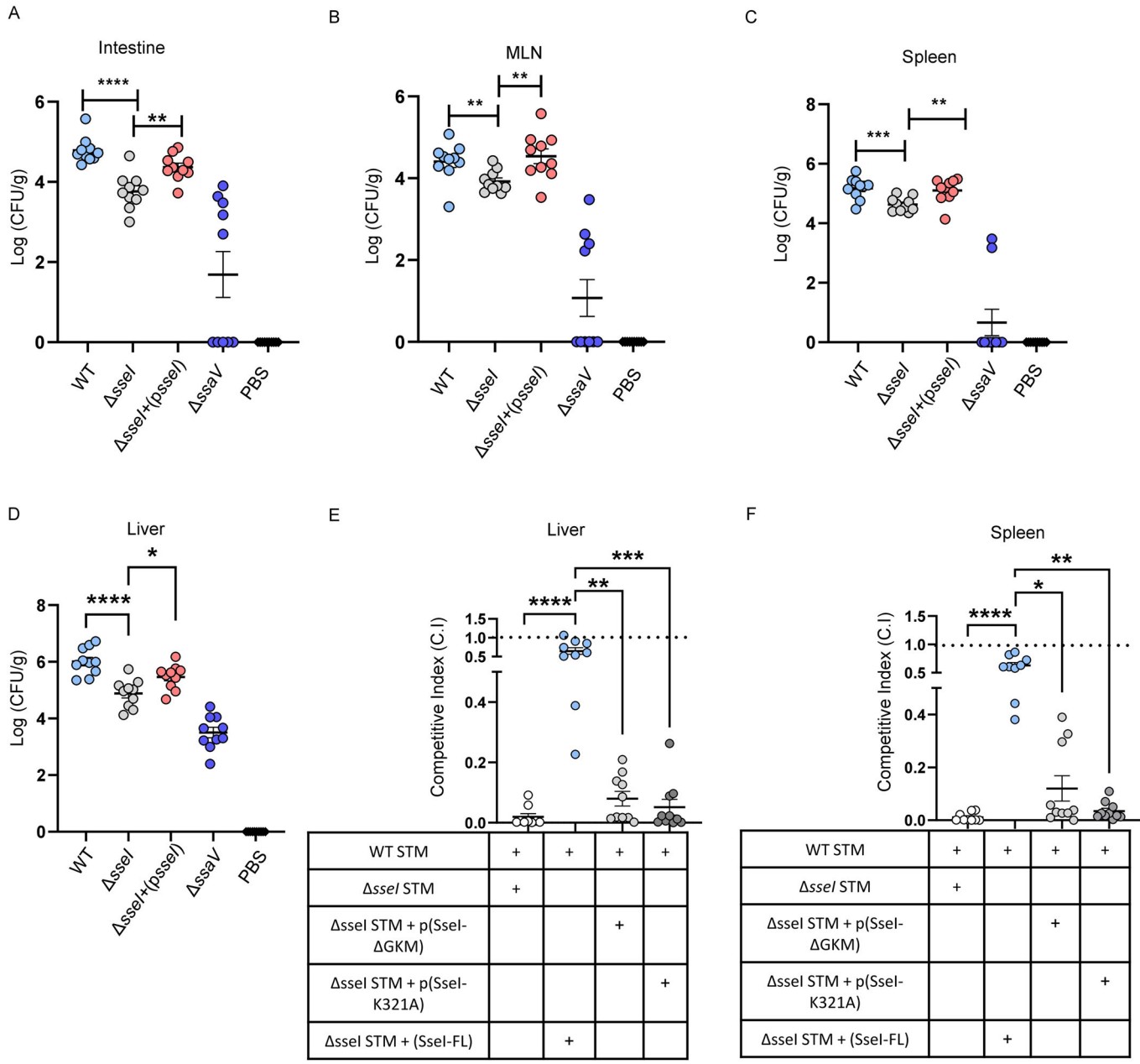

**Figure 6. SseI mutant strain is attenuated in the animal model of infection.**

(A–D) Intracellular STM burden in various organs. C57BL/6 mice (n = 10 per group) were orally infected with $10^7$ CFU of WT or ΔsseI STM, ΔsseI STM complemented with pQE60-SseI, or ΔssaV STM. PBS served as a vehicle control. Mice were sacrificed after 7 days, and STM burden was enumerated by plating. CFU/g of intestine (A), MLN (B), spleen (C), and liver (D) were determined. (E, F) The competitive index of SseI complements strains in the liver (E) and spleen (F). C57BL/6 mice (n = 9 per group) were orally infected with a mixed inoculum containing WT STM, ΔsseI STM, and ΔsseI STM complemented with pQE60-SseI clones (PTS1 deletion – ΔGKM, point mutation on PTS1 $K^{321}A$, and full length (FL)). Competitive index values were determined after 5 days post-infection by calculating the ratio of mutant to WT strains. Statistical analysis: Data were analyzed using student's t-test, (A) (**p = 0.0026, ****p < 0.0001), (B) (**p = 0.0098, **p = 0.0061), (C) (***p = 0.0009, **p = 0.0053), (D) (****p < 0.0001, *p = 0.0111). (E, F) Data were analyzed using one-way ANOVA (non-parametric Kruskal–Wallis test), (E) (**p = 0.0097, ***p = 0.0004, ****p < 0.0001), (F) (**p = 0.0013, *p = 0.0199, ****p < 0.0001). Figure 6 (A–D) was repeated with similar results in n = 6 mice per group. Source data are available online for this figure.

Therefore presence of SseI on the peroxisomal membrane rather than translocation into peroxisomes is crucial for its mechanism of action.

In vivo studies in mice using ΔsseI Salmonella strain have previously shown ambiguous results. At early time points of infection, ΔsseI strain showed enhanced dissemination, and at a later time, the mutant showed reduced growth in vivo after

intraperitoneal infection (Thornbrough and Worley, 2012; Carden et al, 2017). Here, we performed oral inoculation and observed that ΔsseI Salmonella was attenuated in all the organs tested after 7 days of infection. Similarly, co-infection of WT STM and ΔsseI STM showed a reduction in the competitive index, indicating that, indeed, SseI is essential for bacterial virulence. While

complementation of ΔsseI STM with SseI full length showed similar fitness with WT STM, complementation with SseI-ΔGKM or SseI-K321A, showed attenuation in virulence. This difference in the virulence attenuation between full-length complementation and ΔGKM/ K321A complementation highlights the significance of peroxisome targeting of SseI by GKM motif (McLaughlin et al, 2009). This result indicates that GKM-mediated binding of SseI to peroxisomes is essential for the in vivo virulence of the protein, and possibly, the peroxisomal alteration is the dominant effect of SseI.

Cholesterol serves as the central lipid of mammalian cells that participates in various functions, including signal transduction, steroidogenesis, bile acid synthesis, and regulating biological membrane properties (such as curvature, fluidity, and membrane fusion). In addition, cholesterol plays a critical role during infection (Tall and Yvan-Charvet, 2015). For example, HIV and influenza virus (Manes et al, 2003) utilize cholesterol-rich regions on the plasma membrane to facilitate their entry. Further, bacterial pathogens, including Mycobacterium, employ host cholesterol catabolism to acquire nutrients for their survival (Griffin et al, 2012). Studies in Salmonella show that cholesterol is important for the formation of its membranous niche, SCVs. In addition, Salmonella modulates the host cholesterol biosynthetic pathway to acquire the non-sterol precursors such as squalene and squalene oxide (Catron et al, 2004). It is now believed that while non-sterol precursors are obtained from the cholesterol de novo biosynthesis pathway, cholesterol needed for SCV membrane is obtained from low-density lipid (LDL). LDL in the systemic circulation binds to LDL receptors on the plasma membrane and gets internalized. This internalized LDL then gets processed in the lysosomes, releasing free cholesterol. Free cholesterol from lysosomes gets transported to peroxisomes. This transfer is mediated by the interaction of PIP2 on peroxisomes and lysosomal Syt7. It is also shown that peroxisomes redistribute lysosomal-derived free cholesterol to ER (Xiao et al, 2019). Previous studies have also indicated that SCVs are rich in cholesterol (Catron et al, 2002; Kolodziejek et al, 2019). However, studies show that SCVs either exhibit a delayed fusion with lysosomes or fuse with lysosomes whose potency was reduced by Salmonella effectors (Walch et al, 2021). Therefore, how lysosomal-derived cholesterol is directed to the vacuoles-containing bacteria was unclear.

In the current study, we show that Salmonella infection increases lysosome-peroxisome and peroxisome-SCV contacts, thereby transporting cholesterol. Knockdown of lysosomal or peroxisomal tether, Syt7 or ABCD1, respectively, affects bacterial replication by perturbing the cholesterol transfer to peroxisomes and subsequently to SCVs. We therefore demonstrate that, SCVs acquire LDL-derived cholesterol from lysosomes enrouted through peroxisomes. This is achieved by targeting SseI to peroxisomal membrane, which enhances SCVs interaction with peroxisomes. In the absence of functional peroxisomes, SCVs do not form membranous extensions called SIF, which are essential for their stability and bacterial growth. Our results align with the previous hypothesis that SPI2 is needed for cholesterol accumulation in SCV (Catron et al, 2002), and we also provide a compelling mechanism supporting this. Reduced cholesterol transport to the SCV during the ΔsseI strain infection prevents the formation of SIFs, thereby causing immature SCVs. Consequently, there is diminished replication of intracellular Salmonella.

Further, mechanistically, we identified that SseI binds and activates host GTPase ARF1 to assist in this 'lysosome-peroxisome-SCV' interaction. Our pull-down and ELISA data demonstrated that activated ARF1 enhances PIP2 levels on the peroxisomes during Salmonella infection. ARF proteins belong to the Ras superfamily of small GTPases known to act as regulators of vesicular trafficking (Casalou et al, 2020). Like any other small GTPases, ARF family proteins also undergo cycling between GTP-bound (active) and GDP-bound (inactive) states. Multiple GTPase activating proteins (GAPs) and GEFs have been identified to carry out this ARF cycling for specific cellular functions. It is known that certain GAPs of ARF1 (ASAP1) and ARF6 (ACAP1 and ADAP1) are recruited to the plasma membrane at the positions of Salmonella entry (Davidson et al, 2015) and Salmonella effector SopF uses ARF1 as cofactor to activate host autophagy (Xu et al, 2022). Depleting ARF GAPs is shown to reduce bacterial invasion (Humphreys et al, 2013). Here, we show another mechanism by which Salmonella exploits peroxisomal ARF1. We believe that SseI functionally mimics GEF for ARF1, causing GTP binding and subsequent activation. We show that levels of activated ARF1 on isolated peroxisomes increased after Salmonella infection. This aids in enhanced peroxisomal PIP2 generation with the help of non-canonical peroxisomal PI5P4K2A isoform. This peroxisomal PIP2 forms a bridge between 'lysosome-peroxisome-SCV' using Syt7 and conveys the lysosomal free cholesterol to SCV.

Growing evidence has highlighted the importance of host-targeted therapeutics that could be developed against numerous related bacteria, and our study has long-standing implications in this regard. In addition, Inflammatory bowel disease patients have higher peroxisome content in the intestinal mucosa (Du et al, 2020) and fail to efficiently clear pathogens, including Salmonella (Lahiri and Abraham, 2014), thus supporting our findings. Studies have also shown that patients with Salmonella Typhi (S. Typhi) infection exhibit higher cholesterol levels. Further, LDL receptor knockout animals are resistant to Salmonella infection by affecting the uptake of bacteria (Netea et al, 2009). In addition, human genetic variation in the VAC14 component of the PIKFYVE complex (VAC14), which leads to altered cholesterol flux, influences susceptibility to S. Typhi infection. Specifically, reduced VAC14 expression led to enhanced cholesterol in the cell membrane, facilitating Salmonella invasion (Alvarez et al, 2017). In addition, cholesterol-depleting drugs such as ezetimibe enhance the clearance of S. Typhi in zebrafish (Alvarez et al, 2017). Here, we provide compelling evidence that LDL-derived free cholesterol is essential for SCV/SIF integrity and Salmonella proliferation. The SseI-targeted peroxisomes are crucial in providing the same to the growing SCV.

In the current study we show that SseI activates the ARF1 and induces PIP2 synthesis on peroxisomes. However, it is yet to be determined whether SseI activates ARF1 in the cytoplasm and takes it to peroxisomes or SseI acts upon inactivate ARF1 present on peroxisomes. We have partly answered this by showing that SseI devoid of GKM motif does not increase active ARF1 level on peroxisomes. Therefore, we believe that activation of cytoplasmic ARF1 by SseI is not sufficient to increase the peroxisomal ARF1 level. In addition, tagging SseI to SKL also affects ARF1 activation insisting on the importance of membrane localization of SseI. Further, we see that PIP2 synthesis on peroxisomes occurs in a PI5P4K2A-dependent manner. Finally, we observed that the organellar levels of PIP2 are regulated by PI5P4K2A, and their specific inhibition reduced Salmonella proliferation. We speculate that PI5P4K inhibition could be a new host-targeted therapy in gastroenteritis and typhoid fever. However, the mechanism of the interaction between Arf1 and PI5P4K2A on peroxisome and how this helps in SCV cholesterol accumulation is yet to be determined.

# Methods

**Reagents and tools table**

| Reagent/Resource | Reference or Source | Identifier or Catalog Number |
|---|---|---|
| **Experimental models** | | |
| HeLa cells (*H. sapiens*) | ATCC | *CCL-2 LOT: 70056809* |
| Monocytes derived macrophages (*H. sapiens*) | Blood bank at the Transfusion Medicine - King Georges' Medical University | |
| *Salmonella* Typhimurium | Kind gift from Prof. Michael Hensel | ATCC, 14028 |
| *C57BL6/J (M. musculus)* | Central Animal Facility, IISc Bangalore, India. | |
| **Recombinant DNA** | | |
| pLJC5-3XMyc-EGFP-PEX26 | Addgene | 139059 |
| pcDNA3 flag HA: SseI | This study | Vector backbone from #9021, Addgene |
| pcDNA3 flag HA: ARF1 | This study | Vector backbone from #9021, Addgene |
| pFPV-mCherry | Addgene | 20956 |
| pET23A His tagged SseI | This study | N/A |
| pQE60 myc::ssel | This study | N/A |
| pQE60 ssel::myc | This study | N/A |
| SseI, Syt7, and ARF1 (along with mutants) were synthesized commercially | GenScript | |
| **Antibodies** | | |
| Rabbit Anti-LAMP1 | Cell Signalling Technology (CST) | D401S |
| Anti-FLAG | CST | 8146 |
| Anti-myc | CST | 2276 |
| Anti-Pex14 | ABclonal | A7336 |
| Anti-ESYT1 | ABclonal | A15410 |
| Anti-ABCD1 | Proteintech | 60153-1-Ig |
| Anti-pex5 | Proteintech | 12545-1-AP |
| Anti-ARF1 | Proteintech | 10790-1-AP |
| Anti-GAPDH | Invitrogen | T0004 |
| Anti-HA | CiteAb | 2367 |
| Anti-SYT7 | Novus Biologicals | NBP2-22420 |
| PIP2 antibody | Novus Biologicals | NBP2-76433 |
| Goat anti-mouse IgG Alexa fluor 594 | CiteAb | A-11032 |
| Goat anti-Rabbit IgG Alexa Fluor 488 | CiteAb | A-11034 |
| Goat anti-rabbit IgG Cy5 | Invitrogen | A10523 |
| Control Mouse immunoglobulin | Invitrogen | 500-M00-1MG |
| Control rabbit immunoglobulin | PeproTech | 500-P00-500µg |
| horseradish peroxidase | CST | 7074P2 |
| **Oligonucleotides and other sequence-based reagents** | | |
| ARF1 | Santa Cruz | SC-105086 |
| Synaptotagmin VII (SYT7) | Santa Cruz | SC-41320 |
| E-Syt1 | Santa Cruz | SC-95714 |

| Reagent/Resource | Reference or Source | Identifier or Catalog Number |
|---|---|---|
| NPC1 | Santa Cruz | SC-41588 |
| ABCD1 | Santa Cruz | SC-41143 |
| On-Target plusTM Control Pool | Dharmacon DM | D-001810-10-20 |
| **RT Primers** | | |
| PEX11B FWD | CTTCAGTGCTCAGAGCCAAGCC | |
| PEX11B REV | GCATGGCCAAGAAGAGAGCAA | |
| PEX14 FWD | CAGAAAGATGGCGTCCTCGG | |
| PEX14 REV | TCATCTGTCAGCCCTTTCTTCT | |
| PEX19 FWD | CCCCCACAGACAGTGAAACC | |
| PEX19 REV | GTTGAGGCCAGGAGGCATCTC | |
| ABCD1 FWD | GCCCTCCCTGTGGAAATACC | |
| ABCD1 REV | AGCTTCTCGAACTTCCAGCC | |
| ARF1 FWD | GTTTGCTGTGAAGACGGTGTC | |
| ARF1 REV | AGAGGATCGTGGTCTTCCCT | |
| GAPDH FWD | TCGGAGTCAACGGATTTGGT | |
| GAPDH REV | TTCCCGTTCTCAGCCTTGAC | |
| ACBD5::FWD HU | CTGGAAACGCTGACTGCTTTGC | |
| ACBD5::REV HU | CGTTAGCACACCAGGAGACATC | |
| FAR1::FWD HU | GTGGTCTCTTTATTGCGGCAGG | |
| FAR1::REV HU | AATACCAGGCTGCCGCAAGACT | |
| AGPS::FWD HU | GGCTGGCATAACAGGACAAGAG | |
| AGPS::REV HU | CTTCATGCCTGATGCGCGAGTA | |
| HSD17B4::FWD HU | GAGAATGCCAGCAAGCCTCAGA | |
| HSD17B4::REV HU | GCTGTAGACGTTGCACGACTAG | |
| ACAA1::FWD HU | GACAGGTCATCACGCTGCTCAA | |
| ACAA1::REV HU | CCAGGGTATTCAAAGACGGCAG | |
| PPARA::FWD HU | TCGGCGAGGATAGTTCTGGAAG | |
| PPARA::REV HU | GACCACAGGATAAGTCACCGAG | |
| DHCR24::FWD HU | CAGGAGAACCACTTCGTGGAAG | |
| DHCR24::REV HU | CCACATGCTTAAAGAACCACGGC | |
| HMGCR::FWD HU | GACGTGAACCTATGCTGGTCAG | |
| HMGCR::REV HU | GGTATCTGTTTCAGCCACTAAGG | |
| LDLR::FWD HU | GAATCTACTGGTCTGACCTGTCC | |
| LDLR::REV HU | GGTCCAGTAGATGTTGCTGTGG | |
| NPC1 FWD | CAGCTCCGTGTTCAGTGGAA | |
| NPC1 REV | TGGCTTTATTTACTGATGGCCC | |
| ESYT7 FWD | CTGAATTGTACCCACCACAGC | |
| ESYT7 REV | TGGTACTGTAGACAGCCTTGC | |
| SYT7 FWD | TGGAGACCAAGGTGAAGCGGAA | |
| SYT7 REV | TGAAGCGGTCATAGTCCAGGAC | |
| **Silencing primers** | | |
| LDLR huSHRNA 1 FWD | CCGGTGGGCGACAGATGCGAAAGAAATGTGCTTTTTCTTTCGCATCTGTCGCCCTTTTTG | |
| LDLR huSHRNA 1 REV | AATTCAAAAAGGGCGACAGATGCGAAAGAAAAAGCACATTTCTTTCGCATCTGTCGCCCA | |
| PIP4K2A huSH 1 FWD | CCGGTATACATCATCAAGACTATTACTGTGCTTGTAATAGTCTTGATGATGTATTTTTTG | |

| Reagent/Resource | Reference or Source | Identifier or Catalog Number |
| --- | --- | --- |
| PIP4K2A huSH 1 REV | AATTCAAAAAATACATCATCAAGACTATTACAAGCACAGTAATAGTCTTGATGATGTATA | |
| PIP4K2B huSH 1 FWD | CCGGTGTTAGGGAGAAGGGTGTATTTTGTGCTTAAATACACCCTTCTCCCTAACTTTTTG | |
| PIP4K2B huSH 1 REV | AATTCAAAAAGTTAGGGAGAAGGGTGTATTTAAGCACAAAATACACCCTTCTCCCTAACA | |
| **Cloning primers** | | |
| EGFP::SKL/GKM FWD HIND III | CCCAAGCTTATGGTGAGCAAGGGCGAGGAG | |
| EGFP::SKL RVS ECOR1 | GGGGAATTCTTATAACTTGGACTTGTACAGCTCGTCCAT | |
| EGFP::GKM RVS ECOR1 | GGGGAATTCTTACATTTTACCCTTGTACAGCTCGTCCAT | |
| HA-SSEI::SKL FWD HIND III | CCCAAGCTTATGTACCCATACGATG | |
| HA-SSEI::SKL REV ECOR1 | GGGGAATTCTTATAACTTGGATATTAAGGAATATTTTTGC | |
| SSEI: SKL PQE60 REV | CCCAAGCTTTTATAACTTGGATATTAAGGAATATTTTTGC | |
| PQe60::ssei W/O MYC FWD | CGCGGATCCATGCCCTTTCATATTGGAAGC | |
| PQe60::ssei WT REV | CCCAAGCTTTTACATTTTACCTATTAAGGAATA | |
| PQe60::ssei-GKM REV | CCCAAGCTTTTATATTAAGGAATATTTTTGCT | |
| PQe60::ssei K321A REV | CCCAAGCTTTTACATCGCACCTATTAAGGAATA | |
| SseI KO FWD | TTTGTTATGCCCTTTCATATTGGAAGCGGATGTCTTCCCATATGAATATCCTCCTTAG | |
| SseI KO REV | CCTCCACGGTGCGCTTACATTTTACCTATTAAGGAATAGTGTAGGCTGGAGCTGCTTC | |
| **Guide RNAs for lentivirus production** | | |
| PEX5 Forward | 5'CACCGCACCATGGCAATGCGGGGAGC3' | |
| PEX5 Reverse | 5'AAACGCTCCCGCATTGCCATGGTGC3' | |
| Scrambled Forward | 5'CACCGCGGGACGTCGCGAAAATGTA3' | |
| Scrambled Reverse | 5'AAACTACATTTTCGCGACGTCCCGC3' | |
| **Chemicals, Enzymes and other reagents** | | |
| 4-PBA | Sigma-Aldrich | P21005 |
| Lovastatin | Cayman | 75330-75-5 |
| U-18666A | Cayman | 3039-71-2 |
| THZ-P1-2 | MCE | HY-13635 |
| DMEM | Gibco | 12800-058 |
| RPMI Medium 1640 | Gibco | 31800-014 |
| Ampicillin and Kanamycin | BR BIOCHEM | BC0286 |
| Penicillin and streptomycin | HIMEDIA | A001 |
| Bovine Serum Albumin | Sisco Research Laboratories (SRL) | 83803 |
| Triton X-100 | SRL | 64518 |
| RIPA Buffer | HIMEDIA | TCL131 |
| Protease inhibitor cocktail | HIMEDIA | ML051 |
| Tween-20 | GBiosciences | RC1226 |
| Collagenase H | Sigma-Aldrich | 33278623 |
| IPTG | Sigma-Aldrich | I67581G |
| Trypsin EDTA | HIMEDIA | TCL070 |
| Filipin | Sigma-Aldrich | SAE0087 |
| FBS | Gibco | 10270-106 |
| Macrophage colony-stimulating factor (M-CSF) | Genscript | Z02914 |
| Ni-NTA Agarose | Qiagen | 30210 |
| Protein-A Agarose | Invitrogen | 20333 |

| Reagent/Resource | Reference or Source | Identifier or Catalog Number |
|---|---|---|
| Filipin | Sigma | SAE0087 |
| **Software** | | |
| GraphPad Prism | | |
| Fiji | | |
| **Other** | | |

## Bacterial strains and growth condition

The *Salmonella enterica* serovar Typhimurium (STM WT) wild-type strain ATCC 14028s, generously provided by Prof. Michael Hensel. This strain was cultivated in Luria broth (LB-Himedia) overnight under vigorous shaking (180 rpm) at 37 °C using an orbital shaker (primary culture). This was followed by secondary culture incubation for 3 h at a rpm of 180 at 37 °C (1:33 ratio inoculum). Antibiotic selection such as Kanamycin/Chloramphenicol/Ampicillin was incorporated into the growth medium at final working concentrations of 50 µg/ml, 20 µg/ml, and 50 µg/ml, respectively, as per the requirement.

## Experimental study design

All experiments involving HeLa cell lines were performed in biological triplicates. Unless specified, all experiments were repeated three times independently with biological triplicates. No data was excluded from the study. The variation in the data is represented by error bars in the graphs. Sample allocation into experimental groups was random. The relevant cell lines for control and test were cultured together until before infection/compound treatment. Different people performed the experiment and data analysis for blinding. Samples were labeled with generic codes during data analysis and not revealed to the investigator.

## Bacterial gene knock-out generation

Gene knock-out in bacteria was done using a one-step chromosomal gene inactivation strategy (Datsenko and Wanner, 2000). Briefly, primers were designed for the amplification of the Kanamycin resistance gene cassette from the PKD4 plasmid. The 5′terminus of the primers had a sequence homologous to the flanking region of the gene to be knocked out (here, SseI). After amplifying the Kanamycin resistance gene cassette, the amplified products were purified using chloroform-ethanol precipitation. The purified product was then electroporated into the STM WT strain (expressing PKD-46 plasmid to provide λ-Red recombinase system) by a single pulse of 2.25 kV. Immediately fresh recovery media was added, and it was then incubated at 37 °C for 60 min in an orbital-shaker. After incubation, the cultures were centrifuged at 8000 rpm for 6 min. Then, the pellet was dissolved in 100 µL of media and plated at the required dilution on LB agar plates with Kanamycin. The plates were incubated at 37 °C for 12–14 h. The colonies were selected and were confirmed for knock-out using PCR, which was then run on 1% agarose gel to compare the length of the products in the mutant from STM WT bacteria.

## Isolation of monocyte-derived macrophages (MDM)

Buffy coats (≈30 ml) from healthy human donors were obtained from the blood bank at the Transfusion Medicine - King George's Medical University, Lucknow (UP) India. The PBMCs were isolated as follows: Buffy coat was diluted 1:2 with 1X-PBS. Then, PBMCs were isolated using density gradient centrifugation using Histopaque-1077 gradient (Himedia, India). Centrifugation at $400 \times g$ for 30 min was performed at room temperature. After isolation, cells were washed twice using PBS and resuspended in RPMI-1640 (GIBCO) media containing antibiotics without FBS. Cells were seeded (around $2 \times 10^7$ to $3 \times 10^7$ PBMCs/ml in 20 ml) and allowed to adhere for 2 h in a 5% $CO_2$ incubator at 37 °C. Non-adherent cells were removed from the dish. Adherent cells were then cultured with complete maturation media (RPMI-1640 with 10% FBS, penicillin/streptomycin, 10 ng/ml M-CSF) for 5 days for MDM differentiation. Media were changed every 2–3 days. Studies involving human participants were approved by the Institutional Ethics Committee (IEC) of CDRI and SGPGI (Approval reference numbers: CDRI/IEC/2024/A6; CDRI/IEC/2024/A2; 2023-255-EMP-EXP-54). Informed consent was obtained from all the participants, and the experimental procedures were pre-approved by the IEC.

## Plasmid and siRNA transfection

Cells were seeded on appropriate plate and allowed overnight for attachment. In case of microscopy experiments, cells were seeded on plates containing coverslips. Cells were then transfected with plasmids using Lipofectamine-3000, Invitrogen and incubated for 48 h according to the manufacturer's instructions. For siRNA constructs, cells were transfected using Lipofectamine RNAiMAX (ThermoFisher) according to the manufacturer's instructions.

**Gentamicin protection assay:**

1. A single colony of WT/ΔsseI/ΔssaV STM was grown in Luria Broth with constant shaking at a rpm of 180 at 37 °C overnight. This is followed by secondary culture inoculation (1:37 ratio inoculum) and was grown for 3 h at a rpm of 180 at 37 °C.
2. For intracellular infection, cells were infected with WT STM or ΔsseI or ΔssaV STM at a multiplicity of infection (MOI) of 10 (for intracellular survival assay) and MOI 50 (for qRT-PCR and immunofluorescence). The plate containing infected cells was centrifuged at 600 rpm for 10 min to facilitate the proper adhesion.
3. The plate was then incubated for 25 min at 37 °C in a humidified chamber and 5% $CO_2$.
4. Following infection, media is removed from cells and washed with PBS. Fresh media containing 100 µg/mL gentamicin was added and incubated for 60 min at 37 °C humidified chamber and 5% $CO_2$.

5. After incubation, fresh media containing 25 μg/mL of gentamicin was added.

6. The plate was incubated in a humidified chamber at 37 °C, and 5% $CO_2$ was used for either intracellular survival assay or immunofluorescence.

**Intracellular survival and invasion assay:**

1. Cells were seeded on 12-well plates and were allowed to attach overnight.

2. Transfection of plasmid constructs was done using lipofectamine 3000 (Invitrogen) as per the manufacturer's instructions.

3. After 48 h post-transfection, STM infection was conducted as mentioned above.

4. Following the gentamycin protection assay, the cells were lysed using 0.1% Triton X and added 1X PBS at the appropriate time points (2 h and 16 h post-infection).

5. The samples collected from each well were plated at the required dilutions on LB agar plates and kept at 37 °C for 12–14 h.

6. Post incubation, CFU were enumerated for each plate.

7. The fold proliferation and percentage invasion were determined as follows:

$$Fold\ Proliferation = CFU\ at\ 16\,h / CFU\ at\ 2\,h$$

$$Percentage\ Invasion = [CFU\ at\ 2\,h / CFU\ of\ the\ pre-inoculum] \times 100$$

**Immunofluorescence:**

1. Cells were seeded on coverslips and were allowed to attach overnight.

2. Adhered cells were infected as mentioned above in the gentamycin protection assay.

3. After infection, cells were fixed using 2% paraformaldehyde for 15 min at 37 °C.

4. For antibody staining, permeabilization buffer (Triton X-100 (0.3%)/1% BSA in 1X-PBS) containing primary antibody dilution was used according to the manufacturer's instructions and incubated overnight at 4 °C (For experiments involving filipin staining, permeabilization buffer containing 50 μg/ml filipin along with primary antibody dilution was used according to the manufacturer's instructions and incubated for 2 h at room temperature).

5. This was followed by secondary antibody incubation for 1 h at room temperature. The coverslips were mounted with 50% glycerol.

## Fluorescence microscopy and analysis

Images were acquired on BX61-FV1200MPE, Olympus Life Science, using either 100× or 63× objective (oil immersion). Post-acquisition, images were processed using Olympus software (FV10-ASW 4.1) to prepare the representative images. For experiments that involved quantitating colocalization, individual Z sections were analyzed using the "colocalization threshold" plugin in ImageJ. For experiments that involved counting puncta/particles in the cell, the "Analyze particles" plugin of ImageJ was used. Graphs were plotted, and significance levels were obtained using GraphPad Prism.

## Live cell imaging

HeLa cells were seeded onto a glass bottom live cell imaging dishes. At around 60% confluency, cells were transfected with PEROXO-Tag plasmids using polyethylenimine (PEI). Forty-eight hours post-transfection, cells were infected with mCherry-tagged WT STM at MOI of 50. After 30 min, cells were washed twice with 1X-PBS and incubated in a fresh DMEM medium containing 25 μg/ml of gentamicin. Imaging was performed using a Leica DMI 6000 microscope at 37 °C and 5% $CO_2$ till the required time points.

**Immunoblotting:**

1. Following appropriate treatment/infection, cells were lysed in lysis buffer containing protease inhibitor cocktail (PIC) (lysis buffer composition: 50 mM Tris HCl, 150 mM NaCl, 1.0% (v/v) NP-40, 0.5% (w/v) Sodium Deoxycholate, 1.0 mM EDTA, 0.1% (w/v) SDS and 0.01% (w/v) sodium azide at a pH of 7.4).

2. The samples were kept at 95 °C for 10 min and electrophoresed on SDS-PAGE.

3. Following SDS-PAGE separation, samples were transferred onto the PVDF membrane (#1620177, Bio-Rad).

4. Primary and secondary antibody treatments were done according to manufacturers protocol.

5. The signals were obtained using a chemiluminescence substrate (#170-5061, Bio-Rad). The bands were quantified using ImageJ software (NIH).

## Protease degradation assay

SseI-HA construct was transfected into PEROXO-Tag expressing HeLa cells. After 48 h, the cells were used for peroxisome isolation via PEROXO-IP. Isolated peroxisome samples were treated with digestion buffer (150 μg/ml Proteinase K + 300 μg/ml trypsin) in the presence or absence of 1% Triton X-100 (Wang et al, 2017) for indicated time points. The samples were then loaded on SDS-PAGE for immunoblotting.

## ARF1 activation assay

HeLa cells ($2 \times 10^7$ cells per group) were transfected with the PEROXO-tag (3XMyc-EGFP-PEX26). After 48 h, the cells were used for peroxisome isolation via PEROXO-IP as described previously. Isolated peroxisome samples and whole cell lysate were used for pulldown of Active ARF1 levels according to the manufacturer's instructions using the ARF1 activation Assay Biochem kit (Cat.# BK032-S) and the ARF1 activation kit (Cell Biolabs, STA-407-1).

## SseI antibody generation in mice

For primary immunization, a mixture of purified SseI protein and Freund's complete adjuvant (1:1 v/v) 50 μg (per mice) was subcutaneously administered in the mice (Leenaars and Hendriksen, 2005). After 28 days, the first booster dose of 40 μg purified protein and Freund's incomplete adjuvant (1:1) was given subcutaneously. In a similar manner, a second booster dose was administered after 10 days. After 10 days of the second

booster, mice were sacrificed by cardiac puncture, and blood was collected. Serum was isolated by centrifugation at 6000 rpm for 30 min at 4 °C and stored at −20 °C. Western blotting was done using this antiserum as the primary antibody (Hanly et al, 1995).

## Peroxisome isolation by PEROXO-Tag IP

Cells (~40 million HeLa cells) were transfected with the PEROXO-tag (3XMyc-EGFP-PEX26) (Ray et al, 2020). After 48 h of transfection, cells were rinsed twice with pre-chilled PBS and then scraped in 1 ml of KPBS and pelleted at $1000 \times g$ for 2 min at 4 °C. Cells were then resuspended in 1000 µl of KPBS and gently homogenized with 25 strokes for 10 s in a 2 ml homogenizer. It was then centrifuged at $1000 \times g$ for 2 min at 4 °C to pellet nuclei and cells while cellular organelles, including peroxisomes, remained in the supernatant. The supernatant was incubated with the anti-Myc antibody at 4 °C overnight on the rocker. The next day, it was incubated with protein A beads for 4 h at 4 °C and was centrifuged at $1000 \times g$ for 2 min. The bound peroxisomes were resuspended in 1X SDS lysis buffer to extract the proteins.

## SseI protein purification and SseI-ARF1 in vitro and in vivo binding

SseI (322 residues) was cloned into the pET23a vector with His tag. The construct was transformed in *Escherichia* Coli BL21 (DE3) strain, and the protein was purified following standard protocol.

Overexpressed ARF1-HA cell lysates were incubated with purified His-SseI for 2 h at 4 °C quantity of proteins. The samples were then incubated with Ni-NTA beads overnight at 4 °C. The beads were washed, and bound proteins were eluted by boiling them in an SDS-sample buffer. All elutes were subjected to SDS-PAGE followed by coomassie staining or immunoblotting.

## Immunoprecipitation experiments

After transfection with the required plasmids, cells were lysed using a RIPA lysis buffer containing PIC. The cell lysates were incubated overnight at 4 °C with the respective primary antibodies or control-IgG. The antibody-bound proteins were pulled down by protein A beads and subjected to immunoblotting analysis.

## Lentivirus production

Guide RNAs targeting human PEX5 and scrambled were cloned in lentiCRISPRv2 plasmid using BsmBI restriction site.

For lentivirus production, cloned lentiCRISPRv2 plasmid was transfected in HEK293T cells together with helper plasmid pMD2G and psPAX2 with the help of PEI-MAX 40000. After 72 h of transfection, the culture medium containing lentiviral particles was collected for three consecutive days and concentrated using PEG8000. For the generation of knockout, 50% confluent cells were mixed with lentiviral supernatant into the culture medium in a 1:1 ratio along with polybrene (8 µg/ml). After 24 h of transduction, the media (DMEM) was changed to contain 10% FBS/antibiotic. Cells were then incubated in media with 2 µg/ml puromycin for selection.

## RNA extraction and quantitative PCR

Total RNA from cells was isolated using TRIzol (Ambion). Reverse transcription was carried out using a ThermoScientific cDNA synthesis kit (# K1622). Peroxisome and trafficking pathway gene primers were designed by NCBI and purchased from Sigma. The housekeeping gene, GAPDH, was used as the normalizing control to calculate the fold change.

## In vivo animal experiment

5–6-week-old C57BL/6 mice were infected orally by gavaging $10^7$ CFU of either WT STM or the ΔsseI STM strain. 7 days' post-infection, the mice were sacrificed, and the intestine, MLN, spleen, and liver were isolated aseptically. The tissue samples were crushed using 1 mm glass beads in a bead-beater, and the supernatant was plated onto SS agar and incubated at 37 °C. CFU was enumerated after 16 h, and organ load was determined by normalizing the CFU to the weight of the tissue sample. Studies involving mouse strains were approved by the Institutional Animal Ethics Committee (IAEC) of IISc, Bangalore (Approval reference number: CAF/Ethics/852/2021). Experiments were performed in compliance with guidelines provided by The Committee for the Purpose of Control and Supervision of Experiments on Animals (CPCSEA).

## Competitive index (CI)

5–6-week-old C57BL/6 mice were co-infected by orally gavaging equal CFU of $10^7$ CFU/strain (input) of WT STM and ΔsseI STM/ΔsseI STM complement strains. The tissues from the spleen and liver were aseptically isolated 5 days post-oral gavage. Homogenized tissues were plated on Salmonella-Shigella agar and LB agar containing appropriate selection markers (output). The competitive index (CI) between WT vs. mutant or WT vs. complement strains was calculated as the ratio of output and input as follows:

$$CI = (CFU \text{ output of mutant or complement}/ \text{ CFU output of STM WT})/ \\ (CFU \text{ input of mutant or complement}/ \text{ CFU input of STM WT})$$

Data is represented as CI of mutant and complemented strains with WT.

## Mass spectrometry sample preparation

Protein samples were used for digestion and reduced with 5 mM TCEP and further alkylated with 50 mM iodoacetamide and then digested with Trypsin (1:50, Trypsin/lysate ratio) for 16 h at 37 °C. Digests were cleaned using a C18 silica cartridge to remove the salt and dried using a speed vac. The dried pellet was resuspended in buffer A (2% acetonitrile, 0.1% formic acid).

## Mass spectrometric analysis of peptide mixtures

Experiments were performed on an Easy-nlc-1000 system coupled with an Orbitrap Exploris mass spectrometer. 1 µg of peptide sample was loaded on C18 column 15 cm, 3.0 µm Acclaim PepMap (Thermo Fisher Scientific) and separated with a 0–40% gradient of buffer B (80% acetonitrile, 0.1% formic acid) at a flow rate of 300 nl/min) and injected for MS analysis. LC gradients were run for

60 min. MS1 spectra were acquired in the Orbitrap (Max IT = 25 ms, AGQ target = 300%; RF Lens = 70%; R = 60 K, mass range = 375–1500; Profile data). Dynamic exclusion was employed for 30 s, excluding all charge states for a given precursor. MS2 spectra were collected for the top 12 peptides. MS2 (Max IT = 22 ms, R = 15 K, AGC target 200%).

## Mass spectrometric data processing

All samples were processed, and raw files generated were analyzed with Proteome Discoverer (v2.5) against the UniProt Human database. For dual Sequest and Amanda search, the precursor and fragment mass tolerances were set at 10 ppm and 0.02 Da, respectively. The protease used to generate peptides, i.e., enzyme specificity, was set for trypsin/P (cleavage at the C terminus of "K/R: unless followed by "P"). Carbamidomethyl on cysteine as fixed modification and oxidation of methionine, and N-terminal acetylation were considered as variable modifications for database search. Both peptide spectrum match and protein false discovery rate were set to 0.01 FDR. Mass spectrometry experiment and data registration in PRIDE repository was outsourced to Vproteomics, New Delhi, India.

## Cell migration assay

HeLa cells were grown in 6-well plates and transfected with either full-length SseI or ΔGKM-SseI constructs. After 36 h of transfection, the wound was created using 6 well wound markers, and the wound was imaged using light microscopy at 0 h and 36 h, with 5× magnification. All images were analyzed by Fiji software using the 'wound healing size tool' plugin to measure wound distance in the cells.

Phosphatidylinositol (4,5) bisphosphate mass ELISA assay

1. The PI(4,5)P2 levels extracted from peroxisomes were measured using a competitive ELISA assay (Echelon Biosciences Inc, K-4500). Briefly, media was aspirated from HeLa cells ($4 \times 10^7$ cells per group)with/without infection, and 1 ml of ice-cold 0.5 M Trichloroacetic Acid (TCA) was added.
2. After 5 min of incubation, cells were scraped and transferred into a 2 ml centrifuge tube. Centrifugation at 3000 rpm for 7 min at 4 °C was carried out.
3. The pellet was washed using 1 ml of 5% TCA/1 mM EDTA for 5 min and the supernatant was discarded.
4. Neutral lipids were extracted using 1 ml of methanol:chloroform in a ratio of 2:1 and vortexed for 10 min. Centrifugation at 3000 rpm for 5 min was carried out and the supernatant was discarded. This step was repeated and the supernatants were discarded.
5. The acidic lipids were extracted by adding 750 µl of methanol:chloroform:HCl in the ratio of 80:40:1 and vortexed for 25 min. The supernatant was transferred to a 2 ml tube after centrifugation at 3000 rpm for 5 min. The pellet was discarded.
6. The phase split was obtained by adding 250 µl of chloroform and 450 µl of 0.1 N HCl and vortexed for 30 s.
7. Organic and aqueous phases were separated by centrifugation at 3000 rpm for 5 min.
8. The organic phase was collected into a clean vial and dried. The extracted lipid was resuspended in assay buffer.

9. The extractions, controls, and standards were set up in PI(4,5)P2 ELISA plate. A peroxidase-linked secondary detection solution and colorimetric substrate were used to detect PI(4,5)P2 detector protein bound to the plate.
10. The colorimetric signal was read at an absorbance of 450 nm. The PI(4,5)P2 levels extracted from peroxisomes were estimated by comparing the values in the standard curve.

### Catalase activity

1. The catalase activity of the cells infected with STM was measured using BioAssay Systems EnzyChrom catalase assay kit (Universal Biologicals Ltd., Cambridge, UK). Briefly, cells were homogenized in 200 µl cold PBS and centrifuged for 10 min at 14,000 rpm to pellet the debris and use the supernatant as the sample for the assay.
2. The assay involves adding 50 µM $H_2O_2$ as the substrate to the test samples/standard samples and the catalase reaction was initiated.
3. Following 30 min incubation, a detection agent along with HRP enzyme was added for detection.
4. Following 10 min of incubation, the change in color intensity was measured at an optical density of 570 nm.
5. The catalase activity of the sample was calculated by

$$\text{Catalase (U/L)} = \frac{R_{\text{Sample Blank}} - R_{\text{Sample}} \times n}{\text{Slope}(\mu M^{-1}) \times 30 \, \text{min}}$$

## Cholesterol assay

Cholesterol levels on isolated peroxisome samples were measured using an Amplex Red Cholesterol Assay kit (A12216). For the fluorometric quantification of cholesterol, isolated peroxisome, and cholesterol standards were incubated in 96-well plates according to the instructions of the manufacturer kit (A12216).

## C. elegans strain and maintenance

All strains were maintained on the nematode growth medium (NGM) plate seeded with E. coli OP50 as food at 25 °C. Adult worms were treated with diluted bleach to isolate eggs. The extracted eggs were age-synchronized at the first larval stage (L1) in M9 buffer at 20 °C for 16–18 h. The L1 worms were transferred to fresh plates for maintenance or RNAi treatment.

### C. elegans RNA interference treatment
E. coli (HT115) strains transformed with L4440 (empty vector), or prx-5 from the Ahringer library (Brenner, 1974) were used to seed NGM + Ampicillin + IPTG plates. L1 worms were grown on RNAi plates for 48 h, after which they were transferred to the corresponding infection plates (Kamath and Ahringer, 2003).

### Salmonella infection
Infection plates were prepared by seeding NGM plates with a 1:1 v/v mixture of RNAi with S. Typhimurium (mCherry). Plates seeded with RNAi mixed with E. coli HT115 (mCherry) were used as no-infection control. 5′-fluorodeoxyuridine (FUdR -300 µM) was added to achieve sterilization of the adult worms.

### C. elegans fluorescence microscopy

On day 8 of adulthood, worms were washed with M9 buffer and transferred to a 2% agarose pad on a glass slide. Then, worms were anesthetized with 30 mM sodium azide and covered with a coverslip. Images of the nematode digestive tract were taken using a Carl Zeiss fluorescence microscope, and the fluorescence intensity of RFP in the intestine was quantified using ImageJ software.

### Statistics

Statistical analyses were performed using GraphPad Prism software v9.5. Significance was referred as Student *t*-test, ANNOVA, \*, \*\*, \*\*\*, and \*\*\*\* for *p*-values < 0.05, <0.01, <0.001, and <0.0001, respectively.

## Data availability

The mass spectrometry proteomics data have been deposited to the ProteomeXchange Consortium via the PRIDE partner repository with the dataset identifier PXD055151.

The source data of this paper are collected in the following database record: biostudies:S-SCDT-10_1038-S44319-024-00328-x.

## Peer review information

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

## Acknowledgements

We would like to thank Dr. Suresh Subramani, Professor, University of
California, San Diego, for giving suggestions on our peroxisome-related work
and for providing critical comments on our manuscript. The work is supported
by financial grants from the Science and Engineering Board (SERB),
Government of India (SRG/2019/000268), DBT Wellcome Trust India
Alliance, ICMR Ad-hoc(Ortho) 2022-NCD-1, CSIR-MLP 2105, CSIR HCP-0047
and start-up funding from Director CSIR-CDRI to AL and PDF/2021/002843
(SERB), IA/E/21/1/506319 (DBT/Wellcome Trust IA) to VA. DR, JS, and SP
receive a fellowship from CSIR, and SS receives a fellowship from UGC, the
government of India. We want to thank technical help from Mrs. Reema Roy,
Dr. C.P. Pandey, and Mr. Anil Verma for microscopy and Dr. AL Vishwakarma
for Flow cytometry studies, Central Animal house facility at CDRI, Lucknow,
and IISc Bangalore and Sophisticated Analytical Instrument Facility and
Research Division, CSIR-Central Drug Research Institute. CSIR THUNDER
(BSC0102) and MOES (GAP0118) Intravital facility is acknowledged. We thank
Dr. Anil Gaikwad, Dr. Sachin Kumar, and Dr. Dipak Dutta for helping us with
the reagents. This manuscript has a CSIR-CDRI communication number:
10884.

## Author contributions

**Desh Raj**: Conceptualization; Data curation; Investigation; Methodology;
Writing—original draft; Writing—review and editing. **Abhilash Vijay Nair**:
Investigation; Methodology; Writing—original draft. **Anmol Singh**:
Investigation. **Swarnali Basu**: Investigation. **Kabita Sarkar**: Investigation.
**Jyotsna Sharma**: Investigation. **Shiva Sharma**: Investigation. **Sanmi Sharma**:
Investigation. **Manisha Rathore**: Investigation; Methodology. **Shriya Singh**:
Investigation. **Shakti Prakash**: Investigation; Methodology. **Simran**:
Investigation. **Shikha Sahu**: Investigation. **Aman Chandra Kaushik**:
Investigation. **Mohammad Imran Siddiqi**: Investigation. **Uday C Ghoshal**:
Resources; Investigation. **Tulika Chandra**: Resources; Investigation. **Vivek
Bhosale**: Investigation. **Arunava Dasgupta**: Investigation. **Shashi Kumar Gupta**:
Resources; Investigation; Methodology. **Sonia Verma**: Resources; Investigation;
Methodology. **Rajdeep Guha**: Resources; Investigation. **Dipshikha
Chakravortty**: Conceptualization; Resources; Data curation; Supervision;
Investigation; Writing—original draft. **Veena Ammanathan**: Conceptualization;
Resources; Data curation; Supervision; Funding acquisition; Investigation;
Methodology; Writing—original draft; Writing—review and editing. **Amit
Lahiri**: Conceptualization; Resources; Supervision; Funding acquisition;
Investigation; Writing—original draft; Writing—review and editing.

Source data underlying figure panels in this paper may have individual
authorship assigned. Where available, figure panel/source data authorship is
listed in the following database record: biostudies:S-SCDT-10_1038-S44319-
024-00328-x.

## Disclosure and competing interests statement

The authors declare no competing interests.

# Expanded View Figures

**Figure EV1. Peroxisomes are required for efficient intracellular replication of STM.**

(A) Immunoblot and densitometry analysis of PEX14 protein levels in whole cell lysates of HeLa cells after STM infection for 3, 6, and 12 h. Actin served as a loading control. Densitometry analysis quantified PEX14 levels in STM-infected cells relative to uninfected controls with loading control actin. Graph represent the mean ± SEM of six independent experiments. (B) Immunoblot and densitometry analysis of PEX3 protein levels in whole cell lysates of HeLa cells after STM infection for 6 and 12 h. Actin served as a loading control. Densitometry analysis quantified PEX3 levels in STM-infected cells relative to uninfected controls. Graph represent the mean ± SEM of three independent experiments. (C, D) Fold change in mRNA levels of peroxisomal genes FAR1 (C) and HSD17B4 (D) in HeLa cells post 6 and 12 h of STM infection compared to uninfected controls. Graph represent the mean ± SEM of three independent experiments. (E) Immunoblot and densitometry analysis of *PEX14* protein levels after knockdown (KD) in HeLa cells. The membrane was probed with an anti-PEX14 antibody to compare protein levels in WT and PEX14 KD HeLa cells with loading control GAPDH. Graph represent the mean ± SEM of three biological replicates. (F) Immunoblot confirming the generation of PEX5 knockout (KO) HeLa cells by the CRISPR/Cas9 system. The membrane was probed with anti-PEX5 AND anti-GAPDH antibodies to compare protein levels in WT and PEX5 KO HeLa cells. (G) Fluorescence microscopy images of peroxisomes in WT and PEX5 KO HeLa cells stained with anti-LAMP1 (red) and anti-PEX14 (green) antibodies. Scale bar: 10 μm. (H) Fluorescence microscopy images of peroxisomes in WT and PEX5 KO HeLa transfected with EGFP::SKL construct and stained with anti-ABCD1 (red) antibodies. Scale bar: 10 μm. (I) Graph representing cytoplasmic fluorescence intensity of EGFP::SKL in WT HeLa cell and PEX5 KO Cells. Data represent the ± SEM of three independent experiments. (J) Fold change in PEX14 mRNA expression after treatment with 4-PBA (2 mM) in HeLa cells. Graph represent the ± SEM of four biological replicates. (K) Change in fluorescence intensity was used to monitor the replication of mCherry-labeled STM or *E. coli* HT115 in the intestine of control, and Prx-5 silenced *C. elegans* worms. Imaging of the nematode digestive tract was performed on Day 8 of adulthood. Fluorescence intensity measurements were obtained from approximately 30 worms per experimental group. (L–N) Graphs representing changes in cell viability after silencing *PEX14* (L), in PEX5 KO HeLa cells (M), and after treatment with 4-PBA (2 mM) (N) in HeLa cells, as determined by MTT assay. Graph represent the ± SEM of three biological replicates. Data information: Data were analyzed using one-way ANOVA, Figures (A–C) (\*\*$p = 0.0015$), (D) (\*$p = 0.0363$) and (K). 'ns' denotes no significant difference. Data were analyzed using the students' *t*-tests, Figures (E) (\*$p = 0.0252$), (I) (\*\*\*\*$p = 0.0001$), (J) (\*$p = 0.0173$), (L–N), 'ns' denotes non-significant.

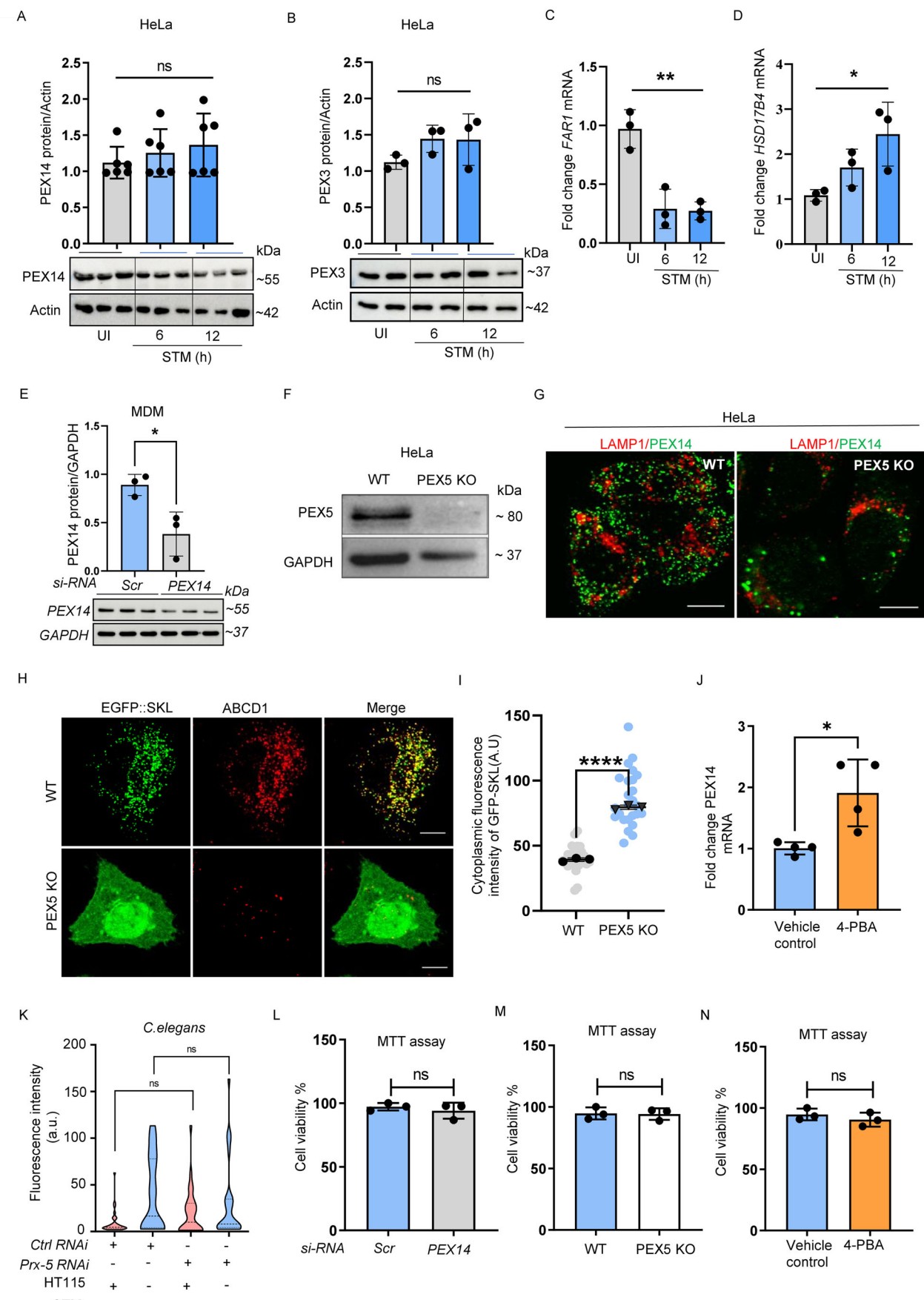

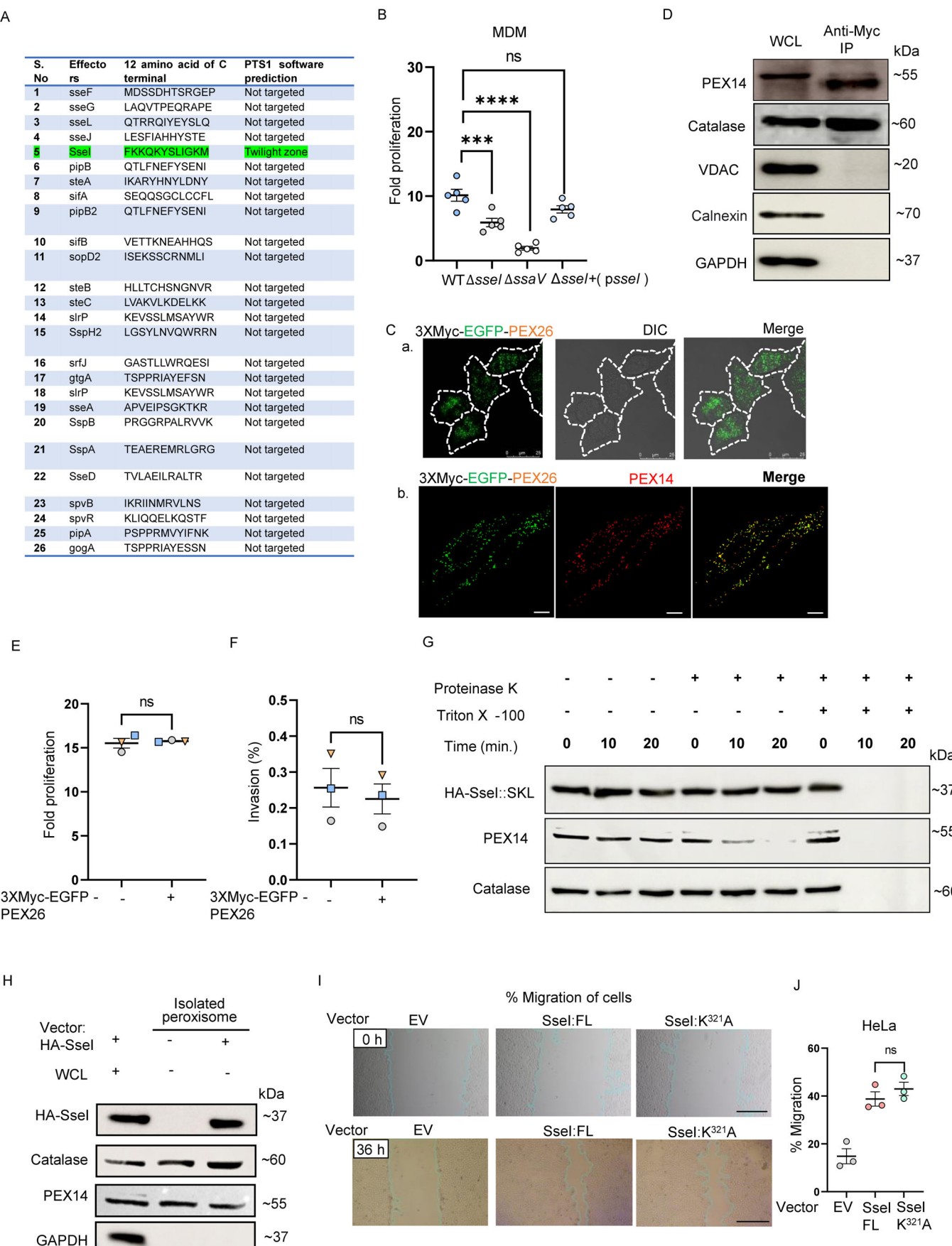

**Figure EV2. Validation of PEROXO-tagged stable cell line expressing 3X myc-EGFP-PEX26.**

(A) Table listing key STM SPI2 effector proteins whose C-terminal amino acids (12 residues) were analyzed using in silico "PTS1 predictor" software to identify STM effector proteins containing the putative PTS1 motif. SseI is highlighted in green. (B) Fold proliferation of WT STM, Δ*ssaV* STM, Δ*sseI* STM, and Δ*sseI* STM complemented with pQE60-SseI in MDM cells. Each dot represents one donor; data are shown for five donors. (C) Fluorescence microscopy images validating the generation of a PEROXO-tagged stable cell line obtained by transfecting HeLa cells with lentivirus overexpressing 3xMyc-EGFP-PEX26 (a). Cells were stained with anti-PEX14 antibody to confirm the peroxisomal localization of the PEROXO tag green (3xMyc-EGFP-PEX26) (b). Scale bars: 25 μm (a), 10 μm (b). (D) Immunoblot analysis showing the purity of isolated peroxisomes using different organelle markers. PEX14, catalase, VDAC, calnexin, and GAPDH were used as markers for peroxisomes, mitochondria, ER, and cytosol, respectively. (E) Graph representing changes in fold proliferation of STM in PEROXO-tag (3xMyc-EGFP-PEX26) expressing stable HeLa cells. Data represent the mean ± SEM of three independent experiments. (F) Graph representing the percentage invasion of STM in PEROXO-tag (3xMyc-EGFP-PEX26) expressing stable HeLa cells. Data represent the mean ± SEM of three independent experiments. (G) Immunoblot analysis of HA-SseI::SKL protein after protease protection assay on isolated peroxisomes. Catalase and PEX14 were used as controls for matrix and membrane proteins, respectively. (H) Immunoblot analysis of HeLa cells were either transfected with SseI-HA or left untransfected. Whole cell lysate (WCL) and isolated peroxisome protein samples were subjected to immunoblot analysis using antibodies against HA, catalase, PEX14, and GAPDH. (I) Light microscopy images of HeLa cells during wound healing assay. HeLa cells were transfected with either full-length or K$^{321}$A mutant SseI constructs. Scale bar: 10 μm. (J) Graph representing the percentage migration of HeLa cells after transfected with either full-length (FL) or K$^{321}$A mutant of SseI constructs. Data information: Data were analyzed using one-way ANOVA, (B) (***$p = 0.0009$, ****$p = 0.0001$) and (J) 'ns' denotes no significant difference. (E, F) Data were analyzed using student's t-test, 'ns' denote non-significant.

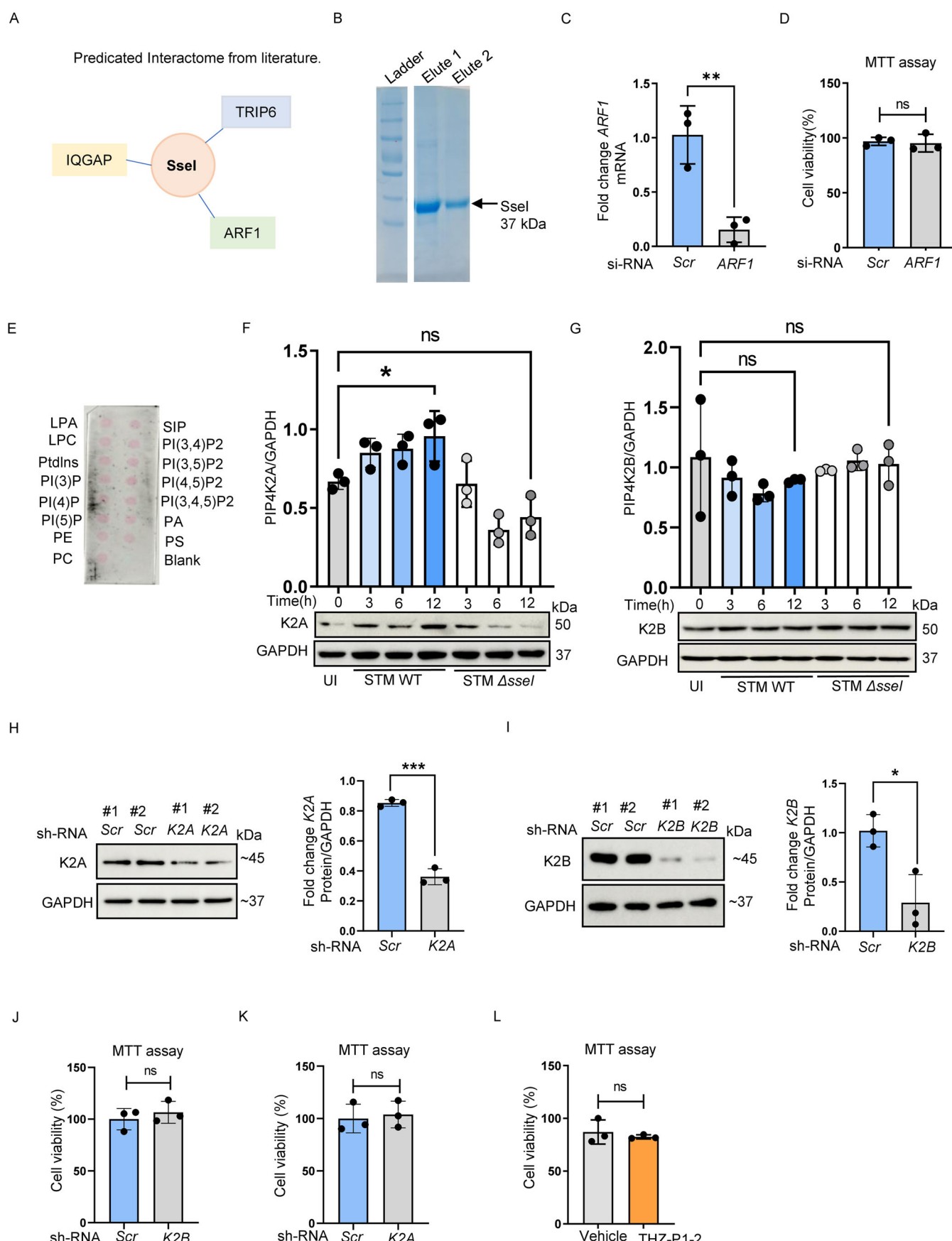

◀  **Figure EV3.  SseI interacts with ARF1 to regulate its activity on the peroxisome membrane.**

(**A**) Interactome of SseI with host proteins. Data retrieved from published literature. (**B**) Protein gel stained with Coomassie brilliant blue indicating purified His-tag SseI (~37 kDa) isolated from BL21 strain of *E. coli*. (**C**) Graph representing the silencing efficiency of *ARF1* in HeLa cells was validated using qPCR. Graph represent the ± SEM of three biological replicates. (**D**) Graph representing percentage cell viability of HeLa cells after silencing *ARF1* measured by MTT assay. Graph represent the ± SEM of four biological replicates. (**E**) PIP assay strip indicating protein-lipid overlay to identify the interaction of SseI with various indicated membrane lipids. Purified SseI protein was incubated on a lipid-coated membrane. A scheme of the PIP-strip membrane is presented. The red line highlights the phospholipid species tested. (**F**) Immunoblot and densitometry analysis of PIP4K2A protein levels after wild type (WT) or ΔsseI STM infection (MOI = 10) for 3, 6, and 12 h. 'UI' denotes uninfected cells. Densitometry analysis is from three biological replicates. (**G**) Immunoblot and densitometry analysis of PIP4K2B protein levels after wild type (WT) or ΔsseI STM infection (MOI = 10) for 3, 6, and 12 h. 'UI' denotes uninfected cells. Densitometry analysis is from three biological replicates. (**H**) Immunoblot analysis of PIP4K2A protein levels after silencing *PIP4K2A* in HeLa cells. GAPDH is used as the loading control. Graph represent the ±SEM of three biological replicates. (**I**) Immunoblot analysis of PIP4K2B protein levels after silencing *PIP4K2B* in HeLa cells. GAPDH is used as the loading control. Graph represent the ± SEM of three biological replicates. (**J, K**) Graph representing percentage cell viability of HeLa cells after silencing *PIP4K2A* (**J**) and *PIP4K2B* (**K**) in HeLa cells measured by MTT assay. Graph represent the ± SEM of three biological replicates. (**L**) Graph representing percentage cell viability of HeLa cells after treatment with THZ-P1-2 (1 μM) measured by MTT assay. Graph represent the ± SEM of three biological replicates. Data information: Data were analyzed using one-way ANOVA, (**F**) (*$p = 0.0274$), and (**G**) 'ns' denotes no significant difference. Data were analyzed using student's t-test, (**C**) (**$p = 0.0065$), (**D, H**) (***$p = 0.0001$), (**I**) (*$P = 0.0184$), (**J–L**), 'ns' denotes non-significant.

                                                              

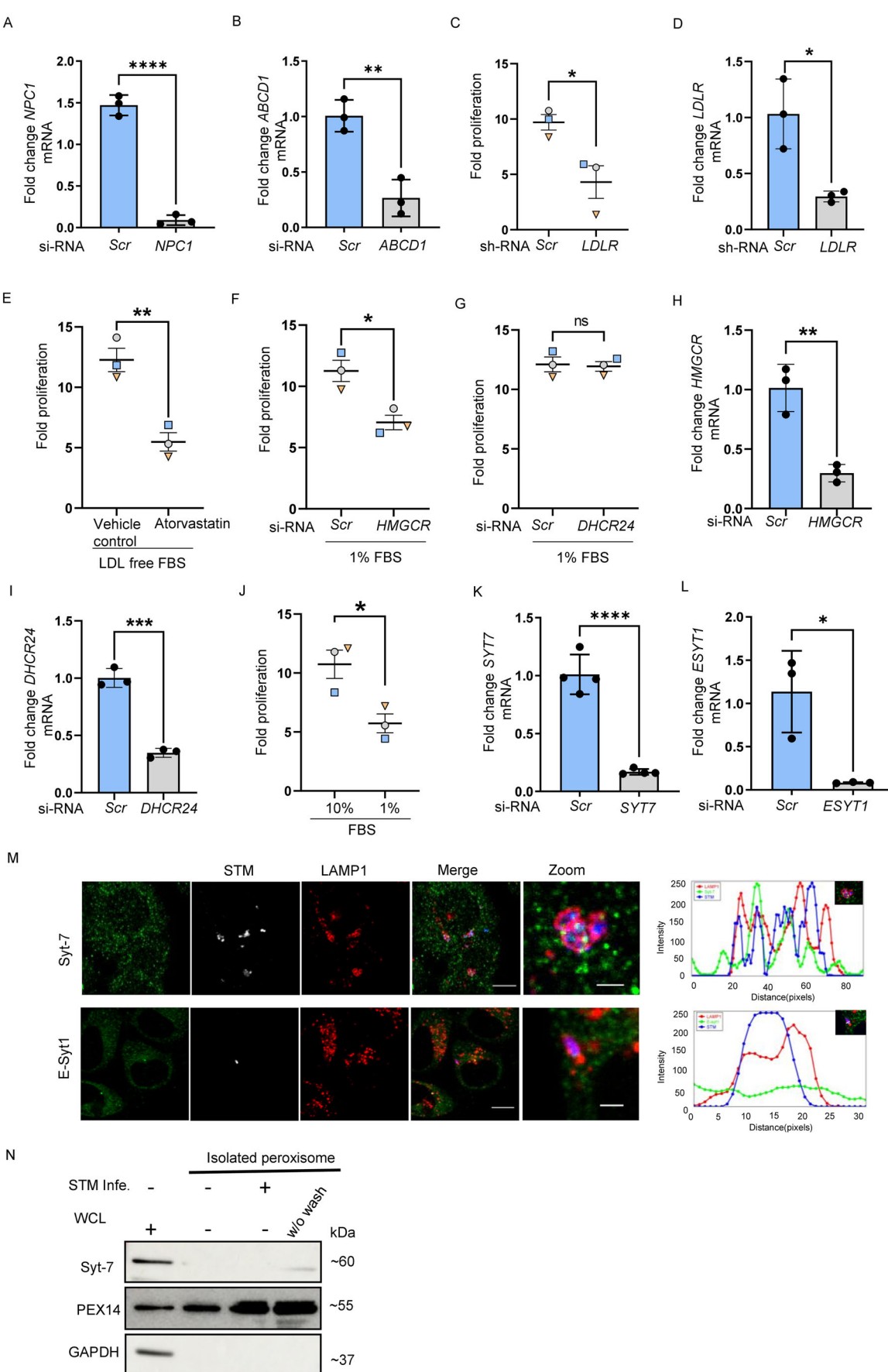

◄ **Figure EV4. Syt7 on SCV tethers PIP2 on peroxisome to facilitate cholesterol transfer.**

(A) Fold change in mRNA levels of *NPC1* after its silencing in HeLa cells. Graph represent the ± SEM of three biological replicates. (B) Fold change in mRNA levels of *ABCD1* after its silencing in HeLa cells. Graph represent the ± SEM of three biological replicates. (C) Graph representing the changes in fold proliferation of STM (MOI = 10) after silencing *LDLR* in HeLa cells. Three independent experiments were performed. Each dot represents the mean of an independent experiment. (D) Fold change in mRNA levels of *LDLR* after its silencing in HeLa cells. Graph represent the ± SEM of three biological replicates. (E) Fold proliferation of STM (MOI = 10) in HeLa cells under LDL-free FBS conditions following Atorvastatin (1 μM). HeLa cells were treated with Atorvastatin and DMSO as Vehicle control. Three independent experiments were performed. Each dot represents the mean of an independent experiment. (F) Fold proliferation of STM (MOI = 10) in HeLa cells under 1% FBS conditions following HMGCR silencing. HeLa cells were transfected with siRNA targeting *HMGCR* or scrambled siRNA (Scr) control. Three independent experiments were performed. Each dot represents the mean of an independent experiment. (G) Graph representing the changes in fold proliferation of STM (MOI = 10) after silencing *DHCR24* in HeLa cells at 1% FBS condition. Three independent experiments were performed. Each dot represents the mean of an independent experiment. (H) Fold change in mRNA levels of *HMGCR* after its silencing in HeLa cells. Graph represent the ± SEM of three biological replicates. (I) Fold change in mRNA levels of *DHCR24* after its silencing in HeLa cells. Graph represent the ± SEM of three biological replicates. (J) Graph representing the changes in intracellular fold proliferation of STM (MOI = 10) in HeLa cells at 10% and 1% FBS condition. Three independent experiments were performed. Each dot represents the mean of an independent experiment. (K) Fold change in mRNA levels of *SYT7* after its silencing in HeLa cells. Graph represent the ± SEM of three biological replicates. (L) Fold change in mRNA levels of *ESYT1* after its silencing in HeLa cells. Graph represent the ± SEM of three biological replicates. (M) Representative microscopy images of HeLa cells infected with GFP-tagged STM (blue) (MOI = 50) and immunostained for endogenous Syt-7/E-syt1 (green) and LAMP1 (red) antibody. The intensity profile indicating the overlap of signals of Syt-7-LAMP1 and E-syt1-LAMP1 is shown on the right. Scale bars: 10 μm, 2 μm. (N) Immunoblot Analysis of Syt-7 Localization on Peroxisomes in HeLa cells. Cells were either infected with STM or left uninfected. Whole-cell lysates (WCL) and isolated peroxisome protein samples (both washed and unwashed) were subjected to immunoblot analysis using antibodies against Syt-7, the peroxisomal marker PEX14, and GAPDH as a loading control. Data information: (A–F) Data were analyzed using student's t-test; (A) (****$p$ = 0.0001), (B) (**$p$ = 0.0043), (C) (*$p$ = 0.0297), (D) (*$p$ = 0.0156), (E) (**$p$ = 0.0053), (F) (*$p$ = 0.0160), (G, H) (**$p$ = 0.0043), (I) (***$p$ = 0.0002), (J) (*$p$ = 0.0253), and (K) (****$p$ = 0.0001), (L) (*$p$ = 0.0182), 'ns' denotes non-significant.

