## [Peer Review File · EMBO Reports]

***Salmonella* Typhimurium effector Ssel regulates host peroxisomal dynamics to acquire lysosomal cholesterol**

DESH RAJ, Abhilash Nair, ANMOL SINGH, SWARNALI BASU, KABITA SARKAR, Jyotsna Sharma, SHIVA SHARMA, SANMI SHARMA, Manisha Rathore, SHRIYA SINGH, SHAKTI PRAKASH, Simran ., SHIKHA SAHU, Aman Kaushik, Mohammad Siddiqi, UDAY GHOSHAL, TULIKA CHANDRA, VIVEK BHOSALE, Arunava Dasgupta, Shashi Gupta, SONIA VERMA, RAJDEEP GUHA, Dipshikha Chakravorty, VEENA AMMANATHAN, and AMIT LAHIRI

Corresponding author(s): AMIT LAHIRI (amit.lahiri@cdri.res.in) , Dipshikha Chakravorty (dipa@iisc.ac.in), VEENA AMMANATHAN (veena.pdf@cdri.res.in)

Review Timeline:

Submission Date:	4th Mar 24
Editorial Decision:	22nd Apr 24
Revision Received:	19th Aug 24
Editorial Decision:	9th Oct 24
Revision Received:	16th Oct 24
Accepted:	25th Oct 24

Transaction Report:

Dear Mr. LAHIRI

Thank you for the submission of your research manuscript to our journal. We have now received the full set of referee reports that is copied below.

As you will see, the referees acknowledge that the findings are interesting and that the conclusions are overall supported by the data presented but they also raise a number of concerns and have suggestions how to further strengthen the data, all of which need to be addressed.

Given these constructive comments, we would like to invite you to revise your manuscript with the understanding that the referee concerns (as detailed above and in their reports) must be fully addressed and their suggestions taken on board. Please address all referee concerns in a complete point-by-point response. Acceptance of the manuscript will depend on a positive outcome of a second round of review. It is EMBO Reports policy to allow a single round of revision only and acceptance or rejection of the manuscript will therefore depend on the completeness of your responses included in the next, final version of the manuscript.

We realize that it is difficult to revise to a specific deadline. In the interest of protecting the conceptual advance provided by the work, we recommend a revision within 3 months (July 22nd). Please discuss the revision progress ahead of this time with the editor if you require more time to complete the revisions.

I am also happy to discuss the revision further via e-mail or a video call, if you wish.

*****IMPORTANT NOTE:

We perform an initial quality control of all revised manuscripts before re-review. Your manuscript will FAIL this control and the handling will be delayed IN CASE the following APPLIES:

- 1) A data availability section providing access to data deposited in public databases is missing. If you have not deposited any data, please add a sentence to the data availability section that explains that.
- 2) Your manuscript contains statistics and error bars based on $n=2$. Please use scatter blots in these cases. No statistics should be calculated if $n=2$.

When submitting your revised manuscript, please carefully review the instructions that follow below. Failure to include requested items will delay the evaluation of your revision.*****

- 1) a .docx formatted version of the manuscript text (including legends for main figures, EV figures and tables). Please make sure that the changes are highlighted to be clearly visible.
- 2) individual production quality figure files as .eps, .tif, .jpg (one file per figure). Please download our Figure Preparation Guidelines (figure preparation pdf) from our Author Guidelines pages <https://www.embopress.org/page/journal/14693178/authorguide> for more info on how to prepare your figures.
- 3) a .docx formatted letter INCLUDING the reviewers' reports and your detailed point-by-point responses to their comments. As part of the EMBO Press transparent editorial process, the point-by-point response is part of the Review Process File (RPF), which will be published alongside your paper.
- 4) a complete author checklist, which you can download from our author guidelines (<<https://www.embopress.org/page/journal/14693178/authorguide>>). Please insert information in the checklist that is also reflected in the manuscript. The completed author checklist will also be part of the RPF.
- 5) Please note that all corresponding authors are required to supply an ORCID ID for their name upon submission of a revised manuscript (<<https://orcid.org/>>). Please find instructions on how to link your ORCID ID to your account in our manuscript tracking system in our Author guidelines (<<https://www.embopress.org/page/journal/14693178/authorguide#authorshipguidelines>>)
- 6) We replaced Supplementary Information with Expanded View (EV) Figures and Tables that are collapsible/expandable online. A maximum of 5 EV Figures can be typeset. EV Figures should be cited as "Figure EV1, Figure EV2" etc... in the text and their respective legends should be included in the main text after the legends of regular figures.

7) Before submitting your revision, primary datasets (and computer code, where appropriate) produced in this study need to be deposited in an appropriate public database (see <<https://www.embopress.org/page/journal/14693178/authorguide#dataavailability>>).

Specifically, we would kindly ask you to provide public access to the mass spectrometry dataset.

The accession numbers and database should be listed in a formal "Data Availability " section (placed after Materials & Method) that follows the model below (see also <<https://www.embopress.org/page/journal/14693178/authorguide#dataavailability>>). Please note that the Data Availability Section is restricted to new primary data that are part of this study.

Data availability

Additional information on source data and instruction on how to label the files are available <<https://www.embopress.org/page/journal/14693178/authorguide#sourcedata>>.

10) Figure legends and data quantification:

- the name of the statistical test used to generate error bars and P values,
- the number (n) of independent experiments (please specify technical or biological replicates) underlying each data point,
- the nature of the bars and error bars (s.d., s.e.m.)
- If the data are obtained from n {less than or equal to} 5, show the individual data points in addition to the SD or SEM.
- If the data are obtained from n {less than or equal to} 2, use scatter blots showing the individual data points.

11) Our journal encourages inclusion of *data citations in the reference list* to directly cite datasets that were re-used and obtained from public databases. Data citations in the article text are distinct from normal bibliographical citations and should directly link to the database records from which the data can be accessed. In the main text, data citations are formatted as follows: "Data ref: Smith et al, 2001" or "Data ref: NCBI Sequence Read Archive PRJNA342805, 2017". In the Reference list,

data citations must be labeled with "[DATASET]". A data reference must provide the database name, accession number/identifiers and a resolvable link to the landing page from which the data can be accessed at the end of the reference. Further instructions are available at <<https://www.embopress.org/page/journal/14693178/authorguide#referencesformat>>.

12) All Materials and Methods need to be described in the main text. We would encourage you to use 'Structured Methods', our new Methods format. According to this format, the Methods section should include a Reagents and Tools Table (listing key reagents, experimental models, software and relevant equipment and including their sources and relevant identifiers) followed by a Methods and Protocols section in which we encourage the authors to describe their methods using a step-by-step protocol format with bullet points, to facilitate the adoption of the methodologies across labs. More information on how to adhere to this format as well as downloadable templates (.doc or .xls) for the Reagents and Tools Table can be found in our author guidelines: <<https://www.embopress.org/page/journal/14693178/authorguide#manuscriptpreparation>>.

An example of a Method paper with Structured Methods can be found here:
<<https://www.embopress.org/doi/10.15252/msb.20178071>>.

13) As part of the EMBO publication's Transparent Editorial Process, EMBO Reports publishes online a Review Process File to accompany accepted manuscripts. This File will be published in conjunction with your paper and will include the referee reports, your point-by-point response and all pertinent correspondence relating to the manuscript.

Yours sincerely,

Referee #1:

This interesting paper by Raj et al., studies the role of an effector protein, Sse1, from Salmonella in enhancing the interaction of mammalian peroxisomes with lysosomes, the contacts between peroxisomes and Salmonella-containing vacuoles (SCVs) and the transport of cholesterol from lysosomes to SCVs, via peroxisomes. This transfer of host cholesterol enhances SCV stability, Salmonella-induced (actin) filaments (SIF) integrity and intracellular bacterial growth. This effect was found in human epithelial and primary macrophage lines.

The authors elucidate mechanistically how this happens. They show that peroxisomes are recruited to SCVs and function as pro-bacterial organelle. This recruitment is facilitated by a putative peroxisome targeting signal (PTS1) in Sse1, which binds the PTS1 receptor, PEX5, but does not significantly enter peroxisomes. This is important for its mode of action. Sse1 on peroxisomes activates a host Ras GTPase, ADP-ribosylation factor-1, (ARF-1). Activation of ARF-1 leads to the recruitment of phosphatidylinositol-5-phosphate-4 kinase to generate PIP2 on peroxisomes, which binds to previously characterized lysosomal synaptotagmin, Syt7. Overall, this is an excellent paper showing how bacterial effectors usurp the host machinery and metabolites for their own survival.

My first suggestion is that since the PTS1 on Sse1 binds the PTS1-receptor, PEX5, but does not enter peroxisomes significantly, the paper should call Sse1 a PEX5-interactor with a putative PTS1-like sequence, rather than a PTS1. A PTS1 has two functions - one to bind PEX5 and the other to transport proteins into peroxisomes. Sse1 has the former, but not the latter function. I find the negligible amount of Sse1 import into peroxisomes (Fig. 3K) irrelevant and unconvincing. Alternatively, they should drive Sse1 completely inside peroxisomes with a strong PTS1 and see if that affects infectivity - my prediction is that it will not have infectivity, which will show where the site of action of Sse1 is.

The observation that PEX14 was not recruited to the PEX5 KO cells, confirming dysfunctional peroxisomes (Figure S2E-F) is unexpected, because PEX14 should be recruited to peroxisomes lacking PEX5. Loss of import of a peroxisomal matrix marker like GFP-PTS1 is a better indicator of dysfunctional peroxisomes in the PEX5-KO cells.

If the location of Sse1 on peroxisomes, via PEX5, is important, does Sse1 inhibit the import of other PTS1 proteins into peroxisomes?

Do the authors know how Sse1 activates ARF1. Presumably this requires PEX5, but is the peroxisome membrane required?

Referee #2:

In this manuscript, Raj et al. show that peroxisomes are recruited to Salmonella-containing vacuoles (SCVs) and function as pro-bacterial organelle. Interestingly, peroxisomal metabolism does not seem to be necessary for the pro-bacterial function, but peroxisomes serve as platform to transfer LDL-derived cholesterol from lysosomes to SCVs. The bacterial effector protein Ssel contains a putative peroxisome targeting sequence 1 (PTS1) motif that mediates the interaction of peroxisomes with SCVs. Ssel interacts with PEX5, the import receptor for PTS1-containing peroxisomal matrix proteins. However, despite its interaction with PEX5, Ssel is predominantly localized on the membrane of peroxisomes and not in the peroxisomal matrix.

Recently, studies by Bao-Liang Song's group have shown that peroxisomes form dynamic membrane contacts with lysosomes to transport LDL-derived cholesterol from lysosomes to peroxisomes. The membrane contacts are formed between the lysosomal protein synaptotagmin VII (SYT7) and the peroxisomal lipid phosphatidylinositol 4,5-bisphosphate [PI(4,5)P₂ or PIP₂]. Now, in this study the authors show that Ssel activates a host GTPase, namely ARF1, to increase PIP₂ levels on peroxisomes, leading to the interaction of peroxisomes with SCVs using SYT7 as tethering protein on SCVs. Hence, peroxisomes serve as a bridge to transfer lysosomal cholesterol to SCVs.

Overall, this is an important study and the data appear to be of sufficient quality. However, there are some concerns in this study, they are:

Movie S1A: No interaction between peroxisomes and lysosomes visible, generally low staining with LysoTracker

Figure 2A: The authors performed a proteomics analysis to determine whether peroxisome abundance was altered after 6 and 12 h of STM infection. Please explain in detail how the heatmap was generated and what is shown. Obviously, the fold changes are not shown, because then there would be no column for the uninfected cells.

Based on the heatmap, I would expect levels of proteins such as ACBD5, PEX5, PPARA to be significantly altered. Why were the mRNA levels of peroxisomal enzymes whose protein levels were significantly altered in infected cells not determined (FAR1 and AGPS involved in plasmalogen synthesis; ACAA1 and HSD17B4 involved in fatty acid oxidation)?

Figure 2B-D: Do blue dots represent technical replicates of the independent experiments? The authors analyzed the expression of the peroxins PEX11B, PEX14, and PEX19. Whereas the authors claim that PEX11B protein levels are significantly altered in infected cells, mRNA levels are similar in uninfected and infected cells. Why were the mRNA levels of PEX14 and PEX19 analyzed, since PEX14 protein levels were similar in uninfected and infected cells and PEX19 was not detected by proteomics?

Figure 2E: Three independent experiments were performed for catalase activity, but triplicates of one experiment are shown. Are these technical replicates of a lysate or measurements of three lysates?

Figure 2G, S2C: PEX14 has been silenced in human primary macrophages, and reduced bacterial proliferation was observed. PEX14 mRNA levels were only reduced by 50%. It's necessary to show whether PEX14 is reduced at the protein level and if these reduction leads to a decrease in peroxisome abundance. The authors should also demonstrate whether silencing PEX14 impairs the import of peroxisomal matrix proteins.

Figure S2D, S2E, S2F: A PEX5 knockout HeLa cell line has been generated. The authors state that PEX14 was not recruited to the PEX5 KO cells. However, immunofluorescence staining for PEX14 clearly shows a punctate pattern (Figure S2E). The number of puncta is decreased in the KO cells compared to WT cells. Furthermore, the puncta are bigger in the KO cells compared to WT cells. This is typical for peroxisomal membrane ghosts in PEX5-deficient cells, which are dysfunctional in the import of peroxisomal matrix proteins. Staining for catalase or another peroxisomal matrix protein would confirm that these ghosts cannot import peroxisomal matrix proteins.

Figure 2I, S2I: HeLa cells have been treated with 4-PBA, an inducer of peroxisome biogenesis. While 4-PBA leads to a strong proliferation of peroxisomes in rodents, the effect in human cells is often not as pronounced. An increase in the mRNA level of

PEX14 is not necessarily a measure of an increased number of peroxisomes. The authors should show by immunofluorescence and Western blot that the peroxisome number is really elevated.

Figure 2J: Is the fluorescence intensity of HT115-infected control cells and Prx5-knockdown cells significantly different?

Using the in silico software "PTS1 predictor" the authors queried Salmonella T3SS effector proteins to identify those that contain a putative peroxisomal targeting signal 1. The effector Ssel might contain a variant of PTS1, namely "GKM".

Figure 3B, S3A: To investigate if the C-terminal tripeptide GKM might be functional as PTS1 signal for the import into the peroxisomal matrix, the authors should show whether a fluorescent protein (e.g., GFP, mCherry) containing the putative C-terminal PTS1 motif [GKM and/or the 12 C-terminal amino acids (important if the putative PTS1 is not conserved)] colocalizes with peroxisomes.

Figure 3E, F: The authors should also show whether the complementation of Ssel lacking the putative PTS1 motif also restores the growth defect of the Δ ssel strain.

Figure 3G-I: The authors state that Ssel is localized "on peroxisomes". However, I would understand this to mean that Ssel is localized on or in the peroxisomal membrane and not in the matrix. Consider changing the wording.

Figure legend for 4J: The caption 'UT' denotes untreated cells. However, nothing is marked with 'UT' in panel J.

Figure 5A: In Figure 1 LAMP1 was used as marker for SCVs, and in Figure 5A LAMP1 is used as marker for lysosomes to determine the % of peroxisome-lysosome colocalization. Representative images should be shown. How much overlap is there between lysosomes and SCVs?

Figure 5C: Filipin was used to stain cholesterol and determine the colocalization with STM and LAMP1? Filipin is known to bleach easily and fast, therefore, I wonder how the authors coped with the bleaching to obtain such bright filipin staining.

Figure 5G-H: Inhibition of de novo cholesterol synthesis with lovastatin did not affect the rate of bacterial replication, whereas the NPC1 inhibitor U18666A, which sequesters cholesterol in lysosomes, reduced bacterial replication. How was the experiment with lovastatin conducted? Which medium was used for the experiment with lovastatin? De novo cholesterol synthesis is very low in cells cultured in medium with FBS, and the effect of lovastatin would be negligible because the cells would cover their cholesterol requirements through LDL uptake. Activation of the SREBP-2 pathway increases the expression of the LDL receptor, which would increase LDL-receptor-mediated cholesterol uptake. Is bacterial replication impaired when cells are cultured in lipoprotein-deficient serum? Does bacterial replication differ when the cells are cultivated in medium with 10 and 1% FBS? How much LDL was added to the medium containing 1% FBS?

Line 336: the authors state that none of the inhibitors altered cell viability, however, in two experiments cell viability was decreased more than 50% in lovastatin-treated cells. Was the treatment with lovastatin carried out in medium with 10% FBS? How long were the cells treated with lovastatin?

Figure 5I: Chu et al. (Cell 2015) showed that lysosomal Syt7 binds peroxisomal PIP2 to bridge the organelle contact. What is the percentage of colocalization of peroxisomes with Syt7 in STM-infected cells?

Figure 6 or line 362: Please define the abbreviation MLN when it is used for the first time.

Figure 6E-F: While complementation with Ssel tagged with myc at its C-terminus is lower than complementation with Ssel tagged with myc at its N-terminus, complementation with Ssel-myc obviously occurs (even if it is not statistically significant). How can this be explained? I would suggest performing the complementation experiment with Ssel- Δ GKM and Ssel-K231A.

Line 87-88: References 38 and 39 do not address cholesterol transfer from peroxisomes to the endoplasmic reticulum. The publication by Xiao et al. (2019) would be the appropriate reference (<https://doi.org/10.1007/s11427-019-9569-9>).

Materials and methods:

The authors state that for antibody staining, permeabilization buffer (Triton X-100 (0.3 %) /1% BSA in 1X-PBS) containing primary antibody dilution was used according to the manufacturer's instructions. Which manufacturer?

Line 670: it should be Bio-Rad instead of Bio-Red

Referee #3:

In this manuscript, Raj et al attempted to address the question of how Salmonella acquires the cholesterol to establish the SCV

during infection. They found a TTSS SPI2 effector Ssel plays an important role in Salmonella intracellular growth and acquisition of cholesterol onto SCVs through the binding to host proteins ARF1 and PEX5. In addition, the authors demonstrate the acquisition of cholesterol in SCV requires the peroxisome-lysosome interaction mediated by synaptotagmin-7. The manuscript is well written and the authors try to clarify the relevance of the interaction between SCV, peroxisome, and lysosome from different angles. However, I feel that the authors may need to add some data to elucidate how ARF1, PEX5, and Ssel coordinately work to recruit the cholesterol for the establishment of SCV during infection.

Major comments

1. In figure 4B, Ssel was found to interact with ARF1 and PEX5. I am wondering if the GKM signal at C-term of Ssel is required for the interaction with ARF1 or PEX5? The authors should try to validate this point using HA-tagged Ssel mutants.
2. The data in Figure 4D doesn't demonstrate that ARF1 is activated dependently on Salmonella infection. Similarly, ARF1 seems to be recruited to peroxisome regardless of Salmonella infection in figure 1. The author should approach this kind of experiment using a tool/kit or mutant to detect specifically active form ARF1. In contrast, it would be interesting if the author could see that inactive form ARF1 is not included on isolated peroxisomes.
3. I am curious whether Synaptotagmin-7 can be detected on peroxisomes in a Salmonella infection- or Ssel-dependent manner.
4. To support data in Figure 4I, the author may try experiments, which are similar to Figure 4J-L, using PI5P4K2B inhibitor.
5. As for the result summary described as a model in supplementary figure 6, it well represents the dynamics of LDL-derived cholesterol compared between a WT host cell vs a PEX5 KO cell over the salmonella infection. However, this model doesn't reflect the manuscript title. I feel the author should try to describe the dynamics of LDL-derived cholesterol, peroxisome-SCV-lysosome interactions, and the roles of ARF and PEX5 in a host cell infected with WT STM or Δ sseI mutant (meaning in the presence or absence of Ssel) because the biological significance of Salmonella effector Ssel during infection is not well pictured in the current figure. It would be better if the GKM signal at C-terminus of Ssel is also described.
6. In Figure 5C, HeLa cells as hosts is labeled as "WT". I think it is better to indicate "HeLa" instead of "WT", and PEX5 KO may be replaced with HeLa PEX5 KO. The authors describe the figure legend properly, but I took a while to recognize that "WT" described in the images means HeLa because I thought it was WT STM. In addition, Figure 5D also seems to require the correction as well. The author should indicate which host cell is infected by which bacterial strain so that readers can avoid misreading the data.

Minor comments

1. In supplementary figures 5A and 5C, NCP1 should be corrected as NPC1.
2. The labels of the Y axis of a graph, which represents the fold change of Salmonella proliferation in the presence of ABCD1, are too small.
3. Line 362: Please place the abbreviation of MLN, but not in Line 707.
4. It is difficult for me to recognize the "Merge" described using mixed-colored fonts. I think it's better to use black for Merge (Fig1F, Fig2K, Fig5I, and Supp.Fig3B).

'Salmonella Typhimurium effector SseI regulates peroxisomal dynamics to acquire lysosomal cholesterol for better intracellular growth'(EMBOR-2024-59092V1)

Dear Mr. LAHIRI

Thank you for the submission of your research manuscript to our journal. We have now received the full set of referee reports that is copied below.

As you will see, the referees acknowledge that the findings are interesting and that the conclusions are overall supported by the data presented but they also raise a number of concerns and have suggestions how to further strengthen the data, all of which need to be addressed.

Given these constructive comments, we would like to invite you to revise your manuscript with the understanding that the referee concerns (as detailed above and in their reports) must be fully addressed and their suggestions taken on board. Please address all referee concerns in a complete point-by-point response. Acceptance of the manuscript will depend on a positive outcome of a second round of review. It is EMBO Reports policy to allow a single round of revision only and acceptance or rejection of the manuscript will therefore depend on the completeness of your responses included in the next, final version of the manuscript.

We realize that it is difficult to revise to a specific deadline. In the interest of protecting the conceptual advance provided by the work, we recommend a revision within 3 months (July 22nd). Please discuss the revision progress ahead of this time with the editor if you require more time to complete the revisions.

I am also happy to discuss the revision further via e-mail or a video call, if you wish.

We sincerely thank the Editorial board members and reviewers for giving us valuable feedback and critical review. We have now responded to the reviewers' comments and made appropriate changes to the manuscript. Necessary efforts are made to precisely address every comment from the reviewers.

We also thank the Editor for continued support and clarifying our concerns during the review process.

Referee #1:

This interesting paper by Raj et al., studies the role of an effector protein, Sse1, from Salmonella in enhancing the interaction of mammalian peroxisomes with lysosomes, the contacts between peroxisomes and Salmonella-containing vacuoles (SCVs) and the transport of cholesterol from lysosomes to SCVs, via peroxisomes. This transfer of host cholesterol enhances SCV stability, Salmonella-induced (actin) filaments (SIF) integrity and intracellular bacterial growth. This effect was found in human epithelial and primary macrophage lines.

The authors elucidate mechanistically how this happens. They show that peroxisomes are recruited to SCVs and function as pro-bacterial organelle. This recruitment is facilitated by a putative peroxisome targeting signal (PTS1) in Sse1, which binds the PTS1 receptor, PEX5, but does not significantly enter peroxisomes. This is important for its mode of action. Sse1 on peroxisomes activates a host Ras GTPase, ADP-ribosylation factor-1, (ARF-1). Activation of ARF-1 leads to the recruitment of phosphatidylinositol-5-phosphate-4 kinase to generate PIP2 on peroxisomes, which binds to previously characterized lysosomal synaptotagmin, Syt7. Overall, this is an excellent paper showing how bacterial effectors usurp the host machinery and metabolites for their own survival.

We thank the reviewer for the valuable comments and suggestions. This has helped us in improving the work significantly. We have addressed the queries and the point-wise response is given below.

My first suggestion is that since the PTS1 on Sse1 binds the PTS1-receptor, PEX5, but does

not enter peroxisomes significantly, the paper should call Sse1 a PEX5-interactor with a putative PTS1-like sequence, rather than a PTS1. A PTS1 has two functions - one to bind PEX5 and the other to transport proteins into peroxisomes. Sse1 has the former, but not the latter function. I find the negligible amount of Sse1 import into peroxisomes (Fig. 3K) irrelevant and unconvincing. Alternatively, they should drive Sse1 completely inside peroxisomes with a strong PTS1 and see if that affects infectivity - my prediction is that it will not have infectivity, which will show where the site of action of Sse1 is.

We thank the reviewer for this critical comment. We have now referred SseI to contain putative PTS1-like sequence rather than PTS1.

We agree with the reviewer that although SseI binds PEX5, it does not efficiently transport the protein to peroxisome matrix. In order to understand the significance of the membrane localization of SseI, we have now tagged SseI to SKL (devoid of GKM) in pQE60 vector to drive SseI into peroxisomes. This is validated by protease protection assay. As seen in the lanes only treated with Proteinase K, presence of HA-SseI:SKL indicates that it is present in the peroxisomal matrix similar to known matrix protein, catalase. **The immunoblot appears as figure EV-2G in the revised manuscript.**

EV- 2G

Figure: EV-2G: Protease protection assay on isolated peroxisomes from HeLa cells expressing HA-SseI-SKL in pQE60 vector.

We observe that plasmid expression of SseI-SKL (pQE60 containing SseI-SKL) does not rescue the grow defect of $\Delta sseI$ strain as predicted by the reviewer. Whereas expressing full length (SseI-FL) does complement the defect. This indicates that targeting SseI to peroxisome membrane is critical. Additionally, we also carried out complementation of SseI lacking GKM (SseI- Δ GKM) or point mutation (SseI-K321A). These strains also fail to rescue the growth defect seen in $\Delta sseI$ strain. Taken together, we show that SseI targeting to peroxisomes by GKM motif is critical for its virulence.

3L

Figure 3L: Change in fold proliferation of the WT STM, $\Delta sseI$ STM, and $\Delta sseI$ STM complemented with SseI full length or its mutants by pQE60 plasmid expression (MOI=10) in HeLa cells. CFU plotted as fold change after 16 h of infection compared to 2 h. Three independent experiments containing biological triplicates per experiment were performed. Each dot represents mean of an independent experiment. Statistical analyses were performed on mean of independent experiments using one-way ANOVA, **** p <0.001, *** p <0.01, n.s. not significant.

This suggestion helped us significantly in understanding that membrane localization of SseI is critical for its downstream effect such as ARF1 activation.

The fold proliferation graph now appears as figure 3L in the revised manuscript.

The observation that PEX14 was not recruited to the PEX5 KO cells, confirming dysfunctional peroxisomes (Figure S2E-F) is unexpected, because PEX14 should be recruited to peroxisomes lacking PEX5. Loss of import of a peroxisomal matrix marker like GFP-PTS1 is a better indicator of dysfunctional peroxisomes in the PEX5-KO cells.

As suggested by the reviewer, we have now used a peroxisomal matrix marker like EGFP-SKL (in pcDNA3.1 vector) to test the import efficiency of PEX5-KO cells.

As seen in the image below, WT cells show EGFP-SKL colocalizing with peroxisomes (marked by ABCD1). This indicates efficient transport of peroxisomal matrix proteins to peroxisomes. Whereas, PEX5 KO cells show diffused cytoplasmic fluorescence staining indicating defect in import efficiency of SKL. This suggests impaired PTS1-protein targeting to peroxisomes in PEX5-KO cells. Additionally, we also stained for a PTS1 variant-containing peroxisomal enzyme, catalase which reaches peroxisomes by PEX5 receptor. As seen in figure AP-1H, PEX5 KO cells show cytoplasmic fluorescence of catalase indicating import deficiency of the knockout cells.

The microscopy images now appear as extended view EV-1H,I and AP-1H in the revised manuscript.

EV-1H

EV-1I

J

AP-1H

Figure: EV 1H: Immunofluorescence (IF) analysis of peroxisomes (ABCD1 in red) in WT and PEX5 KO HeLa cells expressing EGFP-SKL (in pcDNA3.1 vector).

EV 1I: Quantification of cytoplasmic fluorescence intensity of EGFP-SKL in WT and PEX5 KO HeLa cells.

AP-1H. IF analysis of catalase staining in WT and PEX5 KO HeLa cells.

If the location of Sse1 on peroxisomes, via PEX5, is important, does Sse1 inhibit the import of other PTS1 proteins into peroxisomes?

We thank the reviewer for this important and insightful comment.

In order to answer this query, we over expressed pQE60 vector carrying HA-SseI in HeLa cells and studied the protein levels of catalase on isolated peroxisomes. Catalase contains a variant of PTS1 and is transported to peroxisomal matrix by PEX5 receptor. As seen in the figure below, levels of catalase on isolated peroxisomes do not change after over-expressing HA-SseI. Pex14 was used as the marker for peroxisomes. GAPDH was used as control to show purification of isolated peroxisomes. **The immunoblot now appears as extended view EV-2H in the revised manuscript.**

EV-2H

Figure EV-2H: Immunoblot analysis of peroxisomal proteins after over expressing HA tagged SseI. Catalase is used as marker for peroxisomal matrix protein. GAPDH indicates purity of isolated peroxisomes.

Do the authors know how SseI activates ARF1. Presumably this requires PEX5, but is the peroxisome membrane required?

Our results indicate that SseI binds to both ARF1 and PEX5. Now, our new data show that HA-SseI- Δ GKM binds to ARF1 but not PEX5. This indicates that the binding domain of SseI for ARF1 and PEX5 are different (figure 4B). Hence, it is possible that SseI can activate cytoplasmic pool of ARF1 independently of PEX5 (PEX5 is essential for targeting to peroxisomes). Even if there is cytoplasmic activation of ARF1 by SseI, it does not impact STM proliferation. This is validated by no change in the fold proliferation of Δ sseI STM complemented with SseI- Δ GKM (figure 3L). We therefore propose that activation of ARF1 on the peroxisomal membrane by SseI is essential for its intracellular proliferation.

4B

3L

Figure: 4B. Immunoblot analysis of proteins after pull down of SseI from HeLa cells expressing either HA-tagged SseI or HA-tagged SseI- Δ GKM (in pQE60 vector). The cell lysates (input) were used for the pull-down of SseI using an anti-HA antibody. IgG antibody was used as a control. The membrane was probed with anti-HA, anti-ARF1, and anti-PEX5 antibodies to detect HA-SseI, endogenous ARF1, and PEX5, respectively. The result is representative of three independent experiments.

3L. Change in fold proliferation of the WT STM, Δ sseI STM, and Δ sseI STM complemented with SseI full length or its mutants by plasmid expression (MOI=10) in HeLa cells. CFU plotted as fold change after 16 h of infection compared to 2 h. Three independent experiments containing biological triplicates per experiment were performed. Each dot represents mean of an independent experiment. Statistical analyses were performed on mean of independent experiments using one-way ANOVA, **** p <0.001, ** p <0.01.

To answer the second part of the query- “is the peroxisome membrane required” for ARF1 activation, we have used complementation with SseI-SKL during Δ sseI infection.

In our current study, we hypothesize that the levels of active ARF1 on peroxisomes increase after *Salmonella* infection because SseI gets targeted to peroxisomes. We measured levels of active ARF1 on peroxisomes during the following conditions. **The immunoblot now appears as Figure-4F in the revised manuscript.**

- (i) Uninfected sample
- (ii) Infection with wildtype strain
- (iii) Infection with $\Delta sseI$ *Salmonella* strain
- (iv) Infection with $\Delta sseI$ expressing SseI- Δ GKM by pQE60 plasmid
- (v) Infection with $\Delta sseI$ expressing SseI::SKL by pQE60 plasmid
- (vi) Infection with $\Delta sseI$ expressing full length SseI by pQE60 plasmid

Figure 4F: Immunoblot and densitometry analysis of Active ARF1 protein levels after ARF1-GTP pulldown from isolated peroxisomes of HeLa cells infected with WT or $\Delta sseI$ STM, or $\Delta sseI$ STM complemented with pQE60-SseI clones (Δ GKM, SKL, full length). Densitometry analysis represents three independent experiments.

By active-ARF1 assay, we observe that,

i-ii WT STM induces activation of ARF1 on peroxisomes.

iii. Infection with Δ SseI-STM does not induce ARF1 activation on peroxisomes indicating its significance.

iv. Infection with Δ SseI-STM complemented with pSseI- Δ GKM also do not show activation of ARF1. This suggests that GKM-mediated peroxisomal localization of SseI is essential for ARF1 activation on peroxisomes

v. Infection with Δ SseI-STM complemented with pSseI-SKL where SseI is transported into peroxisomal matrix does not activate peroxisomal ARF1. It is therefore essential to retain SseI on membrane for its ARF1-GEF activity.

vi. Infection with Δ SseI-STM complemented with full length SseI- Δ GKM show activation of ARF1.

This result also reassures the significance of membrane localization of SseI for its downstream effect.

Referee #2:

In this manuscript, Raj et al. show that peroxisomes are recruited to Salmonella-containing vacuoles (SCVs) and function as pro-bacterial organelle. Interestingly, peroxisomal metabolism does not seem to be necessary for the pro-bacterial function, but peroxisomes serve as platform to transfer LDL-derived cholesterol from lysosomes to SCVs. The bacterial effector protein SseI contains a putative peroxisome targeting sequence 1 (PTS1) motif that mediates the interaction of peroxisomes with SCVs. SseI interacts with PEX5, the import receptor for PTS1-containing peroxisomal matrix proteins. However, despite its interaction with PEX5, SseI is predominantly localized on the membrane of peroxisomes and not in the peroxisomal matrix.

Recently, studies by Bao-Liang Song's group have shown that peroxisomes form dynamic membrane contacts with lysosomes to transport LDL-derived cholesterol from lysosomes to peroxisomes. The membrane contacts are formed between the lysosomal protein synaptotagmin VII (SYT7) and the peroxisomal lipid phosphatidylinositol 4,5-bisphosphate [PI(4,5)P2 or PIP2]. Now, in this study the authors show that SseI activates a host GTPase, namely ARF1, to increase PIP2 levels on peroxisomes, leading to the interaction of peroxisomes with SCVs using SYT7 as tethering protein on SCVs. Hence, peroxisomes serve as a bridge to transfer lysosomal cholesterol to SCVs.

Overall, this is an important study and the data appear to be of sufficient quality. However, there are some concerns in this study, they are:

We sincerely thank the reviewer for the constructive comments. This has helped in increasing the depth of the study. We have made necessary efforts to answer the queries and comments. The point-wise response is given below.

Movie S1A: No interaction between peroxisomes and lysosomes visible, generally low staining with LysoTracker

We thank the reviewer for bringing this to our notice. We have now replaced the representative video for live cell imaging with good staining of LysoTracker (Figure 1E). In this replaced video, multiple interactions between peroxisomes and lysosomes as well as SCVs are visible (indicated by white arrowheads in zoom montage)

Below is live cell imaging of HeLa cells stably expressing PEROXO-tag (3xMyc-EGFP-PEX26) infected with mCherry-STM (MOI=50). Cells were also treated with 100 nM of LysoTracker (red) for 15 min before imaging to label acidic vacuoles such as lysosomes and SCVs. Time-lapse imaging was performed 5 h post-infection. Images were captured approximately every minute (80 s) till 15 min. Images shown are single Z slices for the indicated time points—scale bar -10 μ m.

Figure 1E: Live STM temporally interacts with peroxisomes

Live cell imaging of HeLa cells stably expressing 3xMyc-EGFP-PEX26 infected with mCherry-STM (shown here in blue; MOI=50). Cells were also treated with 100 nM of LysoTracker (Red) for 15 min prior to imaging to label acidic vacuoles. Live cell imaging was performed 5 h post-infection. Images were captured every one min for 15 min. White arrow heads indicate regions of interaction. Scale bar-10 μ m.

Figure 2A: The authors performed a proteomics analysis to determine whether peroxisome abundance was altered after 6 and 12 h of STM infection. Please explain in detail how the heatmap was generated and what is shown. Obviously, the fold changes are not shown, because then there would be no column for the uninfected cells.

The heatmap shows the log₂ transformed abundance values for the proteins obtained from mass spectrometry. Shown below are the values of individual replicates.

Figure: Mass spectrometry-based quantification of peroxisome-related proteins after 6 and 12 h of STM infection in HeLa cells. Among the 3412 proteins measured, average of raw abundance value of 31 proteins associated with peroxisome and their changes in comparison with uninfected cells are shown in the heat map. Values of triplicate samples for each time point is shown on right.

For better clarity, we have now replaced the figure 2A (heatmap) representing Z scale transformed values as shown below.

Figure 2A: Z scale transformed values of peroxisome-related proteins identified by mass spectrometry after 6 and 12 h of STM infection in HeLa cells.

Based on the heatmap, I would expect levels of proteins such as ACBD5, PEX5, PPARA to be significantly altered.

Significance was calculated based on ANOVA analysis. Only those proteins that had a p-value less than 0.05 were considered significantly altered.

The results of the ANOVA analysis of the three groups for the proteins (uninfected vs 6 hour vs 12 hour infection) follow.

	ACBD5	PEX5	PPARA
F	3.817	3.919	2.281
p Value	0.0852	0.0815	0.1833
p Value summary	ns	ns	ns

R square	0.5599	0.5664	0.4319
-----------------	--------	--------	--------

Why were the mRNA levels of peroxisomal enzymes whose protein levels were significantly altered in infected cells not determined (FAR1 and AGPS involved in plasmalogen synthesis; ACAA1 and HSD17B4 involved in fatty acid oxidation)?

We thank the reviewer for the suggestion. We have now tested the mRNA levels of significantly altered enzymes such as FAR1, AGPS, ACAA1 and HSD17B4. **The graphs appear as extended view EV-1C, D and AP- 1A, B.**

Figure EV-1C, D and AP-1A, B: Fold change in mRNA levels of peroxisomal genes FAR1 (EV-1C), HSD17B4 (EV-1D), AGPS (AP-1A), ACAA1 (AP-1B) in HeLa cells post 6 and 12 h of STM infection compared to uninfected controls. Data represent the mean \pm SEM of three independent experiments.

Figure 2B-D: Do blue dots represent technical replicates of the independent experiments? The authors analyzed the expression of the peroxins PEX11B, PEX14, and PEX19. Whereas the authors claim that PEX11B protein levels are significantly altered in infected cells, mRNA levels are similar in uninfected and infected cells.

The blue dots represent the biological replicates of independent experiments. We accept that on the contrary to proteomic analysis, mRNA expression levels of PEX11B remained unchanged during infection.

In order to maintain uniformity, we have now plotted the graphs as bar graphs and each dot represents mean of independent experiment as shown below.

Figure 2D: Fold change in mRNA levels of indicated peroxisomal genes in HeLa cells post 6 and 12 h of STM infection compared with uninfected (UI). Statistics were performed on the mean of three independent experiments (black dots).

Why were the mRNA levels of PEX14 and PEX19 analyzed, since PEX14 protein levels were similar in uninfected and infected cells and PEX19 was not detected by proteomics?

We wanted to test if *Salmonella* infection induces peroxisome biogenesis. Therefore, we checked members of peroxisome biogenesis factors (PEX19, PEX3) and protein import machinery (PEX14, PEX5). Most of the candidates tested (PEX5, PEX14, PEX3) remained unchanged in both proteomic and expression analysis. As pointed out by the reviewer, PEX19 was not detected by proteomics.

Figure 2E: Three independent experiments were performed for catalase activity, but triplicates of one experiment are shown. Are these technical replicates of a lysate or measurements of three lysates?

Measurements of three lysates were shown in figure 2E. However, we have now pooled data from three independent lysates each containing two/three biological replicates. **Each dot represents mean of independent experiments.** Source data of every independent experiment with biological replicates are uploaded as per journal guidelines.

Figure 2E is now replaced with the graph below.

2E

Figure 2E. Catalase activity in HeLa cells was measured at indicated time points post-STM infection. Data represent the mean of catalase activity from three independent experiments.

Figure 2G, S2C: PEX14 has been silenced in human primary macrophages, and reduced bacterial proliferation was observed. PEX14 mRNA levels were only reduced by 50%. It's necessary to show whether PEX14 is reduced at the protein level and if these reduction leads to a decrease in peroxisome abundance. The authors should also demonstrate whether silencing PEX14 impairs the import of peroxisomal matrix proteins.

We agree with the reviewer that PEX14 silencing yielded only 50% reduction in mRNA levels. Immunoblotting analysis after PEX14 silencing in human MDM cells also gave us a reduction similar to expression analysis. The figure now appears as EV-1E.

EV-1E

Figure EV 1E: Immunoblotting analysis for silencing efficiency of PEX14 in human MDM cells from three healthy donors.

In order to study the import efficiency of PEX14 silenced cells, we used HeLa cells since the transfection efficiency of EGFP-SKL (in pcDNA3.1 vector) was very low in MDM. We observed that PEX14 silencing led to decrease in peroxisome abundance per cell by labelling peroxisomes with ABCD1. Similarly, silenced cells showed diffused fluorescence staining indicating cytosolic localization of SKL. This suggests impaired PTS1-protein targeting to peroxisomes in siPEX14 cells. **The figure now appears as AP-1D-G in the revised manuscript.**

AP-1D

E

F

G

Figure: AP-1D. Immunoblotting analysis for silencing efficiency of PEX14 in HeLa cells.

AP-1E. Representative microscopy image of PEX14 silenced HeLa cells expressing EGFP-SKL.

AP-1F-G: Quantitation indicates number of peroxisomes/cell after PEX14 silencing and cytoplasmic signal of GFP-SKL.

Figure S2D, S2E, S2F: A PEX5 knockout HeLa cell line has been generated. The authors state that PEX14 was not recruited to the PEX5 KO cells. However, immunofluorescence staining for PEX14 clearly shows a punctate pattern (Figure S2E). The number of puncta is decreased in the KO cells compared to WT cells. Furthermore, the puncta are bigger in the KO cells compared to WT cells. This is typical for peroxisomal membrane ghosts in PEX5-deficient cells, which are dysfunctional in the import of peroxisomal matrix proteins. Staining for catalase or another peroxisomal matrix protein would confirm that these ghosts cannot import peroxisomal matrix proteins.

In order to validate the *PEX5* KO cells, we have now used a peroxisomal matrix marker like EGFP-SKL to test its import efficiency.

As seen in the image below, WT cells show EGFP-SKL (in pcDNA3.1 vector) colocalizing with peroxisomes marked by ABCD1. This indicates efficient transport of peroxisomal matrix proteins to peroxisomes. Whereas, *PEX5* KO cells show diffused fluorescence staining indicating cytosolic localization of SKL. This suggests impaired PTS1-protein targeting to peroxisomes in *PEX5*-KO cells. Additionally, we also stained for a PTS1 variant-containing peroxisomal enzyme, catalase which reaches peroxisomes by *PEX5* receptor. As seen in figure, *PEX5* KO cells show cytoplasmic fluorescence indicating import inefficiency of the knockout cells.

The below microscopy images now appear as extended view EV-1H,I and AP-1H in the revised manuscript.

EV-1H

EV-1I

J

Figure EV-1H: IF analysis of peroxisomes (ABCD1 in red) in WT and PEX5 KO HeLa cells expressing EGFP-SKL (in pcDNA3.1 vector).

EV-1I: The cytoplasmic fluorescence intensity of GFP-SKL is measured between the cell types.

AP-1H: IF analysis of catalase staining in WT and PEX5 KO HeLa cells.

Figure 2I, S2I: HeLa cells have been treated with 4-PBA, an inducer of peroxisome biogenesis. While 4-PBA leads to a strong proliferation of peroxisomes in rodents, the effect in human cells is often not as pronounced. An increase in the mRNA level of PEX14 is not necessarily a measure of an increased number of peroxisomes. The authors should show by immunofluorescence and Western blot that the peroxisome number is really elevated.

4-Phenylbutyrate (4-PBA) was used as a chemical inducer of peroxisome biogenesis in mammalian cells. We have used 4-PBA at a concentration of 2mM for 10 days to induce peroxisome proliferation in HeLa cells. As the reviewer has pointed out, 4PBA is used frequently in rodent models where 4PBA is shown to induce peroxisomes at a lower exposure time (24-72 hours) and lower concentration (1mM). However, to induce peroxisomes, a similar concentration and duration of 4PBA is shown in human fibroblasts by Beltran et al., (PMID: 30269970). Similarly, another study by Roczkowsky et al., used 100µM of 4PBA in primary human microglia for inducing peroxisomes (PMID: 35940876).

As suggested by the reviewer, we have now tested the effect of 4-PBA by immunofluorescence and western blotting for peroxisome induction. Immunoblotting analysis indicates increased expression of peroxisome membrane protein, PEX14. Additionally, the number of peroxisomes per cell after 4-PBA treatment was significantly increased as observed by fluorescence microscopy. **The image shown below appears as AP-1K-M in the revised manuscript.**

Figure: AP-1K. IF analysis of PEX14 in 4PBA treated HeLa cells.

AP-1L. Representative microscopy image of PEX14 in 4-PBA treated HeLa cells.

AP-1M: Quantitation represents no. of peroxisomes/cell after 4PBA treatment. Statistics was performed on the mean of three independent experiments.

Figure 2J: Is the fluorescence intensity of HT115-infected control cells and Prx5-knockdown cells significantly different?

The fluorescence intensity of HT-115 infected cells and Prx5 knockdown cells are not significant. Statistical analysis is now added in the graph.

One way ANOVA multiple comparison test	Mean Diff	Summary	Adjusted P value
Control RNAi + HT115-RFP vs. Prx-5 RNAi + HT115-RFP	-10.13	ns	0.7923

Using the in silico software "PTS1 predictor" the authors queried Salmonella T3SS effector proteins to identify those that contain a putative peroxisomal targeting signal 1. The effector SseI might contain a variant of PTS1, namely "GKM". Figure 3B, S3A: To investigate if the C-terminal tripeptide GKM might be functional as PTS1 signal for the import into the peroxisomal matrix, the authors should show whether a fluorescent protein (e.g., GFP, mCherry) containing the putative C-terminal PTS1 motif [GKM and/or the 12 C-terminal amino acids (important if the putative PTS1 is not conserved)] colocalizes with peroxisomes.

We thank the reviewer for this excellent comment. The experiment done in this regard has increased our understanding of the PTS1-variant, GKM. We have now tagged EGFP to the C-terminal tripeptide, GKM (in pcDNA3.1 vector). Microscopic analysis showed localization of EGFP-GKM to peroxisomes labelled by PEX14 antibody. GFP-SKL was used as the positive control and empty GFP vector was used as the negative control.

The image below shows that EGFP-GKM puncta colocalize with PEX14 labelled peroxisomes. This indicates that ‘GKM’ might be a functional PTS1 signal. However, there is also cytoplasmic localization of EGFP-GKM in comparison to GFP-SKL, which shows efficient targeting to peroxisomes. Quantification show on an average 80% of peroxisomes are targeted by GFP-GKM. **The image below now appears as figure 3E-F in the revised manuscript.**

Figure: 3E. Representative microscopy images of HeLa cells expressing either EGFP-SKL or EGFP-GKM. 3F. Quantification for percentage colocalization of GFP-SKL / GFP-GKM with PEX14.

Interestingly, similar localization by PTS1-variants has been reported in literature multiple times. For instance, Prof. Suresh Subramani’s group has reported the targeting efficiencies of various consensus sequences of the PTS tripeptide (Swinkels *et al.*, 1992). In this study they have grouped the tripeptides into three classes based on the targeting efficiency of peroxisomes (image shown below; Panel C-D CAT-AKM, highlighted in red).

REDACTED: Fig. 1 from Swinkels BW *et al* (1992), doi: 10.1016/0014-5793(92)80880-P

We observe a similarity of GKM motif with the class II PTS which exhibits detectable targeting into peroxisomes along with cytoplasmic signal.

Human proteins containing such PTS1-variants are also reported that shows dual targeting of peroxisomes and cytosol. For instance, human epoxide hydrolase, which contains ‘SKM’ motif also shows dual cytosolic and peroxisomal localization (PMID: 8342951, 1743286).

Additionally, dual targeting proteins are difficult to study due to strong signal from cytosol. Buch *et al* used photobleaching in live cells to differentiate peroxisomal localization of bile acid-CoA:amino acid N-acyltransferase from cytoplasmic localization (PMID: 19666004).

Figure 3E, F: The authors should also show whether the complementation of SseI lacking the putative PTS1 motif also restores the growth defect of the Δ sseI strain.

We thank the reviewer for this suggestion. We have shown that infection with Δ sseI STM strain shows growth defect compared to WT STM. Now we show that, complementation of SseI lacking GKM (SseI- Δ GKM in pQE60 vector) in Δ sseI strain does not rescue the growth defect. This indicates that peroxisome localization of SseI is essential for its impact on intracellular replication. Similarly, complementing Δ sseI strain with SseI-K321A (point mutation in GKM) does not rescue the growth defect

In addition, we also complemented Δ sseI strain with SseI replacing ‘GKM’ motif with ‘SKL’. This will target the SseI into peroxisomal matrix (verified by protease protection assay). Interestingly, we observe that expression of SseI-SKL cannot rescue the growth defect of Δ sseI strain indicating that targeting SseI to peroxisome membrane is critical.

The graph below now appears as figure 3L in the revised manuscript.

Figure 3L: Changes in fold proliferation of the WT STM, Δ sseI STM, and Δ sseI STM complemented with SseI or its mutants by plasmid expression (MOI=10) in HeLa cells. CFU plotted as fold change after 16 h of infection compared to 2 h. Three independent experiments containing triplicates per experiment were performed. Each dot represents mean of an independent experiment. Statistical analyses were performed on mean of independent experiments using one-way ANOVA, ****p<0.001, **p<0.01.

Figure 3G-I: The authors state that SseI is localized "on peroxisomes". However, I would understand this to mean that SseI is localized on or in the peroxisomal membrane and not in the matrix. Consider changing the wording.

We have made the required changes.

Figure legend for 4J: The caption 'UT' denotes untreated cells. However, nothing is marked with 'UT' in panel J.

We thank the reviewer for pointing out the miss. We have now replaced ‘UT’ with ‘Vehicle control’.

Figure 5A: In Figure 1 LAMP1 was used as marker for SCVs, and in Figure 5A LAMP1 is used as marker for lysosomes to determine the % of peroxisome-lysosome colocalization. Representative images should be shown. How much overlap is there between lysosomes and SCVs?

We have used LAMP1 colocalizing with bacteria as SCV. For figure 1C only LAMP1⁺Bacteria⁺ (double positive) with PEX14 was quantitated. Whereas, in figure 5A independent LAMP1 (those that do not colocalize with bacteria) were considered for quantitation. Therefore, there is no overlap in the objects (region of interest) used for quantitation for both graphs.

For better clarity, we have re-quantitated the images to show the distribution of populations within the cells. The three populations of peroxisomes after labelling LAMP1 and bacteria are

- i. peroxisomes colocalizing with lysosomes
- ii. Peroxisomes colocalizing with SCVs
- iii. Unbound peroxisomes

Figure: Pie chart representations showing distribution of peroxisomal populations inside cells.

Figure 5C: Filipin was used to stain cholesterol and determine the colocalization with STM and LAMP1? Filipin is known to bleach easily and fast, therefore, I wonder how the authors coped with the bleaching to obtain such bright filipin staining.

We have used filipin III solution from Sigma. Filipin (50ug/ml in 10%FBS) was used along with primary antibody staining for 2 hours in room temperature incubation. This was followed by washing with 1X PBS for 3 times 5 minutes each. After secondary antibody staining, samples were analyzed immediately to avoid photo-bleaching. As we also faced issues of photobleaching, staining procedure was done on the same day of imaging. Prepared slides were stored in dark and imaged on the same day. We also used freshly thawed aliquot of filipin for every use.

Figure 5G-H: Inhibition of *de novo* cholesterol synthesis with lovastatin did not affect the rate of bacterial replication, whereas the NPC1 inhibitor U18666A, which sequesters cholesterol in lysosomes, reduced bacterial replication.

How was the experiment with lovastatin conducted? Which medium was used for the experiment with lovastatin? *De novo* cholesterol synthesis is very low in cells cultured in medium with FBS, and the effect of lovastatin would be negligible because the cells would cover their cholesterol requirements through LDL uptake. Activation of the SREBP-2 pathway increases the expression of the LDL receptor, which would increase LDL-receptor-mediated cholesterol uptake.

Is bacterial replication impaired when cells are cultured in lipoprotein-deficient serum?

Line 336: the authors state that none of the inhibitors altered cell viability, however, in two experiments cell viability was decreased more than 50% in lovastatin-treated cells. Was the treatment with lovastatin carried out in medium with 10% FBS? How long were the cells treated with lovastatin?

We indeed checked the effect of lovastatin in 10% FBS containing media. We thank the reviewer for pointing out that *de novo* synthesis of cholesterol is negligible in cells grown in media containing 10% FBS.

We also agree with the reviewer that two out of three experiments show decreased cell viability during lovastatin (1uM for 24 hours in 10% FBS containing media). As the compound shows significant cell death in our cells, we have used another 3-hydroxy-3-methylglutaryl coenzyme A (HMG-CoA reductase), Atorvastatin (1uM) at a concentration that does not affect cell viability for 24 hours (as shown by MTT assay, AP-2B).

To understand that contribution of *de novo* synthesized cholesterol during STM infection, we used chemical inhibitor of *de novo* synthesis, atorvastatin. The analysis was performed in either 1% FBS or lipoprotein free media to remove exogenous cholesterol. In both these scenarios, where LDL mediated cholesterol is restricted, we observe a reduction in the fold proliferation on STM after atorvastatin treatment. (Preparation of lipoprotein-free media is explained in the methods section. Cholesterol levels of the processed media was measured using commercially available 'LDL cholesterol direct kit', #catalog LDL080, Medsource ozone biomedical, figure a shown below).

5H**EV-4E****AP-2B****a**
Figure 5H: Graph representing the changes in fold proliferation of STM (MOI=10) after treatment with Atorvastatin (1 μ M), the media containing 1% FBS in HeLa cells. Data represent the mean \pm SEM of three independent experiments.

EV-4E: E. Fold proliferation of STM (MOI=10) in HeLa cells under LDL-free FBS conditions following Atorvastatin (1 μ M). HeLa cells were treated with Atorvastatin and DMSO as Vehicle control. Three independent experiments containing triplicates per experiment were performed. Each dot represents the mean of an independent experiment.

AP-2B. Graph representing percentage cell viability of HeLa cells after treatment with Atorvastatin measured by MTT assay.

a Cholesterol levels of the processed media was measured using commercially available 'LDL cholesterol direct kit', #catalog LDL080, Medsource ozone biomedical, figure a shown below).

This reduction in bacterial proliferation is probably due to the need for sterol intermediates that are derived from the *de novo* pathway but not cholesterol, as previously reported by kasturi *et al* (PMID: 14742551). To verify this possibility, we silenced *de novo* synthesis pathway either in its initial stage or at the final stage to differentiate the sterol intermediates from cholesterol. While silencing HMGCoA leads to complete perturbation of *de novo* pathway, silencing DHCR24, only affects cholesterol production from desmosterol. We see silencing HMG CoA (HMG-CoA reductase) affects STM proliferation but silencing DHCR24 (24-Dehydrocholesterol reductase) does not have any impact on STM proliferation.

EV-4F. Fold proliferation of STM (MOI=10) in HeLa cells under 1% FBS conditions following HMGCR silencing. HeLa cells were transfected with siRNA targeting HMGCR or scrambled siRNA (Scr) control. Cell proliferation was assessed in triplicate for three independent experiments. Data points represent mean fold proliferation per experiment.

EV-4G. Graph representing the changes in fold proliferation of STM (MOI=10) after silencing *DHCR24* in HeLa cells at 1% FBS condition. Three independent experiments containing triplicates per experiment were performed. Each dot represents the mean of an independent experiment.

EV-4H-I. Fold change in mRNA levels of *HMGCR* (H) and *DHCR24* (I) after its silencing in HeLa cells.

Additionally, we also silenced LDL-receptor (LDL-R) to reduce uptake of LDL from exogenous media and studied the effect on STM proliferation. We observe decrease in intracellular bacterial replication after silencing LDL-R

Figure EV-4C. Graph representing the changes in fold proliferation of STM (MOI=10) after silencing *LDLR* in HeLa cells. Three independent experiments containing triplicates per experiment were performed. Each dot represents the mean of an independent experiment.

EV-4D. Fold change in mRNA levels of *LDLR* after its silencing in HeLa cells.

Taken together, these results suggest that *Salmonella* acquires cholesterol from LDL processed in lysosomes. **The graphs appear as figure 5H, EV-4C-I, AP-2B in the revised manuscript.**

Does bacterial replication differ when the cells are cultivated in medium with 10 and 1% FBS?

We have carried out the CFU assay to measure the rate of proliferation in 10% and 1% FBS containing media.

We see that there is a relative reduction in the fold proliferation measured at 16 hours post infection. This supports our above data that exogenous LDL impact the growth of STM.

Figure: Change in fold proliferation of the WT STM grown in media containing 10% and 1% FBS.

We have not included the above data in the revised manuscript. Only treatment of U18 or atorvastatin during media containing 1% FBS are included (Figure 5G and 5H).

How much LDL was added to the medium containing 1% FBS?

After 30 minutes of incubation in bacteria containing media (for bacterial invasion), fresh media was replaced containing 25ug/ml of LDL. The change in fold proliferation was measured after 16 hours.

Figure 5I: Chu et al. (Cell 2015) showed that lysosomal Syt7 binds peroxisomal PIP2 to bridge the organelle contact. What is the percentage of colocalization of peroxisomes with Syt7 in STM-infected cells?

A previous study by Chu *et al* indicates that lysosomal synaptogamin VII (Syt7) is crucial for lysosome-peroxisome membrane contacts (LPMC). Syt7 is mainly colocalized with lysosome markers such as LAMP1. However, they also have significant interaction with the peroxisomal marker, PMP70 (data shown below).

REDACTED: Fig. 4A-B from Chu BB *et al* (2015), doi:10.1016/j.cell.2015.02.019

In our current study, we see a similar interaction between peroxisomes and Syt7 present on SCVs. HeLa cells expressing Syt7-FLAG (in pcDNA3.1 vector) were infected with STM and immunostained with anti-PEX14 antibody. As seen in the figure 5I, Syt7 mainly colocalizes with both lysosomes and SCVs. We observe interaction with peroxisomes marked by PEX14 (figure 5J-K). We observe an increase in the percentage colocalization of peroxisomes in STM-infected cells (figure 5L).

Figure 5I. Representative confocal micrographs of HeLa cells infected with either GFP-tagged WT STM with Syt-7 FLAG overexpressing plasmid. In Zoom, the yellow arrowhead represents the co-localization between Syt-7(red), LAMP1 (green) and STM (blue). Scale bars: 10 μm (main panel), 2 μm (inset).

5J. Representative confocal micrographs of uninfected HeLa cells with Syt-7 FLAG overexpressing plasmid. In Zoom, the yellow arrowheads represent the co-localization between Syt-7(red) and PEX14 (green). Scale bars: 10 μm (main panel), 2 μm (inset).

5K. Representative confocal micrographs of Syt-7 FLAG overexpressing HeLa cells infected with GFP-tagged WT STM (Blue). In Zoom, the yellow arrowheads represent the co-localization between Syt-7(red), PEX14 (green) and STM(Blue). Scale bars: 10 μm (main panel), 2 μm (inset).

5L. Graph representing the percentage co-localization of peroxisome-Syt-7 in HeLa cells after infection with WT STM with MOI=50 and uninfected cells. Data of three independent experiments containing more than 80 cells are shown.

Figure 6 or line 362: Please define the abbreviation MLN when it is used for the first time.

We thank the reviewer for pointing the error. We have now given the abbreviation in its first instance (line 456 in the revised manuscript).

Figure 6E-F: While complementation with SseI tagged with myc at its C-terminus is lower than complementation with SseI tagged with myc at its N-terminus, complementation with SseI-myc obviously occurs (even if it is not statistically significant). How can this be explained? I would suggest performing the complementation experiment with SseI-ΔGKM and SseI-K231A.

We thank the reviewer for this suggestion. This suggestion has helped us in better understanding the significance of GKM motif during *in vivo* infection.

It is known that tagging of PTS1 at its C-terminus prevents recognition by PEX5 receptor. Therefore, we believe that tagging GKM motif at its C-terminus prevented targeting of peroxisomes to SCVs, reducing the rescue efficiency of SseI-Myc. However, we agree with the reviewer that complementation occurs to some extent. This rescue is because of peroxisome-independent function(s) of SseI. Similarly, published literature on SseI shows that SseI-deleted strain of *Salmonella* shows

reduced *in vivo* replication. This virulence is attributed to the effect of SseI on the migration of infected macrophages (McLaughlin et al., 2009; PMID: 19956712). In the current study we show that apart from the migratory effect, SseI by its C-terminal GKM motif also targets the protein to peroxisomes.

As suggested by the reviewer, we also carried out complementation with SseI- Δ GKM and SseI-K321A to better understand the significance of GKM motif. We performed a competitive index analysis of wild-type strain with the mutant strains.

The groups are as follows:

1. WT STM vs Δ sseI STM
2. WT STM vs Δ sseI STM complemented with full length SseI (in pQE60 vector)
3. WT STM vs Δ sseI STM complemented with SseI- Δ GKM
4. WT STM vs Δ sseI STM complemented with SseI-K321A

The Δ sseI strain complemented with full length SseI displayed similar fitness with wild-type strain. However, the Δ sseI strain complemented with SseI that is devoid of GKM or carries mutation in PTS1(K321A), displays virulence attenuation. This difference in the virulence attenuation between full length complementation and Δ GKM/ K321A complementation highlights the significance of peroxisome targeting of SseI by GKM motif.

It is to be noted that virulence of Δ sseI STM complemented with Δ GKM/ K321A is higher than Δ sseI STM (group 1 vs group 3/4) because of the peroxisome-independent effect of SseI such as migration. Therefore, our result shows peroxisome targeting by SseI is crucial during *in vivo* STM infection.

Figure 6E-F is now replaced with the figure shown below.

Figure 6E-F: 5-6 weeks old C57BL/6 mice (n=9 mice/group) were orally infected with a mixed inoculum containing WT STM and either Δ sseI STM or Δ sseI STM complemented with plasmid (pQE60) expression of SseI/SseI- Δ GKM/SseI-K321A. Competitive index (CI) values were determined after 5 days of post infection, The CI was calculated between 'WT vs mutant' and 'WT vs complement strains'. Data is represented as CI of mutant and complemented strains with WT. Statistical analyses were performed using one-way ANOVA, non-parametric test, Kruskal Wallis test *p<0.05, **p<0.01.

Line 87-88: References 38 and 39 do not address cholesterol transfer from peroxisomes to the endoplasmic reticulum. The publication by Xiao et al. (2019) would be the appropriate reference (<https://doi.org/10.1007/s11427-019-9569-9>).

We thank the reviewer for suggesting the reference. We have now included the reference.

Materials and methods: The authors state that for antibody staining, permeabilization buffer (Triton X-100 (0.3 %) /1% BSA in 1X-PBS) containing primary antibody dilution was used according to the manufacturer's instructions. Which manufacturer?

Detailed explanation for antibody staining is included in the materials and methods section.

Line 670: it should be Bio-Rad instead of Bio-Red.

Correction is done.

Referee #3

In this manuscript, Raj et al attempted to address the question of how Salmonella acquires the cholesterol to establish the SCV during infection. They found a TTSS SPI2 effector SseI plays an important role in Salmonella intracellular growth and acquisition of cholesterol onto SCVs through the binding to host proteins ARF1 and PEX5. In addition, the authors demonstrate the acquisition of cholesterol in SCV requires the peroxisome-lysosome interaction mediated by synaptotagmin-7. The manuscript is well written and the authors try to clarify the relevance of the interaction between SCV, peroxisome, and lysosome from different angles. However, I feel that the authors may need to add some data to elucidate how ARF1, PEX5, and SseI coordinately work to recruit the cholesterol for the establishment of SCV during infection.

We thank the reviewer for the constructive comments and insightful suggestions. This has helped in improving the quality of the study. We have made necessary efforts to answer the queries and comments. The point-wise response is given below.

- 1. In figure 4B, SseI was found to interact with ARF1 and PEX5. I am wondering if the GKM signal at C-term of SseI is required for the interaction with ARF1 or PEX5? The authors should try to validate this point using HA-tagged SseI mutants.**

We thank the reviewer for this important suggestion. We have now used SseI-ΔGKM construct to understand the binding of SseI with ARF1 and PEX5. PTS1 motif and its variants are known to bind to PEX5 receptor to be targeted to the peroxisome matrix. We therefore hypothesize that GKM is required for its binding to PEX5.

As expected, HA-SseI-ΔGKM construct after immunoprecipitation showed binding to ARF1 but not with PEX5. This validates our hypothesis that GKM motif mediates binding with PEX5 which is essential for its peroxisome localization. **Figure 4B is now replaced with the figure shown below.**

4B

Figure 4B: Immunoblot analysis of proteins after pull down of SseI from HeLa cells expressing either HA-tagged SseI or HA-tagged SseI- Δ GKM. The cell lysates (input) were used for the pull-down of SseI using an anti-HA antibody. IgG antibody was used as a control. The membrane was probed with anti-HA, anti-ARF1, and anti-PEX5 antibodies to detect HA-SseI, endogenous ARF1, and PEX5, respectively. The result is representative of three independent experiments.

2. The data in Figure 4D doesn't demonstrate that ARF1 is activated dependently on Salmonella infection. Similarly, ARF1 seems to be recruited to peroxisome regardless of Salmonella infection in figure 1. The author should approach this kind of experiment using a tool/kit or mutant to detect specifically active form ARF1.

We thank the reviewer for this critical comment. This suggestion has helped us in better understanding the ARF1 activation during infection and Δ sseI *Salmonella* infection.

The figure 4D demonstrates the level of active ARF1 on whole cell lysate. Active ARF1 was pulled down using ARF1 activation kit (Cell Biolabs, STA-407-1). We observe that the levels of active ARF1 increased after infection with WT STM. This increase in ARF1 levels is lost during infection with SseI strain indicating that SseI acts as GEF for ARF1.

We hypothesize that the levels of active ARF1 on peroxisomes increase after wildtype *Salmonella* infection which is inhibited in infection with Δ sseI STM strain. To test our hypothesis, we have used Δ sseI STM mutants and measured levels of active ARF1 on isolated peroxisomes (Cytoskeleton, #BK032-S).

Following experimental conditions were employed to quantitate levels of active ARF1 on peroxisomes,

- (i) Uninfected sample
- (ii) Infection with wildtype strain
- (iii) Infection with Δ sseI *Salmonella* strain
- (iv) Infection with Δ sseI expressing SseI- Δ GKM by pQE60 plasmid
- (v) Infection with Δ sseI expressing SseI-SKL by pQE60 plasmid
- (vi) Infection with Δ sseI expressing full length SseI by pQE60 plasmid

Figure 4F: Immunoblot and densitometry analysis of Active ARF1 protein levels after ARF1-GTP pulldown from isolated peroxisomes of HeLa cells infected with WT or Δ SseI STM, or Δ SseI STM complemented with pQE60-SseI clones (Δ GKM, SKL, full length). Densitometry analysis represents three independent experiments.

By active-ARF1 assay, we observe that,

i-ii WT STM induces activation of ARF1 on peroxisomes.

iii. Infection with Δ SseI-STM does not induce ARF1 activation on peroxisomes indicating its significance.

iv. Infection with Δ SseI-STM complemented with pSseI- Δ GKM also do not show activation of ARF1. This suggests that GKM-mediated peroxisomal localization of SseI is essential for ARF1 activation on peroxisomes

v. Infection with Δ SseI-STM complemented with pSseI-SKL where SseI is transported into peroxisomal matrix does not activate ARF1. It is therefore essential to retain SseI on membrane for its ARF1-GEF activity.

vi. Infection with Δ SseI-STM complemented with full length SseI- Δ GKM show activation of ARF1.

The immunoblot now appears as Figure-4F in the revised manuscript.

In contrast, it would be interesting if the author could see that inactive form ARF1 is not included on isolated peroxisomes.

In most cases, ARFs in GDP bound form (inactive) are cytosolic and GTP-ARFs (active) are membrane bound. To understand if inactive form of ARF1 is present on peroxisomes, we performed treatment with a fungal toxin, Brefeldin A (BFA) that directly binds to Sec7 protein domain. ARF-GEFs are multidomain proteins and its guanine nucleotide-exchange activity lies in Sec7 domain. Therefore, treatment with BFA leads to accumulation of intermediates and prevents the formation of GTP-ARF1.

As seen in the figure below, there is presence of ARF1 after BFA treatment on the isolated peroxisomes by immunoblotting analysis. However, we cannot conclude that it is inactive ARF1 because there are certain BFA insensitive ARF-GEFs reported in the literature. For instance, low molecular weight ARF-GEFs contain subset of Sec7 domain (ARNO, GBF1, cytohesin1) which has little/no sensitivity to BFA (PMID: 10652516).

Therefore, we cannot conclude whether or not ARF1-GDP is excluded from peroxisomes and therefore did not add this data in the manuscript. This remains to be explored and the limitation is included in the discussion of the manuscript.

Figure: Immunoblotting analysis of total ARF1 after isolating peroxisomes during Brefeldin A treatment.

3. I am curious whether Synaptotagmin-7 can be detected on peroxisomes in a Salmonella infection- or SseI-dependent manner.

In order to answer the query if Syt7 is present on peroxisomes after infection, we infected HeLa cells and isolated peroxisomes. We do not detect Syt7 on isolated pure peroxisomes indicating that peroxisomes do not contain Syt7. Extensive washing for 3 times with 1X KPBS (136 mM KCl, 10 mM, KH₂PO₄, pH 7.25) is performed to dissociate peroxisome and other inter-organelle contacts. In case of no washing, we see slight presence of Syt7 probably due to contamination from lysosomes or SCVs.

Figure EV-4M: Immunoblotting analysis of Syt7 after isolating peroxisomes during STM infection.

Additionally, we also performed fluorescence microscopy analysis to understand the interaction of Syt7 with peroxisomes. We over expressed Syt7-FLAG (in pcDNA3.1 vector) in HeLa cells to understand the interaction. As seen in the images below, Syt7 and LAMP1 show complete colocalization whereas Syt7 and PEX14 interact partially. This observation is similar to the study by Chu et al (Cell 2015) that showed interaction of lysosomal Syt7 with peroxisomal PIP2.

Figure 5I. Representative confocal micrographs of HeLa cells infected with either GFP-tagged WT STM with Syt-7 FLAG overexpressing plasmid. In Zoom, the yellow arrowhead represents the co-localization between Syt-7(red), LAMP1 (green) and STM (blue). Scale bars: 10 μ m (main panel), 2 μ m (inset).

5J. Representative confocal micrographs of uninfected HeLa cells with Syt-7 FLAG overexpressing plasmid. In Zoom, the yellow arrowheads represent the co-localization between Syt-7(red) and PEX14 (green). Scale bars: 10 μ m (main panel), 2 μ m (inset).

5K. Representative confocal micrographs of Syt-7 FLAG overexpressing HeLa cells infected with GFP-tagged WT STM (Blue). In Zoom, the yellow arrowheads represent the co-localization between Syt-7(red), PEX14 (green) and STM(Blue). Scale bars: 10 μ m (main panel), 2 μ m (inset).

4. To support data in Figure 4I, the author may try experiments, which are similar to Figure 4J-L, using PI5P4K2B inhibitor.

Figure 4I indicates that level of PI5P4K2B do not change during WT/ Δ SseI STM infection. There are no specific inhibitors for the PI5P4K isoforms- α, β, γ encoded by three different genes, namely, *PI5P4K2A*, *PI5P4K2B*, and *PI5P4K2C*. To decisively show that PI5P4K2B does not impact STM growth, we now silenced *PI5P4K2A* or *PI5P4K2B* separately and monitored the intracellular replication. As seen in figure, only silencing *PI5P4K2A* reduced the STM proliferation, whereas silencing *PI5P4K2B* did not affect the bacterial replication. **The graphs now appear as figure 4I-J, EV-3H-I in the revised manuscript.**

Figure: Changes in fold proliferation of the WT STM after silencing K2A (A) or K2B (C). Silencing efficiency was estimated by immunoblotting analysis (B,D).

5. As for the result summary described as a model in supplementary figure 6, it well represents the dynamics of LDL-derived cholesterol compared between a WT host cell vs a PEX5 KO cell over

the salmonella infection. However, this model doesn't reflect the manuscript title. I feel the author should try to describe the dynamics of LDL-derived cholesterol, peroxisome-SCV-lysosome interactions, and the roles of ARF and PEX5 in a host cell infected with WT STM or Δ *SseI* mutant (meaning in the presence or absence of *SseI*) because the biological significance of *Salmonella* effector *SseI* during infection is not well pictured in the current figure. It would be better if the GKM signal at C-terminus of *SseI* is also described.

We thank the reviewer for the suggestion. We have now improved the model significantly. We have emphasized on the peroxisome-SCV-lysosome interactions, and the roles of ARF and PEX5 in a host cell infected with WT STM or Δ *SseI* mutant. **The schematic now appears as figure 6G in the revised manuscript.**

6. In Figure 5C, HeLa cells as hosts is labeled as "WT". I think it is better to indicate "HeLa" instead of "WT", and PEX5 KO may be replaced with HeLa PEX5 KO. The authors describe the figure legend properly, but I took a while to recognize that "WT" described in the images means HeLa because I thought it was WT STM. In addition, Figure 5D also seems to require the correction as well. The author should indicate which host cell is infected by which bacterial strain so that readers can avoid misreading the data.

We have now indicated the cell type as 'HeLa' or 'PEX5 KO HeLa'. Figure 5D is also re-labelled for better clarity on cell type and bacterial strain used.

Minor comments

1. In supplementary figures 5A and 5C, NCP1 should be corrected as NPC1.
2. The labels of the Y axis of a graph, which represents the fold change of *Salmonella* proliferation in the presence of ABCD1, are too small.
3. Line 362: Please place the abbreviation of MLN, but not in Line 707.
4. It is difficult for me to recognize the "Merge" described using mixed-colored fonts. I think it's better to use black for Merge (Fig1F, Fig2K, Fig5I, and Supp.Fig3B).

We thank the reviewer for the pointing the errors in labelling. We have made appropriate changes.

Dear Amit,

Thank you for the submission of your revised manuscript to EMBO reports. I apologize for the delay in handling your manuscript. As you know, we have received the full set of referee reports and all referees are positive about the study and support publication pending some minor changes to clarify text and figures.

From the editorial side, there are also a few things that we need before we can proceed with the official acceptance of your study.

- Please provide up to 5 keywords.
- Please update the 'Conflict of interest' paragraph to our new 'Disclosure and competing interests statement'. For more information see <https://www.embopress.org/page/journal/14693178/authorguide#conflictsofinterest>
- We noticed an author name discrepancy that needs to be rectified: Aman Chandra Kaushik in the manuscript file vs. Aman Kaushik in the online manuscript tracking system. Please correct either the name in the manuscript file or the one in the online system.
- Please note that all corresponding authors must provide an institutional e-mail address. This is currently missing for the ; co-corresponding author, Dr. Veena Ammanathan.
- Regarding the Author Contributions, we now use CRediT to specify the contributions of each author in the journal submission system. Therefore, please remove the Author Contributions from the manuscript file and make sure that the author contributions in our online manuscript tracking system are correct and up-to-date. The information you specified in the system will be automatically retrieved and typeset into the article. You can enter additional information in the free text box provided, if you wish.
- The refereneces need to be alphabetical instead of numerical; et al needs to be used after 10 author names.
- The information on funding needs to be congruent between the manuscript text and the information in the online manuscript tracking system. We note that only one funder is listed in the system. Moreover, the following also need to be provided in our system:
 - the Science and Engineering Board (SERB), Government of India (SRG/2019/000268)
 - ICMR Ad- hoc(Ortho) 20202-NCD-1, CSIR-MLP 2105, CSIR HCP-0047
 - start-up funding from Director CSIR-CDRI and PDF/2021/002843 (SERB), IA/E/21/1/506319 (DBT/Wellcome Trust IA)
 - fellowship from CSIR
 - fellowship from UGC, the government of India
- Please add callouts for Figure EV1 and the figures from the Appendix in the manuscript text, where appropriate.
- Figure 1: the line intensity graphs on the right of the figure have a rather low resolution. Can this be enhanced?
- Figure 2: cosmetic, but the scale bars in panel K do not align well.
- The scale bar of the zoomed images in Figure 5C is very faint and might be difficult to see.
- Appendix: please correct the nomenclature and the figure callouts in the manuscript to Appendix Figure S1-S2 (instead of Appendix-1 and Appendix -2).
- Appendix: please add a title page with a table of content (with page numbers).
- Legend of Appendix Figure S1: The descriptions for panels (A-B) and (I) are not aligned at the left margin in contrast to the other panel legends.
- Legends of Appendix figures: Please add information on "n" (technical or biological repliates) for all panels that show quantifications, define the bars and error bars, and please provide the exact p-values instead of e.g. *p<0.05.
- Appendix Figure S2: You add abbreviations ofr e.g., DHCR24(R) but the (R) is not shown in the figure panel. Are these one-letter codes required here?
- Movies: the correct nomenclature should be Movie EV1; the legend should be removed from the manuscript and provided as a readme.txt file and then both the legend and the movie should be zipped up and uploaded as one folder called "Movie EV1".

- Please remove the Reagents and Tools table from the manuscript and upload it as separate file (type Reagent Table). "Methods and Material" should be called "Methods"

- Please upload the source data as one folder per figure instead of the current single ZIP file for all figures.

- The manuscript sections should be in the following order: Title page - Abstract & Keywords - Introduction - Results - Discussion - Methods - Data Availability - Acknowledgments - Disclosure Statement & Competing Interests - References - Figure Legends - (Main Tables with legends) - Expanded View Figure Legends.

- Our production/data editors have asked you to clarify several points in the figure legends (see below). Please incorporate these changes in the manuscript and return the revised file with tracked changes with your final manuscript submission.

A) Statistical test information. Only p-values that are actually shown in the figure panel(s) should (and must) be defined in the legends, all others should be removed from (or added to) the legend. Moreover, we ask for the specification of exact p-values:

- Please note that the exact p values are not provided in the legends of figures 1c-d, g; 2g-j, l; 3d, g, i, l; 4e-i, k-m; 5a-b, d-h, l-m, o; 6a-f; EV 1c-e, i-j; EV 2b; EV 3c, f, h-i; EV 4a-f, h-k.

- Please note that in figure EV 3h; there is a mismatch between the annotated p values in the figure legend and the annotated p values in the figure file that should be corrected.

B) Replicates and error bars:

- Please note that information related to n is missing in the legends of figures EV 1a-b, e, j, l-n, j; EV 3c-d, h-l; EV 4a-b, d, h-k.

- Please note that the error bars are not defined in the legends of figures 1c-d, g; 2b-f, l; 3f, i; 4f, h; 5a, l; EV 1a-b, e, j, l-n; EV 3c-d, f-l; EV 4a-k.

C) Data presentation:

- Please note that for heatmap present in figure 2a; a numbered scale bar is not provided. This needs to be rectified.

- Please provide a complete author checklist, which you can download from our author guidelines

(<<https://www.embopress.org/page/journal/14693178/authorguide>>). Please insert information in the checklist that is also reflected in the manuscript. The completed author checklist will also be part of the RPF.

- We perform a routine analysis of all quantification .xls files. In this case I noticed that in a few cases, the same numbers up to their 6th decimal appear 2 or 3 times in the same quantification sheet. Please double-check these quantifications and numbers (color-coded files attached).

- Finally, EMBO Reports papers are accompanied online by

A) a short (1-2 sentences) summary of the findings and their significance,

B) 2-3 bullet points highlighting key results and

C) a schematic summary figure that provides a sketch of the major findings (not a data image).

Please provide the summary figure as a separate file in PNG or JPG format at a size of 550x300-600 pixels (width x height).

Please note that the size is rather small and that text needs to be readable at the final size. Please send us this information along with the revised manuscript.

- On a different note, I would like to alert you that EMBO Press offers a new format for a video-synopsis of work published with us, which essentially is a short, author-generated film explaining the core findings in hand drawings, and, as we believe, can be very useful to increase visibility of the work. This has proven to offer a nice opportunity for exposure i.p. for the first author(s) of the study. Please see the following link for representative examples and their integration into the article web page:

<https://www.embopress.org/doi/full/10.15252/embj.2019103932>

I look forward to seeing a new revised version of your manuscript as soon as possible.

Kind regards,

Martina

Martina Rembold, PhD
Senior Editor

=====

Referee #1:

The authors have painstakingly addressed all my concerns. The excellent manuscript is now ready for publication.

Referee #2:

The authors have addressed the reviewers' concerns and significantly improved the quality of the manuscript significantly. Therefore, I support the publication of this manuscript.

However, there is one point on which more clarity should be provided. The authors state that "these results suggest that Salmonella acquires cholesterol from LDL processed in lysosomes and not de novo pathway (line 394). By knocking down LDLR, NPC1 and ABCD1 and using the NPC1 inhibitor U18666A, the authors clearly show that cholesterol from LDLR-mediated endocytosis and transfer of cholesterol from the lysosome is required for STM proliferation. To examine the contribution of de novo cholesterol biosynthesis during STM proliferation, the rate-limiting enzyme of cholesterol synthesis, HMGCR, was knocked down or inhibited with atorvastatin, resulting in reduced STM proliferation. The experiment was performed in either 1% FBS or lipoprotein-free media, where cells depend on de novo cholesterol synthesis. In contrast, knockdown of DHCR24, which catalyzes the final step of cholesterol synthesis (Bloch pathway), did not affect STM proliferation. Therefore, the authors reasoned that sterol intermediates other than cholesterol are required for STM proliferation, as suggested by Catron et al. (2004). However, de novo cholesterol synthesis could also occur via the Kandutsch-Russell pathway using 7-dehydrocholesterol reductase (DHCR7). However, in the response to the reviewers, the authors present a figure showing that proliferation of WT STM grown in 1% FBS is reduced when compared to 10% FBS. This nicely demonstrates that LDL-derived exogenous cholesterol is indeed necessary for STM proliferation. Therefore, I would recommend showing this figure in the manuscript.

Referee #3:

In the revised manuscript, the authors answered all my suggestions. The newly added results indicate a molecular mechanism of how a Salmonella effector Ssel contributes to the interaction of peroxisomes with SCVs. I think the manuscript is almost ready for publication. There are some minor comments as follows.

1. For the table in the Methods & materials section, some parts are incompleting such as the ATCC number of HeLa, and reference of Salmonella strain. Please add the information.
2. The information on how the Salmonella strain is maintained and grown before the infection experiments are lacking in the manuscript. Please add the details in the methods section.
3. As for the experiments regarding Figure 4D, there are no descriptions of the experimental procedure including the used kit, ARF1 activation kit (Cell Biolabs, STA-407-1).
4. In the previous version of the manuscript, the Figure 4 legends described the details such as MOI used for each experiment. But they are missing in the revised manuscript. Please add the details as previously.
5. In all updated figures, different types of fonts are used. Please choose a single font. I like Arial or Helvetica as previously (Pre-revised manuscript figures looked good).
6. Figure 2A: difficult to recognize 12 hours, 6hours, uninfected...I liked the previous version. Also, the authors should provide all values of individual replicates and Z scale transformed values as an appendix figure or supporting data.
7. Figure 6G: This is a great figure describing all new findings in this study. It would be better if the author could use bigger fonts if possible.

‘Salmonella Typhimurium effector SseI regulates peroxisomal dynamics to acquire lysosomal cholesterol for better intracellular growth’(EMBOR-2024-59092V3)

Dear Amit,

Thank you for the submission of your revised manuscript to EMBO reports. I apologize for the delay in handling your manuscript. As you know, we have received the full set of referee reports and all referees are positive about the study and support publication pending some minor changes to clarify text and figures.

From the editorial side, there are also a few things that we need before we can proceed with the official acceptance of your study.

We sincerely thank the Editorial board members and reviewers for supporting the publication of our manuscript. We have made appropriate changes in the manuscript as suggested by the editorial board and reviewers.

The following is the point-wise response for the changes suggested to the editorial board and reviewers

- Please provide up to 5 keywords.

The following 5 keywords are added to the manuscript.

‘Peroxisomes, *Salmonella* Typhimurium, Cholesterol, peroxisome-targeting-sequence, Arf1 activation’.

- Please update the 'Conflict of interest' paragraph to our new 'Disclosure and competing interests statement'. For more information see

<https://www.embopress.org/page/journal/14693178/authorguide#conflictsofinterest>

We have now updated the ‘conflict of interest’ to ‘Disclosure and competing interests statement’.

- We noticed an author name discrepancy that needs to be rectified: Aman Chandra Kaushik in the manuscript file vs. Aman Kaushik in the online manuscript tracking system. Please correct either the name in the manuscript file or the one in the online system.

We have now rectified the error. In both manuscript and online tracking system, it is ‘Aman Chandra Kaushik’.

- Please note that all corresponding authors must provide an institutional e-mail address. This is currently missing for the ; co-corresponding author, Dr. Veena Ammanathan.

We have now updated the institutional email ID for the co-corresponding author, Dr. Veena Ammanathan (veena.pdf@cdri.res.in)

- Regarding the Author Contributions, we now use CRediT to specify the contributions of each author in the journal submission system. Therefore, please remove the Author Contributions from the manuscript file and make sure that the author contributions in our online manuscript tracking system are correct and up-to-date. The information you specified in the system will be automatically retrieved and typeset into the article. You can enter additional information in the free text box provided, if you

wish.

We have updated the author contributions in CRediT and removed the existing 'author contributions' section from the manuscript.

- The references need to be alphabetical instead of numerical; et al needs to be used after 10 author names.

References now appear alphabetical in the manuscript.

- The information on funding needs to be congruent between the manuscript text and the information in the online manuscript tracking system. We note that only one funder is listed in the system.

Moreover, the following also need to be provided in our system:

- the Science and Engineering Board (SERB), Government of India (SRG/2019/000268)

- ICMR Ad- hoc(Ortho) 20202-NCD-1, CSIR-MLP 2105, CSIR HCP-0047

- start-up funding from Director CSIR-CDRI and PDF/2021/002843 (SERB), IA/E/21/1/506319 (DBT/Wellcome Trust IA)

- fellowship from CSIR

- fellowship from UGC, the government of India

The information regarding funding is updated in the online tracking system.

-Please add callouts for Figure EV1 and the figures from the Appendix in the manuscript text, where appropriate.

- Figure 1: the line intensity graphs on the right of the figure have a rather low resolution. Can this be enhanced?

- Figure 2: cosmetic, but the scale bars in panel K do not align well.

- The scale bar of the zoomed images in Figure 5C is very faint and might be difficult to see.

- Appendix: please correct the nomenclature and the figure callouts in the manuscript to Appendix Figure S1-S2 (instead of Appendix-1 and Appendix -2).

- Legend of Appendix Figure S1: The descriptions for panels (A-B) and (I) are not aligned at the left margin in contrast to the other panel legends.

- Appendix: please add a title page with a table of content (with page numbers).

All the above suggestions regarding alignment, scale bars, figure call-outs and title page for appendix have been incorporated.

- Legends of Appendix figures: Please add information on "n" (technical or biological replicates) for all panels that show quantifications, define the bars and error bars, and please provide the exact p-values instead of e.g. $p < 0.05$.

The information regarding technical and biological replicates, as well as exact p-values, are included in the legend.

- Appendix Figure S2: You add abbreviations of e.g., DHCR24(R) but the (R) is not shown in the figure panel. Are these one-letter codes required here?

We have made the corrections.

- Movies: the correct nomenclature should be Movie EV1; the legend should be removed from the manuscript and provided as a readme.txt file and then both the legend and the movie should be zipped

up and uploaded as one folder called "Movie EV1".

We have now uploaded a zipped file as one folder containing legend.txt file and movie.

- Please remove the Reagents and Tools table from the manuscript and upload it as separate file (type Reagent Table). "Methods and Material" should be called "Methods"

The reagent table is now added as a separate file. The methods and material section is replaced as 'methods'.

- Please upload the source data as one folder per figure instead of the current single ZIP file for all figures.

We have now uploaded the source data accordingly.

- The manuscript sections should be in the following order: Title page - Abstract & Keywords - Introduction - Results - Discussion - Methods - Data Availability - Acknowledgments - Disclosure Statement & Competing Interests - References - Figure Legends - (Main Tables with legends) - Expanded View Figure Legends.

The manuscript has been re-arranged in the given order.

- Our production/data editors have asked you to clarify several points in the figure legends (see below). Please incorporate these changes in the manuscript and return the revised file with tracked changes with your final manuscript submission.

A) Statistical test information. Only p-values that are actually shown in the figure panel(s) should (and must) be defined in the legends, all others should be removed from (or added to) the legend. Moreover, we ask for the specification of exact p-values:

- Please note that the exact p values are not provided in the legends of figures 1c-d, g; 2g-j, l; 3d, g, i, l; 4e-i, k-m; 5a-b, d-h, l-m, o; 6a-f; EV 1c-e, i-j; EV 2b; EV 3c, f, h-i; EV 4a-f, h-k.

- Please note that in figure EV 3h; there is a mismatch between the annotated p values in the figure legend and the annotated p values in the figure file that should be corrected.

B) Replicates and error bars:

- Please note that information related to n is missing in the legends of figures EV 1a-b, e, j, l-n, j; EV 3c-d, h-l; EV 4a-b, d, h-k.

- Please note that the error bars are not defined in the legends of figures 1c-d, g; 2b-f, l; 3f, i; 4f, h; 5a, l; EV 1a-b, e, j, l-n; EV 3c-d, f-l; EV 4a-k.

C) Data presentation:

- Please note that for heatmap present in figure 2a; a numbered scale bar is not provided. This needs to be rectified.

All the above suggestions have been incorporated.

- Please provide a complete author checklist, which you can download from our author guidelines (<<https://www.embopress.org/page/journal/14693178/authorguide>>). Please insert information in the checklist that is also reflected in the manuscript. The completed author checklist will also be part of the RPF.

The filled author checklist is uploaded.

- We perform a routine analysis of all quantification .xls files. In this case I noticed that in a few cases, the same numbers up to their 6th decimal appear 2 or 3 times in the same quantification sheet. Please double-check these quantifications and numbers (color-coded files attached).

We have now verified the data and either removed/replaced the repeating values. Values that are '0' do not indicate repetition, but cells that do not have any co-localization for the proteins tested. Therefore, those values are retained.

- Finally, EMBO Reports papers are accompanied online by

A) a short (1-2 sentences) summary of the findings and their significance,

B) 2-3 bullet points highlighting key results and

C) a schematic summary figure that provides a sketch of the major findings (not a data image).

Please provide the summary figure as a separate file in PNG or JPG format at a size of 550x300-600 pixels (width x height). Please note that the size is rather small and that text needs to be readable at the final size. Please send us this information along with the revised manuscript.

A) Summary

The current study shows that *Salmonella* infection induces transient interactions between Salmonella Containing Vacuoles (SCVs) and peroxisomes. **This is the first report of a bacterial protein containing a putative peroxisomal targeting sequence-1 (PTS1), and it mediates the transport of lysosomal cholesterol to the SCV via the peroxisome.**

Bullet points

- A *Salmonella* protein, SseI, contains a Peroxisome targeting sequence-1 (PTS-1) motif that mediates the interaction of SCVs with peroxisomes.
- This interaction is crucial for the accumulation of LDL-derived cholesterol on SCVs needed for its growth and sif formation.
- SseI activates a host GTPase, ARF1, to induce PIP2 levels on peroxisomes. PIP2 facilitates the interaction of peroxisomes with SCVs using Syt7 as the tethering protein on SCVs.

B) Schematic

A separate JPEG file illustrating the key findings of the study is uploaded in the mentioned size and resolution.

Referee #1:

The authors have painstakingly addressed all my concerns. The excellent manuscript is now ready for publication.

We sincerely thank the reviewer for accepting the manuscript.

Referee #2:

The authors have addressed the reviewers' concerns and significantly improved the quality of the manuscript significantly. Therefore, I support the publication of this manuscript.

However, there is one point on which more clarity should be provided. The authors state that "these results suggest that *Salmonella* acquires cholesterol from LDL processed in lysosomes and not de novo pathway (line 394). By knocking down LDLR, NPC1 and ABCD1 and using the NPC1 inhibitor U18666A, the authors clearly show that cholesterol from LDLR-mediated endocytosis and transfer of

cholesterol from the lysosome is required for STM proliferation. To examine the contribution of de novo cholesterol biosynthesis during STM proliferation, the rate-limiting enzyme of cholesterol synthesis, HMGCR, was knocked down or inhibited with atorvastatin, resulting in reduced STM proliferation. The experiment was performed in either 1% FBS or lipoprotein-free media, where cells depend on de novo cholesterol synthesis. In contrast, knockdown of DHCR24, which catalyzes the final step of cholesterol synthesis (Bloch pathway), did not affect STM proliferation. Therefore, the authors reasoned that sterol intermediates other than cholesterol are required for STM proliferation, as suggested by Catron et al. (2004). However, de novo cholesterol synthesis could also occur via the Kandutsch-Russell pathway using 7-dehydrocholesterol reductase (DHCR7). However, in the response to the reviewers, the authors present a figure showing that proliferation of WT STM grown in 1% FBS is reduced when compared to 10% FBS. This nicely demonstrates that LDL-derived exogenous cholesterol is indeed necessary for STM proliferation. Therefore, I would recommend showing this figure in the manuscript.

We thank the reviewer for accepting the manuscript. As suggested by the reviewer, we have now included the graph indicating changes in intracellular growth proliferation of STM in media containing 1% and 10% FBS (Figure EV-4J).

Referee #3:

In the revised manuscript, the authors answered all my suggestions. The newly added results indicate a molecular mechanism of how a *Salmonella* effector SseI contributes to the interaction of peroxisomes with SCVs. I think the manuscript is almost ready for publication. There are some minor comments as follows.

We thank the reviewer for accepting the manuscript. We have also incorporated the minor comments suggested in the manuscript.

1. For the table in the Methods & materials section, some parts are incompleting such as the ATCC number of HeLa, and reference of *Salmonella* strain. Please add the information.

We have added the missing information in the table.

2. The information on how the *Salmonella* strain is maintained and grown before the infection experiments are lacking in the manuscript. Please add the details in the methods section.

The following information is added in the methods section

Bacterial Strains and Growth Condition: The *Salmonella enterica* serovar Typhimurium (STM WT) wild-type strain ATCC 14028s, generously provided by Prof. Michael Hensel. This strain was cultivated in Luria broth (LB-Himedia) overnight under vigorous shaking (180 rpm) at 37 °C using an orbital shaker (primary culture). This was followed by secondary culture incubation for 3 h at a rpm of 180 at 37 °C (1:33 ratio inoculum). Antibiotic selection such as Kanamycin/Chloramphenicol/Ampicillin was incorporated into the growth medium at final working concentrations of 50 µg/ml, 20 µg/ml, and 50 µg/ml, respectively as per the requirement.

3. As for the experiments regarding Figure 4D, there are no descriptions of the experimental procedure including the used kit, ARF1 activation kit (Cell Biolabs, STA-407-1).

The following information is added in the methods section

ARF1 Activation Assay: HeLa cells (2×10^7 cells per group) were transfected with the PEROXO-tag (3XMyC-EGFP-PEX26). After 48 hours, the cells were used for peroxisome isolation via PEROXO-IP as described previously. Isolated peroxisome samples and whole cell lysate were used for pulldown of Active ARF1 levels according to the manufacturer's instructions using the ARF1 activation Assay Biochem kit (Cat.# BK032-S) and the ARF1 activation kit (Cell Biolabs, STA-407-1).

4. In the previous version of the manuscript, the Figure 4 legends described the details such as MOI used for each experiment. But they are missing in the revised manuscript. Please add the details as previously.

MOI details are added in figure 4 legends

5. In all updated figures, different types of fonts are used. Please choose a single font. I like Arial or Helvetica as previously (Pre-revised manuscript figures looked good).

The font type for the figures are now changed to Arial.

6. Figure 2A: difficult to recognize 12 hours, 6hours, uninfected...I liked the previous version. Also, the authors should provide all values of individual replicates and Z scale transformed values as an appendix figure or supporting data.

We now replaced the labelling to match the previous version and included replicate values in Appendix

7. Figure 6G: This is a great figure describing all new findings in this study. It would be better if the author could use bigger fonts if possible.

Font size is increased in the schematic of summary.

Mr. AMIT LAHIRI
CSIR-CENTRAL DRUG RESEARCH INSTITUTE
SITAPUR ROAD
LUCKNOW, UP 226031
India

Dear Amit,

I am very pleased to accept your manuscript for publication in the next available issue of EMBO reports. Thank you for your contribution to our journal.

Kind regards,

Martina
